# JUCAL: Jointly Calibrating Aleatoric and Epistemic Uncertainty in Classification Tasks

## Abstract

We study post-calibration uncertainty for trained ensembles of classifiers. Specifically, we consider both aleatoric uncertainty (i.e., label noise) and epistemic uncertainty (i.e., model uncertainty). Among the most popular and widely used calibration methods in classification are temperature scaling (i.e., *pool-then-calibrate*) and conformal methods. However, the main shortcoming of these calibration methods is that they do not balance the proportion of aleatoric and epistemic uncertainty. Nevertheless, not balancing epistemic and aleatoric uncertainty can lead to severe misrepresentation of predictive uncertainty, i.e., can lead to overconfident predictions in some input regions while simultaneously being underconfident in other input regions. To address this shortcoming, we present a simple but powerful calibration algorithm *Joint Uncertainty Calibration (JUCAL)* that jointly calibrates aleatoric and epistemic uncertainty. JUCAL jointly calibrates two constants to weight and scale epistemic and aleatoric uncertainties by optimizing the *negative log-likelihood (NLL)* on the validation/calibration dataset. JUCAL can be applied to any trained ensemble of classifiers (e.g., transformers, CNNs, or tree-based methods), with minimal computational overhead, without requiring access to the models' internal parameters. We experimentally evaluate JUCAL on various text classification tasks, for ensembles of varying sizes and with different ensembling strategies. Our experiments show that JUCAL significantly outperforms SOTA calibration methods across all considered classification tasks, reducing NLL and predictive set size by up to 15% and 20%, respectively. Interestingly, even applying JUCAL to an ensemble of size 5 can outperform temperature-scaled ensembles of size up to 50 in terms of NLL and predictive set size, resulting in up to 10 times smaller inference costs. Thus, we propose JUCAL as a new go-to method for calibrating ensembles in classification.

## 1 Introduction

*Machine learning (ML)* systems have been widely adopted in various applications, and the rate of adoption is only increasing with recent advancements in generative *artificial intelligence (AI)* (Bick et al., 2024). *Deep learning (DL)* models, often at the core of ML systems, can learn meaningful representations by mapping complex high-dimensional data to lower-dimensional feature spaces (LeCun et al., 2015). However, many DL frameworks only provide point predictions without accompanying uncertainty estimates, which poses significant challenges in high-stakes decision-making scenarios (Kendall & Gal, 2017; Weissteiner et al., 2023).

Uncertainty in ML is commonly categorized into *aleatoric* and *epistemic* uncertainty (Der Kiureghian & Ditlevsen, 2009; Liu et al., 2019; Hüllermeier & Waegeman, 2021; Kendall & Gal, 2017). *Aleatoric uncertainty* refers to the inherent randomness in the data-generating process, such as noise or class overlap, which cannot be reduced by collecting more training observations and is therefore often considered irreducible[1]. In contrast, *epistemic uncertainty*, also referred to as *model uncertainty*,

---

[1]In practice, aleatoric uncertainty can sometimes be reduced by reformulating the problem, e.g., by including additional informative covariates. For example, a model predicting whether houses will sell within a month based only on price and square footage faces high aleatoric uncertainty. Many houses with identical features have different outcomes. Adding a covariate like location can explain much of this variance, reducing the average aleatoric uncertainty across the dataset.

captures the model's lack of knowledge about the data-generating process, typically arising from limited number of training observations. It is considered reducible through collecting additional training observations or by incorporating stronger inductive biases, such as priors or architectural constraints. For more details on these concepts, see Appendix A.

While we adopt the conventional distinction between aleatoric and epistemic uncertainty, we note that this dichotomy reflects a theoretical abstraction. In real-world data science workflows, uncertainty arises from a broader range of sources, including modeling choices, data collection, data preprocessing, and domain assumptions. While most aspects of modeling choices fall into the category of epistemic uncertainty, some aspects of the data collection process and imputation methods for missing values do not always fit well into either of these two categories. The *Predictability-Computability-Stability (PCS)* framework for veridical data science offers a more comprehensive view of the *data science life cycle (DSLC)* and highlights the importance of stability in analytical decisions (Yu, 2020; Yu & Barter, 2024). Appendix C.1 provides more details on PCS and how it relates to JUCAL.

In classification, *neural networks (NNs)* typically output class probabilities via the softmax outputs. However, modern NNs often yield poorly calibrated probabilities, where the predicted confidence scores do not reliably reflect the true conditional likelihoods of the labels (Guo et al., 2017). Calibration, therefore, is critical to ensure that uncertainty estimates are meaningful and trustworthy, particularly in high-stakes or safety-critical applications (Naeini et al., 2015; Kuleshov et al., 2018). In the PCS framework (Yu, 2020), calibration directly supports the *Predictability* principle, acting as a statistical reality check to ensure that model outputs are well aligned with empirical results.

Calibration can prevent a model from being *on average* too over- or underconfident on a given dataset. However, a more challenging task is to develop models that accurately adapt their uncertainty for *different* data points. For example, in the absence of strong prior knowledge, one would expect higher epistemic uncertainty for inputs that are far *out-of-distribution* (OOD), where predictive accuracy typically deteriorates (Garg et al., 2022; Heiss et al., 2021).[2] Conversely, lower epistemic uncertainty is expected for inputs densely surrounded by training data. However, modern NNs typically do not exhibit this sensitivity: softmax outputs tend to remain overconfident far from the training data, and standard calibration techniques cannot change the relative ranking of uncertainties across inputs (see Figure 1(a)). As a result, even calibrated softmax outputs are often overconfident OOD and underconfident in-distribution (while achieving marginal calibration averaged over the validation set).

Although many methods exist for uncertainty estimation in DL, Gustafsson et al. (2020) suggest that *deep ensembles* (DEs), introduced by Lakshminarayanan et al. (2017), should be considered the go-to method. Additionally to incorporating aleatoric uncertainty via softmax outputs, DEs also incorporates epistemic uncertainty via ensemble diversity (which is typically higher OOD). They achieve this simply by averaging the softmax outputs of multiple trained NNs. However, they are inherently not well-calibrated (Kumar et al., 2022; Rahaman et al., 2021; Wu & Gales, 2021).

Again, standard post-hoc calibration techniques, such as conformal methods (Angelopoulos & Bates, 2021) or the *pool-then-calibrate* temperature scaling approach (Rahaman et al., 2021), mitigate the tendency of DEs to be *on average* too under- or overconfident; however, they do not address the balancing of aleatoric and epistemic uncertainty during calibration. The epistemic uncertainty's dependency on its hyperparameters can be highly unstable. For example, Yu & Barter (2024); Agarwal et al. (2025) recommend training every ensemble member on a different bootstrap sample of the data. This increases the ensemble's diversity and thus the estimated epistemic uncertainty. On the other hand, Lakshminarayanan et al. (2017) recommend training every ensemble member on the whole training dataset, which is expected to reduce the diversity of the ensemble. Also, other hyperparameters such as batch-size, weight-decay, learning-rate, dropout-rate, and initialization affect the diversity of the ensemble. In practice, all these hyperparameters are usually chosen without considering the ensemble diversity, and we cannot expect that they result in the right amount of epistemic uncertainty. There is also no reason to believe that the miscalibration of DEs' aleatoric and DEs' epistemic uncertainty has to be aligned: For example, if we regularize too much, DEs usually overestimate the aleatoric uncertainty and underestimate the epistemic uncertainty. In such cases, decreasing the temperature of the predictive distribution results in overconfident OOD predictions,

---

[2]This behavior depends on the assumptions encoded in the model (prior knowledge). For example, if the true logits are known to be a linear function of $x$, extrapolation beyond the training domain in certain directions may be justified with high confidence.

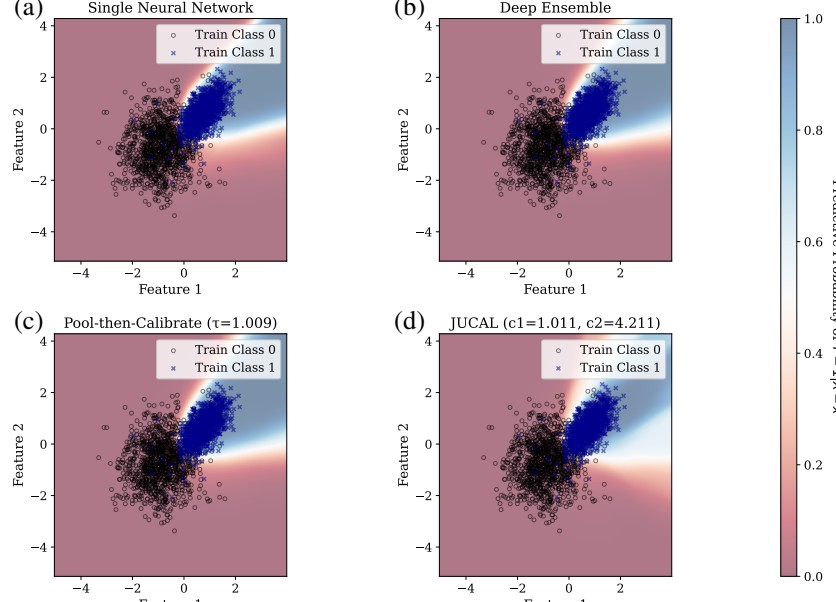

Figure 1: **Predictive probability estimation** for a synthetic 2D binary classification task. (a) Softmax outputs from a single NN. (b) Deep Ensemble. (c) & (d) show the same ensemble as in (b) but with different calibration algorithms applied to it. In all cases, the uncertainty peaks near the decision boundary, but only JUCAL sufficiently accounts for epistemic uncertainty by widening the uncertain region (bright colors) as the distance to the training data increases. This reflects the model's limited knowledge in data-sparse regions, highlighting the ensemble's ability to distinguish between aleatory and epistemic components.

while increasing it leads to underconfidence in regions dominated by aleatoric uncertainty. Classical temperature scaling cannot resolve this imbalance between the two types of uncertainty.

*To address this shortcoming*, we propose *JUCAL*, a novel method specifically for classification that jointly calibrates both *aleatoric* and *epistemic* uncertainty. Unlike standard post-hoc calibration approaches, our method explicitly balances these two uncertainty types during calibration, resulting in well-calibrated point-wise predictions (visualized in Figure 1(b)) and informative decomposed uncertainty estimates. Our algorithm can be easily applied to any already trained ensemble of models that output "probabilities". Our experiments across multiple text-classification datasets demonstrate that our approach consistently outperforms existing benchmarks in terms of NLL (up to 15%), predictive set size (up to 20% given the same coverage), and $AOROC = (1 - AUROC)$ (up to 40%). Our method reduces the inference cost of the best-performing ensemble proposed in Arango et al. (2024) by a factor of about 10, while simultaneously improving the uncertainty metrics.

## 2 RELATED WORK

Bayesian methods, such as Bayesian NNs (BNNs) (Neal, 1996; Gal et al., 2016), estimate both epistemic and aleatoric uncertainty by placing a prior over the NN's weights. If the true prior were known, the posterior predictive distribution would theoretically be well calibrated in a Bayesian sense. However, in practice, the prior is often unknown or misspecified, and thus BNNs are not guaranteed to produce calibrated predictions. We note that our algorithm can easily be extended to BNNs.

As an alternative, DEs, introduced by Lakshminarayanan et al. (2017), have demonstrated competitive or superior performance compared to BNNs across several metrics (Abe et al., 2022; Gustafsson et al., 2020; Ovadia et al., 2019). DEs, from a Bayesian perspective, approximate the posterior predictive distribution by averaging predictions (i.e. softmax outputs) from multiple models trained from independent random initializations. However, like BNNs, DEs are not inherently well-calibrated and often require additional calibration to ensure reliable uncertainty estimates (Ashukha et al., 2020).

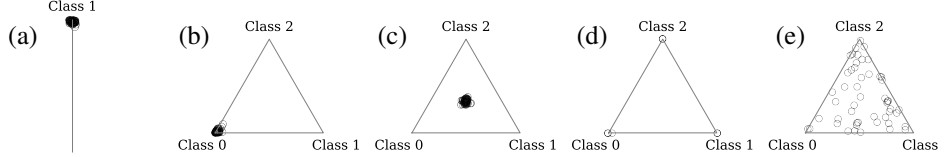

Figure 2: **Scatter plots of ensemble members' softmax outputs** for (a) binary ($K = 2$) and (b-e) ternary ($K = 3$) classification. Each subplot shows a different possibility of how the $M = 50$ predictions could be arranged for a fixed input point $x$. Each point represents a probability vector $p(y|x, \theta_m)$ over $K$ classes estimated by an ensemble member. (a)&(b) low total predictive uncertainty; (c) very high aleatoric and low epistemic uncertainty; (d) low aleatoric and very high epistemic uncertainty; (d)&(e) high epistemic uncertainty. *Theoretically* (d) claims that the aleatoric uncertainty is certainly low, while (e) is uncertain about the aleatoric uncertainty, but in practice, both (d)&(e) should usually be simply interpreted as high epistemic uncertainty (see Remark A.1).

Guo et al. (2017) suggest temperature scaling as a simple, yet effective, calibration method for modern NNs. Rahaman et al. (2021) criticize the calibration properties of ensembles and recommend *pool-then-calibrate*, aggregating ensemble member predictions before applying temperature scaling to the combined log-probabilities, using a proper scoring rule such as NLL. Although this approach can improve the calibration of DEs (Rahaman et al., 2021), it relies on a single calibration parameter to scale the total uncertainty, without using separate parameters to explicitly account for aleatoric and epistemic uncertainty. Thus, *pool-then-calibrate* implicitly assumes that aleatoric and epistemic uncertainty are both equally miscalibrated. In contrast, our algorithm calibrates both epistemic and aleatoric uncertainty with *individual* scaling factors, allowing us to increase one of them while simultaneously reducing the other one.

Recently, Azizi et al. (2025) have demonstrated that the conceptual idea of using two calibration constants to balance epistemic and aleatoric uncertainty can also be successfully applied to regression while facing different technical challenges. See Appendix C for further related work.

## 3 PROBLEM SETUP

Consider the setting of supervised learning, where we are given a training dataset $\mathcal{D}_{\text{train}} = \{(\boldsymbol{x}_1, y_1), \ldots, (\boldsymbol{x}_N, y_N)\} \subset \mathcal{X} \times \mathcal{Y}$, where the pairs $(\boldsymbol{x}_i, y_i)$ are assumed to be *independent and identically distributed (i.i.d.)* and $\mathcal{Y} = \{1, \ldots, K\}$ consists of $K$ classes. Similar to the setup described in Lakshminarayanan et al. (2017), let $\{f_m\}_{m=1}^M$ be an ensemble of $M$ independently[3] trained NN classifiers and let $\{\theta_m\}_{m=1}^M$ denote the parameters of the ensemble. For each $\boldsymbol{x} \in \mathcal{X}$, each ensemble member $f_m$, followed by a softmax activation, produces a probability-vector

$$\text{Softmax}\,(f_m(x)) = p(y \mid \boldsymbol{x}, \theta_m) = (p(y = 0 \mid \boldsymbol{x}, \theta_m), \ldots, p(y = K - 1 \mid \boldsymbol{x}, \theta_m))$$

in the simplex $\triangle_{K-1}$, as visualized in Figure 2 (this can be seen as an approximation of a Bayesian posterior as described in Appendix A.2.2). A classical DE would now simply average these probability vectors to obtain a predictive distribution over the $K$ classes for a given input datapoint $\boldsymbol{x}_{N+1}$:

$$\overline{p}\left(y \mid \boldsymbol{x}_{N+1}, \{\theta_m\}_{m=1}^M\right) \;=\; \frac{1}{M} \sum_{m=1}^{M} p(y \mid \boldsymbol{x}_{N+1}, \theta_m) \in \triangle_{K-1}. \tag{1}$$

### 3.1 ALEATORIC AND EPISTEMIC UNCERTAINTY

There are fundamentally different reasons to be uncertain. Case 1: If each ensemble of the $M$ ensemble members outputs a probability vector in the center of the simplex without favoring any

---

[3]The neural networks are not statistically independent if the dataset $\mathcal{D}$ is treated as a random variable, since all models are trained on the same $\mathcal{D}$. However, they can be considered conditionally independent given $\mathcal{D}$, due to independent random initialization and data shuffling at the beginning of each training epoch.

class, you should be uncertain (aleatoric uncertainty; similar to Figure 2(c)).[4] Case 2: If each ensemble member outputs a probability vector in a corner of the simplex, where each corner is chosen by $\frac{M}{K}$ ensemble members, you should be uncertain too (epistemic uncertainty; similar to Figure 2(d)).[5] Both cases result in a predictive distribution $\bar{p}$ that is uniform over the $K$ classes. However, in practice, this can lead to very different decisions. The diversity of the ensemble members describes the epistemic uncertainty, while each individual ensemble member estimates the aleatoric uncertainty. There are multiple different approaches to quantify them mathematically (see Appendix A.2). In our method, we calibrate these two uncertainty components separately.

## 4 JOINTLY CALIBRATING ALEATORIC AND EPISTEMIC UNCERTAINTY

### 4.1 TEMPERATURE SCALING

For any probability vector $p \in \triangle_{K-1}$, one can transform $p$ by temperature scaling

$$p^{\text{TS}(T)} := \text{Softmax}\left(\text{Softmax}^{-1}(p)/T\right), \quad \text{with logits } f^{\text{TS}(T)} := \text{Softmax}^{-1}(p)/T,$$

which moves $p$ towards the center of the simplex for temperatures $T > 1$ and away from the center towards the corners for $T < 1$, where $\text{Softmax}(z) = \frac{1}{\sum_{j=1}^{K} \exp(z_j)}\left(\exp(z_1), \ldots, \exp(z_K)\right)$.

*Pool-then-calibrate* applies temperature scaling to the predictive probabilities $\bar{p}$ from Equation (1). This allows to increase the total predictive uncertainty with $T > 1$ or reducing it with $T < 1$.

*Calibrate-then-pool* applies temperature scaling on each individual ensemble-member $p(y \mid \boldsymbol{x}, \theta_m)$ before averaging them. Thus, *Calibrate-then-pool* mainly adjusts the aleatoric uncertainty.

### 4.2 JUCAL

*JUCAL* uses two calibration constants $c_1$ and $c_2$. JUCAL applies temperature scaling on each individual ensemble-member $p(y \mid \boldsymbol{x}, \theta_m) = \text{Softmax}(f_m(x))$ with temperature $T = c_1$, resulting in temperature-scaled logits $f_m^{\text{TS}(c_1)} = \frac{f_m}{c_1} \in \mathbb{R}^K$, as in *Calibrate-then-pool*. This allows us to increase the estimated aleatoric uncertainty by setting $c_1 > 1$ and to reduce it by setting $c_1 < 1$. However, $c_1$ is not sufficient to calibrate the epistemic uncertainty.

Therefore, we introduce a second calibration mechanism for calibrating the epistemic uncertainty via $c_2$. Concretely, $c_2$ adjusts the ensemble-diversity of the already temeperature-scaled logits $f_m^{\text{TS}(c_1)}(x)$ without changing their mean $\bar{f}^{\text{TS}(c_1)}(x) := \frac{1}{M}\sum_{m=1}^{M} f_m^{\text{TS}(c_1)}(x)$. I.e., the diversity-adjusted ensemble-logits $f_m^{\text{JUCAL}(c_1,c_2)}(x) := (1-c_2)\bar{f}^{\text{TS}(c_1)}(x) + c_2 f_m^{\text{TS}(c_1)}(x)$ increase their distance to their mean $\bar{f}^{\text{TS}(c_1)}(x)$ for $c_2 > 1$ and decrease it for $c_2 < 1$. By applying Softmax we obtain an ensemble of $M$ probability-vectors $p_m^{\text{JUCAL}(c_1,c_2)}(x) = \text{Softmax}\left(f_m^{\text{JUCAL}(c_1,c_2)}(x)\right) \in \triangle_{K-1}$.

By combining these steps and averaging, JUCAL obtains the calibrated predictive distribution

$$\bar{p}^{\text{JUCAL}(c_1,c_2)}(x) := \frac{1}{M}\sum_{m=1}^{M} \text{Softmax}\left(\frac{(1-c_2)}{c_1}\bar{f}(x) + \frac{c_2}{c_1}f_m(x)\right) \tag{2}$$

from the uncalibrated logits $f_m(x)$ of the $M$ ensemble members and their mean $\bar{f} := \sum_{m=1}^{M} f_m(x)$. In practice, we usually don't know a priori how to set $c_1$ and $c_2$. Hence, JUCAL picks

$$(c_1^*, c_2^*) \in \underset{(c_1,c_2)\in(0,\infty)\times[0,\infty)}{\arg\min} \text{NLL}(\bar{p}^{\text{JUCAL}(c_1,c_2)}, \mathcal{D}_{\text{cal}}) \tag{3}$$

that minimize the $\text{NLL}(p, \mathcal{D}_{\text{cal}}) := \frac{-1}{|\mathcal{D}_{\text{cal}}|}\sum_{(x,y)\in\mathcal{D}_{\text{cal}}} \log p(y \mid x)$ on a calibration dataset $\mathcal{D}_{\text{cal}}$. The NLL is a proper scoring rule, and rewards low uncertainty for correct predictions and strongly penalizes low uncertainty for wrong predictions. In our experiments, we are reusing the validation dataset $\mathcal{D}_{\text{val}}$ as a calibration dataset while evaluating our results on a separate test set $\mathcal{D}_{\text{test}}$. For a pseudo-code implementation of JUCAL, see Algorithm 1.

---

[4]This is analogous to multiple doctors telling you that they are too uncertain to make a diagnosis.
[5]This is analogous to multiple doctors telling you highly contradictory diagnoses.

---

**Algorithm 1: JUCAL** (simplified). See Algorithm 2 for a faster implementation.

---

**Input** : Ensemble $\mathcal{E} = (f_1, \ldots, f_M)$, calibration set $\mathcal{D}_{\text{cal}}$ (e.g., $\mathcal{D}_{\text{cal}} = \mathcal{D}_{\text{val}}$), grid $C$ for candidate values $(c_1, c_2)$

1  Initialize best NLL $\leftarrow \infty$ and $(c_1^*, c_2^*)$ arbitrarily

2  **foreach** $(c_1, c_2) \in C$ **do**

3     current_NLL $\leftarrow 0$

4     **foreach** $(x, y) \in \mathcal{D}_{\text{cal}}$ **do**

5        **foreach** $m = 1, \ldots, M$ **do**

6           $f_m^{\text{TS}}(x) \leftarrow f_m(x) / c_1$                                           ▷ Temperature scaling

7        **foreach** $m = 1, \ldots, M$ **do**

8           $f_m^{\text{JUCAL}}(x) \leftarrow (1 - c_2) \cdot \frac{1}{M} \sum_{m'=1}^{M} f_{m'}^{\text{TS}}(x) + c_2 \cdot f_m^{\text{TS}}(x)$      ▷ Diversity adjustment

9        $\bar{p}^{\text{JUCAL}}(x) \leftarrow \frac{1}{M} \sum_{m=1}^{M} \text{Softmax}(f_m^{\text{JUCAL}}(x))$

10       current_NLL $\leftarrow$ current_NLL + NLL$(\bar{p}^{\text{JUCAL}}(x), y)$

11    **if** *current_NLL < best_NLL* **then**

12       best_NLL $\leftarrow$ current_NLL

13       $(c_1^*, c_2^*) \leftarrow (c_1, c_2)$

**return** : $(c_1^*, c_2^*)$

---

### 4.3 FURTHER INTUITION ON JUCAL

In Figure 3, we show a simple toy example where all the ensemble members manage to quite precisely learn the true conditioned class-probability within the body of the distribution, but not in data-scarce regions. Also, the disagreement of the ensemble logits increases in data-scarce regions, indicating a higher epistemic uncertainty in these regions. However, the amount by which disagreement increases in these regions is too low in this example, while at the same time, the aleatoric uncertainty is slightly overestimated (e.g., at $x = \frac{-\pi}{2}$). This leads to overconfidence OOD (i.e., outside $[-7, 7]$) and slight underconfidence in the body of the distribution. Pool-then-calibrate can only globally increase or decrease the uncertainty, which cannot resolve this problem. In contrast, JUCAL can simply increase the ensemble diversity via $c_2 \gg 1$ and simultaneously decrease the aleatoric uncertainty via $c_1 < 1$, resulting in reasonable input-conditional predictive uncertainty across the entire range of $x \in [-10, 10]$. In the low epistemic-uncertainty regions, the logits of different ensemble members almost perfectly agree; therefore, linearly scaling up their disagreement by $c_2$ does only have a small effect. Conversely, in regions where disagreement of pre-calibrated logits is already elevated, scaling this further up by $c_2$ has a large effect. This way, $c_2$ can more selectively calibrate the epistemic uncertainty without manipulating the aleatoric uncertainty too much.

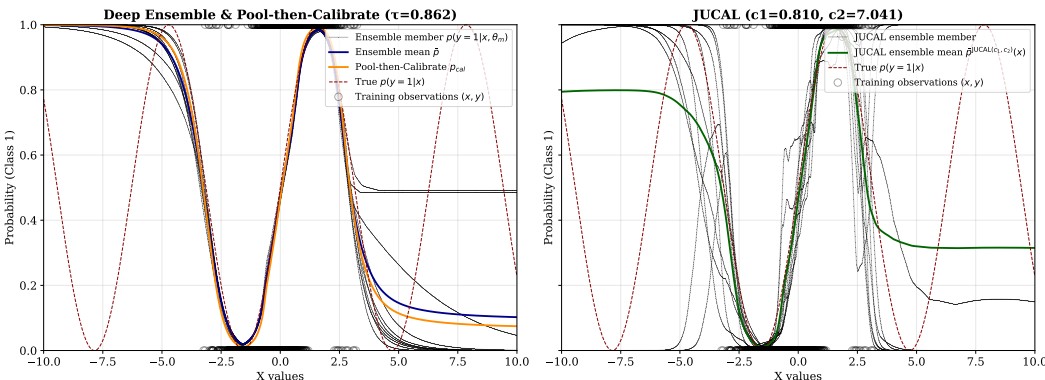

Figure 3: Binary classification example with $X \sim \mathcal{N}(0, 1)$. The ensemble logits strongly agree in the center of the distribution $x \in [-2, 2]$, but disagree more as one moves away from the center. The two subplots show the same ensemble before and after applying JUCAL to it.

## 5 RESULTS

In this section, we empirically evaluate JUCAL based on a comprehensive set of experiments. In Section 5.1 we describe the experimental setup and in Section 5.2 the experimental results.

### 5.1 EXPERIMENTAL SETUP

Arango et al. (2024) introduce a comprehensive metadataset containing prediction probabilities from a large number of fine-tuned *large language models (LLMs)* on six text classification tasks. For each task, predictions are provided on both validation and test splits. The underlying models include GPT2, BERT-Large, BART-Large, ALBERT-Large, and T5-Large, spanning a broad range of architectures and parameter counts, from 17M to 770M parameters. This metadataset is particularly valuable as it allows us to use already finetuned models for our experiments. Arango et al. (2024) used 3800 GPU hours to fine-tune these models, allowing us to isolate and study the effects of aggregation and calibration strategies independently of model training. In comparison, applying JUCAL to these expensively fine-tuned models only requires a few CPU-minutes. Six full-sized datasets and six reduced mini-datasets were used. Additional details about the metadataset are provided in Table 10.

#### 5.1.1 EVALUATION METRICS AND BENCHMARKS

Model performance is evaluated using the average NLL, which is commonly used in related work and also reported by Arango et al. (2024) for their ensemble methods. It is computed as $\mathrm{NLL}(p, \mathcal{D}_{\mathrm{test}}) := -\sum_{(x,y) \in \mathcal{D}_{\mathrm{test}}} \log p(y \mid x)$. In addition, we report $\mathrm{AORAC} = 1 - \mathrm{AURAC}$, representing the area over the rejection-accuracy curve. Each point on this curve gives the accuracy on a subset of the dataset, where the model is most certain, i.e., the model is allowed to reject answering questions for which it estimates high uncertainty. The AORAC is equal to the average misclassification rate, averaged over all different rejection rates. As a third metric, we report $\mathrm{AOROC} = 1 - \mathrm{AUROC}$, representing the area over the *receiver-operator-curve (ROC)*. Here, the AUROC is computed by averaging the one-vs-rest AUROC scores obtained for each class. Both AORAC and AOROC measure how well the model is able to rank the uncertainty of different input datapoints. As a fourth metric, we evaluate the average size of the prediction set required to cover the true label with high confidence (coverage threshold). For most datasets we use a 99% coverage threshold, but for DBpedia we increase this to 99.9% due to the high accuracy of the model predictions.

Among the ensemble methods presented by Arango et al. (2024), *Greedy-50*, a greedy algorithm that iteratively adds the model providing the largest performance gain (in terms of $\mathrm{NLL}(p, \mathcal{D}_{\mathrm{val}})$) until an ensemble of size 50 is formed, achieves the best overall performance. The authors demonstrate that *Greedy-50* outperforms several ensemble construction strategies, including: *Single Best*, which selects the single model with the best validation performance; *Random-M*, which builds an ensemble by randomly sampling $M$ models; *Top-M*, which selects the $M$ models with the highest validation scores; *Model Average (MA)*, which averages predictions from all models using uniform weights without model selection. They evaluated $M = 5$ and $M = 50$, and *Greedy-50* had the best performance in terms of NLL across all 12 datasets.

Given its strong empirical performance, we adopt *Greedy-50* as our benchmark. Additionally, we adopt *Greedy-5* as another benchmark, due to its up to 10 times lower computational prediction costs, which can be crucial in certain applications. For both of these ensembles, we compare three different calibration strategies: JUCAL (Algorithm 1), *pool-then-calibrate*, and no calibration.

### 5.2 EXPERIMENTAL RESULTS

Figures 4–5 present the performance of different calibration techniques on the Greedy-50 and Greedy-5 ensembles across six metrics. For detailed tables and further ablation studies, see Appendix F.

Arango et al. (2024) demonstrated the strength of the *Greedy-50* ensemble, which we further improve with our calibration method at a negligible computational cost (see Appendix H). The state-of-the-art *pool-then-calibrate* method improves NLL (Figures 4(a)&5(a)) but only rarely the other metrics. Our proposed method, JUCAL, simultaneously improves all four metrics compared to both the uncalibrated and state-of-the-art calibrated ensembles across most datasets. We observe similar performance gains for JUCAL on the smaller *Greedy-5* ensembles.

In line with Arango et al. (2024), the uncalibrated *Greedy-50* ensemble consistently outperforms *Greedy-5* in terms of test-NLL, but at an approximately 10x higher computational inference cost. However, applying JUCAL to *Greedy-5* requires only a negligible one-time computational investment and maintains its low inference costs, while achieving superior performance to both the uncalibrated *Greedy-50* and the *pool-then-calibrate Greedy-50* across most datasets and metrics. This demonstrates JUCAL's ability to significantly reduce inference costs without sacrificing predictive quality. We recommend *JUCAL Greedy-5* for cost-sensitive applications and *JUCAL Greedy-50* for scenarios where overall performance is the top priority.

Figure 4(a) shows the NLL on a held-out test set $\mathcal{D}_{\text{test}}$, our primary metric due to its property as a strictly proper scoring rule. JUCAL consistently improves the NLL of the Greedy-50 ensemble, outperforming all other non-JUCAL ensembles across all 12 datasets, with most improvements being statistically significant (see Tables 2 and 6 in Appendix F). Even more notably, for the smaller Greedy-5 ensembles, JUCAL achieves the best average test-NLL among all size-5 ensembles, with NLL reductions up to 30%. For example, on DBpedia, JUCAL Greedy-5 trained on just 10% of the data achieves a lower average NLL than all non-JUCAL ensembles, including the 10x larger ensembles trained on the full dataset. This demonstrates that JUCAL offers a more effective and computationally efficient path to improving performance than simply scaling up the training data or ensemble size.

Figure 4(b)&(c) show the AORAC $= 1 -$ AURAC and AOROC $= 1 -$ AUROC, respectively, as defined in Section 5.1.1. The pool-then-calibrate method shows no effect for these metrics. This is expected because these metrics measure the **relative uncertainty** which is invariant to monotonic transformations. They assess whether positive examples have higher certainty than negative ones, irrespective of absolute uncertainty level. In contrast, JUCAL and calibrate-then-pool consistently improve AOROC across all datasets, with statistically significant gains in most cases. This shows that JUCAL actively improves the relative uncertainty ranking of the model.

Figure 4(d), presents the predictive set size results. JUCAL and calibrate-then-pool achieve significantly smaller predictive sets. Already, a reduction in set size from 1.2 to 1.1 can equate to halving the costs of human interventions, if a set size of one corresponds to zero human intervention.

## 5.3 JUCAL's Disentanglement Into Aleatoric and Epistemic Uncertainty.

Figure 6 demonstrates that the epistemic uncertainty estimated by *JUCAL Greedy-50* substantially decreases as more training observations are collected for each of the 6 datasets, and for 5 out of 6 datasets in the case of *JUCAL Greedy-5*. Conversely, the estimated aleatoric uncertainty usually does not show any systematic tendency to decrease as more training observations are collected. These results align well with the theoretical understanding that epistemic uncertainty is reducible by collecting more training observations and aleatoric uncertainty is not. We used Equations (6) and (7) from Appendix A.2.1 to compute the values presented in Figure 6, while there would be other alternatives too. More research is needed to interpret different scales of estimated epistemic and aleatoric uncertainty across different datasets and different ensembles to better estimate the potential benefits of collecting more data to guide efficient data collection. For more details, see Appendix A.3.

## 6 CONCLUSION

We have presented a simple yet effective method that jointly calibrates both aleatoric and epistemic uncertainty in DEs. Unlike standard post-hoc approaches such as temperature scaling, our method addresses both absolute and relative uncertainty through structured fitting of prediction distributions. Experiments on several datasets show that our method consistently and often significantly improves upon state-of-the-art baselines, including *Greedy-50* and *Pool-then-Calibrate Greedy-50*, and is almost never significantly outperformed by any of the baselines on any evaluated metric. Our method is remarkably stable and reliable, making it a safe and practical addition to any classification task. It can also be used to reduce inference costs without sacrificing predictive performance or uncertainty quality by compensating for the weakness of *Greedy-5*. **Limitations and future work:** So far, our empirical evaluation focused on text classification with fine-tuned LLMs and image classification with CNNs, using rather large calibration datasets. Future work includes evaluating JUCAL on other data modalities and models and extending it to Chatbots.

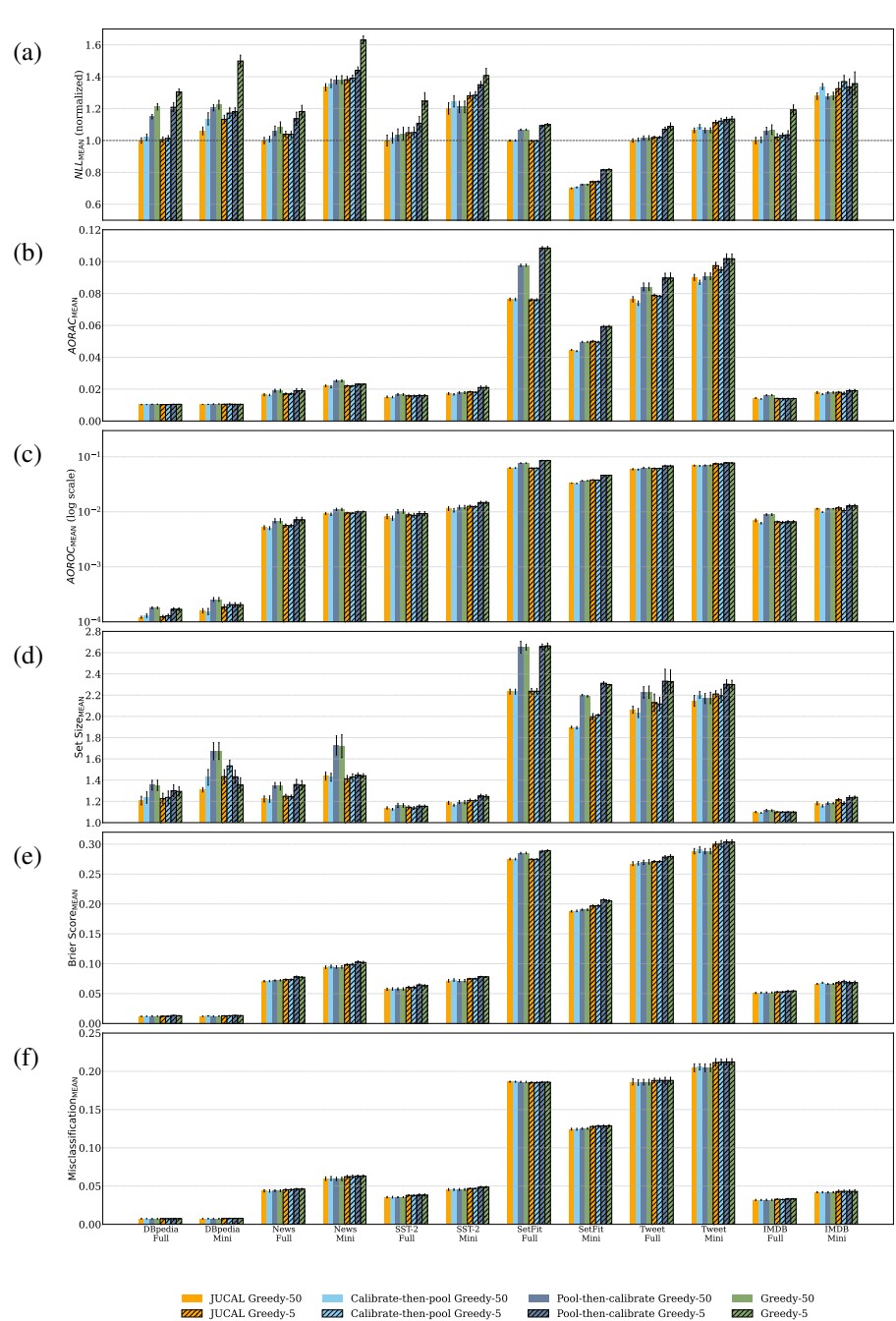

Figure 4: **Text Classification Results.** For each of the six subplots, lower values of the metrics (displayed on the y-axis) are better. On the x-axis, we list 12 text classification datasets (a 10%-mini and a 100%-full version of 6 distinct datasets). The striped bars correspond to ensemble size $M = 5$, while the non-striped bars correspond to $M = 50$. JUCAL's results are yellow. For all six metrics (defined in Section 5.1.1), we show the average and $\pm 1$ standard deviation across 5 random validation-test splits. **(a) NLL** normalized by the mean of JUCAL Greedy-50 on the corresponding full dataset; **(b) AORAC** $= 1 - $ **AURAC**; **(c) AOROC** $= 1 - $ **AUROC**; **(d) Average set size** for the coverage threshold of 99.9% for *DBpedia* (Full and Mini) and 99% for all other datasets; **(e) Brier Score**; **(f) Misclassification Rate** $= 1 - $ **Accuracy**. For more detailed results, see the corresponding tables in Appendix F.

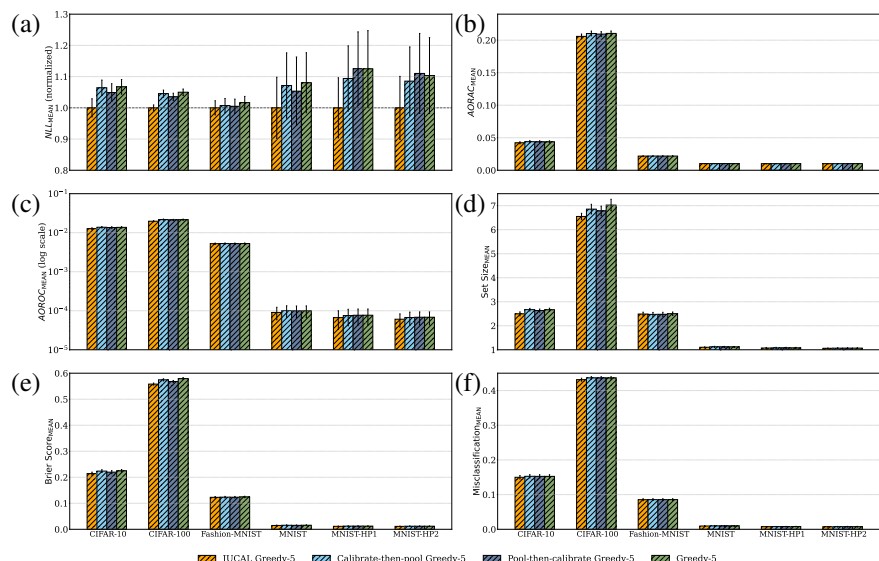

Figure 5: **Image Classification Results.** For each of the six subplots, lower values of the metrics (displayed on the y-axis) are better. On the x-axis, we list distinct image classification datasets (and two hyperparameter-ablation studies for MNIST). JUCAL's results are yellow. For all six metrics (defined in Section 5.1.1), we show the average and $\pm 1$ standard deviation across 10 random train-validation-test splits. **(a) NLL** normalized by the mean of JUCAL Greedy-5; **(b) AORAC** = $1 - $ **AURAC**; **(c) AOROC** = $1 - $ **AUROC**; **(d) Average set size** for the coverage threshold of 99% for *CIFAR-10*, 90% for *CIFAR-100*, and 99.9% for al variants of *MNIST* and *Fashion-MNIST*; **(e) Brier Score**; **(f) Misclassification Rate** = $1 - $ **Accuracy**.

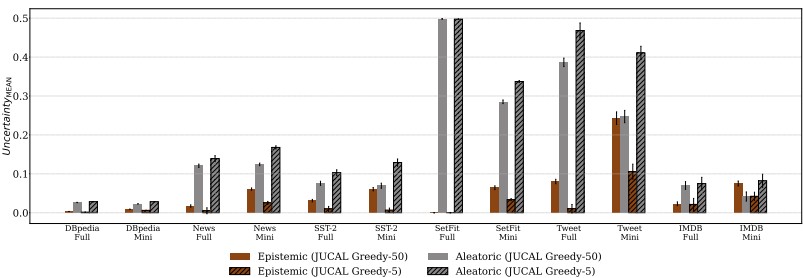

Figure 6: **Epistemic and Aleatoric Uncertainty** (computed as in Appendix A.2.1) of JUCAL applied on Greedy-50 ensembles across six datasets in the metadataset. We compare the full (100%) and the mini (10%) metadataset configurations for both epistemic and aleatoric uncertainty. Bars indicate the mean uncertainty, and error bars denote one standard deviation over random seeds.

## REPRODUCIBILITY STATEMENT

Our source code for all experiments is available at `https://github.com/anoniclr2/iclr26_anon`. Upon final publication, we will provide a permanent public repository with an installable package.

## THE USE OF LARGE LANGUAGE MODELS (LLMS)

We used Large Language Models (LLMs), specifically ChatGPT and Gemini, to assist with improving the English writing on a sentence or paragraph level. The content and scientific ideas presented in the paper are entirely our own. Every suggestion provided by the LLM was carefully reviewed, iterated upon, and corrected by a human. We confirm that every sentence in the paper and the appendix has been checked and verified by a human author.

In writing the code, we used standard LLM-based coding tools, specifically ChatGPT and Claude Code, to increase efficiency. These LLMs were used mainly for generating figures rather than for developing core modules of the source code. All changes made with the help of an LLM were carefully reviewed before being committed to the GitHub repository.

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

LIST OF APPENDICES

## A  ALEATORIC VS. EPISTEMIC UNCERTAINTY

There are many different point of views on *aleatoric* and *epistemic Uncertainty* (Kirchhof et al., 2025). While Kirchhof et al. (2025) emphasizes the differences between these points of views, we want to highlight their connection, while also mentioning some subtle differences.

### A.1  A CONCEPTUAL POINT OF VIEW ON ALEATORIC AND EPISTEMIC UNCERTAINTY

In this subsection, we try to provide a high-level discussion of the underlying philosophical ideas of epistemic and aleatoric uncertainty, which might be slightly vague from a mathematical point of view.

Aleatoric uncertainty describes the inherent randomness in the data-generating process (such as label noise) or class overlap. This is the uncertainty some with perfect knowledge of the true distribution $p(y|x)$ would face. This uncertainty cannot be reduced by observing further i.i.d. training samples. For this reason, aleatoric uncertainty is sometimes seen as "irreducible". In practice, one can reduce aleatoric uncertainty by reformulating the problem: E.g., by measuring additional features that can be added as additional coordinates to $x$.

Epistemic uncertainty describes the lack of knowledge about the underlying data-generating process. Epistemic uncertainty captures the limits in understanding the unknown distribution of the data on a population level. If we knew exactly the distribution $p(y|x)$, then we would have no epistemic uncertainty for this $x$, even if $p(y|x)$ gives a non-zero probability mass to multiple different classes. We expect this uncertainty to shrink as we observe more training data.

These are descriptions should not be understood as precise mathematical definitions, but rather provide some basic guidance for intuition. They are vague in the sense that different mathematical formalisms have been proposed to quantify them, which do not agree on a quantitative level.Some parts of the literature even (slightly) disagree with these descriptions (Kirchhof et al., 2025).

### A.2  AN ALGORITHMIC/MATHEMATICAL POINT OF VIEW ON ALEATORIC AND EPISTEMIC UNCERTAINTY

Now the question arises, how to precisely quantify aleatoric and epistemic uncertainty and how to estimate them with tangible algorithms.

For an ensemble of classifiers, the uncertainty estimated by individual classifiers is often considered as an estimator for *aleatoric uncertainty*, while the disagreement among different classifiers is often considered as an estimator for *epistemic uncertainty*. Before we give an example for a possibility to quantify the "disagreement", we discuss the alignment and the misalignment of this algorithmic description with the conceptual description from the previous section.

If we use a too restricted class of models (e.g., using only linear models for a highly non-linear problem), then typical ensembles would estimate an increased aleatoric uncertainty, counting this approximation error as part of the aleatoric uncertainty, while according to our conceptual description from Appendix A.1, one should not count this approximation error as part of aleatoric uncertainty. While (Kirchhof et al., 2025, Section 2.2) portrays this as a dramatic inconsistency among different definitions, we want to emphasize that this inconsistency vanishes when sufficiently expressive models are chosen. E.g., the universal approximation theorem (UAT) (Cybenko, 1989; Hornik, 1991; Leshno et al., 1993) shows that sufficiently large neural networks with non-polynomial activation function can approximate any measurable function on any compact subset of $\mathbb{R}^n$.

#### A.2.1  QUANTIFYING THE MAGNITUDE OF ESTIMATED ALEATORIC AND EPISTEMIC UNCERTAINTY

Here, we will quantify the estimated magnitude of the aleatoric, the epistemic, and the total predictive uncertainty, each with a number for each input data point $x$. First we want to note, that there are many alternatives to quantifying uncertainties via numbers: One could quantify uncertaitnes via sets (e.g., confidence/credible/credal sets for frequentist/Bayesian/Levi epistemic uncertainty (Hofman et al., 2024), or predictive sets for the total predictive uncertainty, see Figure 4(d)) or via distributions (e.g., distributions over the classes for aleatoric or total predictive uncertainty, or a distribution over

such distributions for epistemic uncertainty, see Appendix A.2.2). While distributions give a more fine-grained quantification of uncertainty, numbers can be easier to visualize, for example.

**Shannon Entropy**     One way to quantify the amount of uncertainty of $p \in \triangle_{K-1}$ as a single number is the Shannon entropy

$$H(p) = -\sum_{i=1}^{K} p(y = i) \log p(y = i), \tag{4}$$

which increases with the level of uncertainty (Jaynes, 1957).[6] We can compute the Shannon entropy of the predictive distribution $\bar{p}$ to quantify the total uncertainty

$$U_{\text{total}}(\boldsymbol{x}) = H[\bar{p}] = H\left[\frac{1}{M} \sum_{m=1}^{M} p(y \mid \boldsymbol{x}_{N+1}, \theta_m)\right], \tag{5}$$

In classification, *mutual information* (MI) has become widely adopted to divide uncertainty into *aleatoric* and *epistemic* uncertainty. As proposed by Depeweg et al. (2017; 2018).

We define, analogously to the Bayesian equivalent in Appendix A.2.3, *aleatoric* uncertainty as

$$U_{\text{aleatoric}}(\boldsymbol{x}) = \frac{1}{M} \sum_{m=1}^{M} H\left[p(y \mid \boldsymbol{x}_{N+1}, \theta_m)\right], \tag{6}$$

which is highest if all ensemble members output a probability vector in the center of the simplex, as in Case 1 from Section 3.1. We can use the MI to quantify *epistemic* uncertainty

$$U_{\text{epistemic}}(\boldsymbol{x}) = U_{\text{total}}(\boldsymbol{x}) - U_{\text{aleatoric}}(\boldsymbol{x}), \tag{7}$$

which is highest in Case 2 from Section 3.1. Numerous works have employed MI for decomposing uncertainty into aleatoric and epistemic components (Hüllermeier & Waegeman, 2021; Sensoy et al., 2018; Malinin et al., 2019; Malinin & Gales, 2018; Liu et al., 2019).

In our method, JUCAL, we calibrate these two uncertainty components separately, where $c_1$ is primarily calibrating the aleatoric uncertainty and $c_2$ is primarily calibrating the epistemic uncertainty.[7]

Note that there are multiple other alternative decomposition-formulas (Kirchhof et al., 2025). While they differ on a quantitative level, most of them roughly agree on a qualitative level. On a qualitative level, Kirchhof et al. (2025); Wimmer et al. (2023) criticize that the MI is maximal if the the ensemble members' predictions are symmetrically concentrated on the $K$ corners of the simplex $\triangle_{K-1}$, while one could also argue that the epistemic uncertainty should be maximal if the ensemble members' predictions are uniformly spread over the simplex. Our opinion is that both cases should be considered as "very high epistemic uncertainty", while it is often not that important in practice to decide which of them has even higher epistemic uncertainty.

*Remark* A.1 (Uniform over the Simplex vs. Corners of the Simplex). From the conceptual description of epistemic uncertainty in Appendix A.1, we would expect the uniform distribution over the simplex $\triangle_{K-1}$ to have very high or even maximal epistemic uncertainty. From this perspective, it can be surprising that the MI (7) assigns an even larger value to Case 2 from Section 3.1. For example, Wimmer et al. (2023) argues that Case 2 should have a lower epistemic uncertainty than the uniform distribution over the simplex, since Case 2 (interpreted as a Bayesian posterior) seems to know already about the absence of aleatoric uncertainty, which is some knowledge about the data-generating process, while the uniform distribution represents the absence of any knowledge on the data-generating process. However, in practice, typically, Case 2 does not actually imply any knowledge of the absence of aleatoric uncertainty. For example, ReLU-NNs have the property that they extrapolate the logits almost linearly in a certain sense (Heiss et al., 2019; 2023; 2022; Heiss, 2024), which results in ReLU-NNs' softmax outputs typically converging to a corner of the simplex as you move further

---

[6]The entropy $H : \triangle_{K-1} \to [0, \infty)$ is a concave function. The entropy is zero at the corners of the simplex, positive everywhere else, and maximal in the center of the simplex. [link to plot]

[7]No calibration method can adjust aleatoric and epistemic uncertainty in complete isolation. The two are inherently linked: e.g., when total uncertainty is maximal (i.e., a uniform mean prediction), an increase in one type must decrease the other. Thus, while JUCAL's parameters have primary targets—$c_1$ for aleatoric and $c_2$ for epistemic—they inevitably have secondary effects on the other uncertainty component.

away from the training distribution. Therefore, it is very common that far out of distribution all ensemble members' softmax outputs lie in the corners of the simplex $\triangle_{K-1}$, which usually should *not* be interpreted as having very reliable knowledge that the true probability is not in the center of the simplex, but rather simply as being very far OOD. Overall, we think the most pragmatic approach is to consider every value of MI larger than the MI of the uniform distribution over the simplex as high epistemic uncertainty, without differentiating much among even higher values of MI. We think this pragmatic approach can be sensible in both settings (a) when using a typical DE, where Case 2 should not be overinterpreted, and (b) when having access to a reliable posterior that (for some exotic reason) is really purposefully only concentrated on the corners of the simplex.

### A.2.2 A BAYEISAN POINT OF VIEW

In a Bayesian setting, we place a prior distribution $p(\theta)$ over the model parameters[8].The posterior predictive distribution for a new input $\boldsymbol{x}_{N+1}$ and class label $k$ is given by:

$$p(y = k \mid \boldsymbol{x}_{N+1}, \mathcal{D}) \;=\; \int p(y = k \mid \boldsymbol{x}_{N+1}, \theta) \, p(\theta \mid \mathcal{D}) \, d\theta, \tag{8}$$

and can be approximated by averaging the ensemble members:

$$\overline{p}(y \mid \boldsymbol{x}_{N+1}, \mathcal{D}) \;=\; \frac{1}{M} \sum_{m=1}^{M} p(y \mid \boldsymbol{x}_{N+1}, \theta_m, \mathcal{D}), \tag{9}$$

if the ensemble members $\theta_m$ are approximately sampled from the posterior $p(\theta \mid \mathcal{D})$.

For any fixed input data point $x$, each sample from the posterior corresponds to a point on the simplex $\triangle_{K-1} := \left\{ p \in [0,1]^K : \sum_{k=0}^{K-1} p_k = 1 \right\}$. Thus, for any fixed input data point $x$, the posterior distribution corresponds to a distribution on the simplex $\triangle_{K-1}$. Such a distribution on the simplex (illustrated in Figure 7) can be referred to as a *higher-order distribution*, since each point on the simplex corresponds to a categorical distribution over the $K$ classes. Each point on the simplex $\triangle_{K-1}$ corresponds to a hypothetical aleatoric uncertainty. The posterior distribution over the simplex describes the epistemic uncertainty over these hypotheses. The posterior predictive distribution (8) contains the total predictive uncertainty over the $K$ classes, incorporating both aleatoric and epistemic uncertainty in a principled Bayesian way.

*Remark* A.2 (Ensembles as Bayesian approximation). One interpretation of DEs is that they approximate an implicit distribution over the simplex $\triangle_{K-1}$, conditioned on the input (see Figure 2). We can use the collection of member outputs to apply moment matching and fit the $\boldsymbol{\alpha}(x) \in \mathbb{R}_{>0}^K$ parameters of a Dirichlet distribution. This results in an explicit higher-order distribution over the simplex. For example, for $K > 3$ it is hard to visualize the $m$ $K$-dimensional outputs of the ensembles, whereas it is easier to visualize the $K$-dimensional $\boldsymbol{\alpha}(x)$-vector for multiple $x$-values simultaneously.

*Remark* A.3 (Applying JUCAL to Bayeisan methods). Mathematically, JUCAL could be directly applied to Bayesian methods by replacing the sums in Algorithm 1 by posterior-weighted integrals. In practice, we sample $m$ ensemble members from the Bayesian posterior and then apply Algorithm 1 to this ensemble, which corresponds to using Monte-Carlo approximations of these posterior-weighted integrals.

### A.2.3 QUANTIFYING THE MAGNITUDE OF BAYESIAN ALEATORIC AND EPISTEMIC UNCERTAINTY

As discussed in Appendix A.2.1, the Shannon entropy (4) can summarize the magnitude of uncertainty into a single numerical value. Analogously to Appendix A.2.1, we can use the Shanon entropy $H$ to quantify the magnitude of epistemic and aleatoric uncertainty in the Bayesian setting by replacing sums by expectations:

---

[8]For a Bayesian neural network (BNN) (Neal, 1996), the parameters $\theta$ correspond to a finite-dimensional vector. However, the concepts of epistemic and aleatoric uncertainty and JUCAl are much more general and can also be applied to settings where $\theta$ corresponds to an infinite-dimensional object. E.g., it is quite common in Bayesian statistics to consider a prior over functions that has full support on the space of L2-functions. For example, (deep) Gaussian processes (often with full support on L2) are popular choices. The notation $p(\theta)$ should be taken with a grain of salt, as in the infinite-dimensional case, probability densities usually don't exist, but one can still define priors as probability measures.

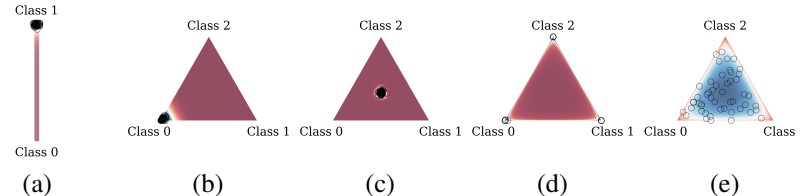

Figure 7: Different possible behaviors of a higher-order distribution over the simplex $\triangle_{K-1}$ in a binary (a) and ternary (b-e) classification task. We both show the density of a higher-order distribution (such as a posterior distribution) via colors and $M = 50$ samples from this distribution via semi-transparent black circles. Each point on the simplex $\triangle_{K-1}$ corresponds to a (first-order) distribution over the $K$ classes. Sub-figure (a)&(b) show almost **no aleatoric or epistemic uncertainty** (i.e., very low aleatoric and epistemic uncertainty, leading to a low total predictive uncertainty), (c) shows almost **only aleatoric uncertainty**, (d) shows almost **only epistemic uncertainty** and (e) shows **both aleatoric and epistemic uncertainty**. More precisely, (e) shows epistemic uncertainty on whether the aleatoric uncertainty is large or small, whereas (d) is theoretically more certain that the aleatoric uncertainty is large; (a), (b), and theoretically (d) are more certain that the aleatoric uncertainty is low. Note that (d)'s "certaitny" on the absence of aleatoric uncertainty, should not be trusted in typical settings as discussed in Remarks A.1 and A.4. (c) is certain that the aleatoric uncertainty is high.

In classification, *mutual information* (MI) has become widely adopted to divide uncertainty into *aleatoric* and *epistemic* uncertainty. As proposed by Depeweg et al. (2017; 2018), we define total uncertainty as

$$U_{\text{total}}(\boldsymbol{x}) = H\left[\mathbb{E}_m\left[p(y \mid \boldsymbol{x}, \theta_m)\right]\right], \tag{10}$$

and *aleatoric* uncertainty as

$$U_{\text{aleatoric}}(\boldsymbol{x}) = \mathbb{E}_m\left[H\left[p(y \mid \boldsymbol{x}, \theta_m)\right]\right], \tag{11}$$

we can use MI to quantify *epistemic* uncertainty

$$U_{\text{epistemic}}(\boldsymbol{x}) = U_{\text{total}}(\boldsymbol{x}) - U_{\text{aleatoric}}(\boldsymbol{x}). \tag{12}$$

Numerous works have employed mutual information for decomposing uncertainty into aleatoric and epistemic components (Hüllermeier & Waegeman, 2021; Sensoy et al., 2018; Malinin et al., 2019; Malinin & Gales, 2018; Liu et al., 2019).

*Remark* A.4 (Bayesian version of Remark A.1). Remark A.1 analogously also holds in the Bayesian setting. Note that ReLU-BNNs also have the property to put the majority of the posterior mass into the corners of the simplex $\triangle_{K-1}$ for far OOD data points. In practice, this should usually not be interpreted as actually being certain about the absence of aleatoric uncertainty.

### A.3  AN APPLIED GOAL-ORIENTED POINT OF VIEW: HOW CAN ALEATORIC AND EPISTEMIC UNCERTAINTY BE REDUCED?

In applications, one of the most important questions is how one can reduce the uncertainty. In simple words, epistemic uncertainty can be reduced by collecting more samples (which doesn't affect aleatoric uncertainty), and aleatoric uncertainty can be reduced by measuring more features per sample (which can even increase epistemic uncertainty). In the following, we will give a more detailed point of view. First, we want to note that the reducibility properties of uncertainty could even serve as a useful definition of epistemic and aleatoric uncertainty. While other definitions rely more on mental constructs (e.g., Bayesian or frequentist probabilistic constructs), this definition relies more on properties that can be empirically measured in the real world.

Epistemic uncertainty can be reduced by increasing the number of training observations and by incorporating additional prior knowledge (i.e., improving your modeling assumptions), while these actions have no effect on aleatoric uncertainty. In particular, increasing the number of training observations in a specific region of the input space $\mathcal{X}$, typically reduces mainly the epistemic uncertainty in this region. Adding more covariates (also denoted as features) decreases the aleatoric uncertainty on average if they provide additional useful information without ever harming the aleatoric uncertainty. In contrast, epistemic uncertainty typically increases when more covariates are added,

especially if the additional covariates are not very useful. Decreasing the noise has a very strong direct effect on reducing the aleatoric uncertainty. Additionally, decreasing the noise indirectly also decreases the epistemic uncertainty. However, if the epistemic uncertainty is already negligible (e.g., if you have already seen a very large number of training observations), then decreasing the scale of the noise can obviously not have any big effect on the epistemic uncertainty anymore in terms of absolute numbers (since the epistemic uncertainty can obviously not become smaller than zero). For a summary, see Table 1.

|  | **More observations** | **Better prior** | **More covariates** | **Smaller noise** |
|---|---|---|---|---|
| **Epistemic** | ☺ Decreases | ☺ Decreases | ☹ Increases (typically) / 😐/ ☺ | ☺ Decreases |
| **Aleatoric** | 😐 No effect | 😐 No effect | ☺ Decreases / 😐 | ☺☺ Decreases |

Table 1: Expected effects of different factors on epistemic and aleatoric uncertainty.

*Remark* A.5 (Table 1 should be understood on average). While adding covariates decreases aleatoric uncertainty *on average*, it can increase it for specific subgroups. Consider a 1,000 sq. ft. apartment listed for USD 10 million on an online platform. Based on these features alone, the probability of a sale is near zero (low aleatoric uncertainty). However, adding the covariate `location='Park Avenue Penthouse'` may shift the sale probability closer to 0.5, thereby *increasing* the aleatoric uncertainty for this specific data point.

**Empirical Evaluation.** The experimental results displayed in Figure 6 strongly support our hypothesis that adding more training observations clearly decreases our estimated epistemic uncertainty, in contrast to the aleatoric uncertainty. For all 6 datasets, the estimated epistemic uncertainty significantly decreases as we increase the number of training observations for JUCAL Greedy-50, and for 5 out of 6 for JUCAL Greedy-5. For DBpedia, the models already had quite small epistemic uncertainty when only trained on the reduced dataset; thus, the estimated aleatoric uncertainty was already quite accurate, and adding more training observations did not change much, except for further decreasing the already small epistemic uncertainty. For most other datasets, the epistemic uncertainty of the models trained on the reduced dataset significantly contributed to the overall uncertainty. When adding more training observations, for some of them, the *estimated* aleatoric uncertainty increased, while for others it decreased. This is expected, as in the presence of significant epistemic uncertainty, the initial estimate of aleatoric uncertainty can be very imprecise. As the true aleatoric uncertainty is not affected by adding more training observations, in contrast to epistemic uncertainty, we do not expect the aleatoric uncertainty to significantly decrease on average when adding more training observations (in our experiments, the *estimated* aleatoric uncertainty even increased on average). This empirically shows that epistemic and aleatoric uncertainty react very differently to increasing the number of training observations. Azizi et al. (2025) demonstrated in an experiment that adding more covariates can reduce the aleatoric uncertainty. We think that many more experiments should be conducted to better empirically evaluate how well different estimators of epistemic and aleatoric uncertainty agree with Table 1. More insights in this direction could help practitioners to gauge the potential effects of expensively collecting more training data or expensively measuring more covariates before investing these costs. For example, by only looking at the results for the reduced dataset (mini) in Figure 6, one could already guess that for datasets such as IMBD, Tweet, and SST-2 (for JUCAL Greedy-50), where a relatively large proportion of the estimated total uncertainty is estimated to be epistemic, there is a big potential for improving the performance by collecting more observations; while for DBpedia and SetFit, where the estimated total uncertainty is clearly dominated by estimated aleatoric uncertainty, there is little potential for benefiting from increasing the number of training observations. However, the quantification of epistemic and aleatoric uncertainty via Equations (6) and (7) from Appendix A.2.1 seem quite noisy and hard to interpret across different datasets and different ensembles, and our experiments in this direction are still way too limited. Therefore, we think further research in this direction is needed.

**This Definition Is Relative To the Definition of a "Training Observation".** This applied goal-oriented definition (i.e., epistemic uncertainty can be reduced by increasing the number of training observations, whereas aleatoric uncertainty can be reduced by increasing the number of covariates) heavily relies on the notion of a "training observation". For some ML tasks, it is quite clear what a training observation $(x, y)$ is; however, for other ML tasks, this is more ambiguous. For example, for time-series classification as in Heiss et al. (2025), you can (a) consider each partially observed

labeled path (corresponding, for example, to each patient in a hospital) as one training observation, or you can (b) consider each single measurement of any path at any time as one training observation. In case (a), each measurement in time can be seen as a covariate of a path; therefore, the proportion of the uncertainty that can be reduced by taking more frequent measurements per path should be seen as part of the aleatoric uncertainty in case (a). However, in case (b), each measurement of the path is seen as a training observation; therefore, the proportion of the uncertainty that can be reduced by taking more frequent measurements per path should be seen as part of the epistemic uncertainty in case (b). Hence, especially in the context of time series, one should first agree on a definition of what a "training observation" is before talking about epistemic and aleatoric uncertainty for less ambiguous communication. E.g., for the text-classification datasets that we study in this paper, we consider each labeled text as a training observation $(x, y)$ (and *not* every token, for example).

**Imprecise Formulations of this Definition.** We refrain from saying that epistemic uncertainty can be reduced by collecting "more data". Collecting more labeled training observations (e.g., increasing the number of rows in your tabular dataset) can reduce the epistemic uncertainty without affecting the aleatoric uncertainty, whereas collecting more covariates (e.g., increasing the number of columns in your tabular dataset) tends to increase the epistemic uncertainty and can reduce the aleatoric uncertainty instead. We also refrain from saying that aleatoric uncertainty is "irreducible", since in practice it can be reduced by measuring more covariates or by reducing the label noise.[9]

### A.4 ALEATORIC AND EPISTEMIC UNCERTAINTY FROM THE POINT OF VIEW OF THEIR PROPERTIES

Some readers might find it useful to think about how one could intuitively guess in which regions one should estimate large/low epistemic uncertainty and in which regions one should estimate large/low aleatoric uncertainty when looking at a dataset.

For regression, Heiss et al. (2021) discusses that, roughly speaking, epistemic uncertainty usually increases as you move further away from the training data. For classification, this is more complicated.[10] At least in regions with many data points, the epistemic uncertainty should be low, both for regression and classification. However, an input data-point $x$ is an unusually extreme version of a particular class can be far away from the training data, but can still be considered to quite certainly belong to this class as the following thought-experiments demonstrate.

*Example* A.6 (Electronic component). Consider a binary classification dataset where $x$ is the temperature of an electronic component and $y = 1$ denotes the failure of the component. If the training dataset only contains temperatures $x \in [10C, 120C]$ and all the electronic components with temperatures larger than $100C$ fail, then an electrical component at temperature $X = 500C$ is very far OOD; however, we can still be rather certain that it will also fail. Here in this example, we have quite a strong prior knowledge, allowing us to have very little epistemic uncertainty.

*Example* A.7 (Similar example for a more generic prior). Imagine the situation where, for a generic dataset, within the training dataset, there is a clearly visible trend that the further the input $x$ moves into the direction $v$, the more likely it is to have label $y = A$. Imagine a datapoint $x$ which is moved exceptionally far away from the center in the direction $v$. Knowing that for many real-world datasets, such trends are continued as in Example A.6, one should not have maximal epistemic uncertainty for this $x$, as one would intuitively guess that label $y = A$ is more likely than other labels without any

---

[9]In theoretical settings, the aleatoric uncertainty is often seen as "irreducible", if you consider the input space $\mathcal{X}$ and the data distribution is fixed. This makes sense from a theoretical point of view, and sometimes makes sense practically when you are for example in a kaggle-challenge setting; however, for real-world problems, it is sometimes possible to reformulate the learning problem by measuring further covariates, resulting in a different higher-dimensional input space $\mathcal{X}$, or to improve the labeling quality in the data collection process. Note that parts of the literature also denote uncertainty that can be reduced by measuring more covariates as epistemic (Kiureghian & Ditlevsen, 2009; Faber, 2005), which is not compatible which is not compatible with our definition. However, Kiureghian & Ditlevsen (2009); Faber (2005) also mention that depending on the application, it sometimes makes more sense to count this type of uncertainty as aleatoric, which then again agrees more with our definition.

[10]Also, for regression, there are some subtleties discussed in Heiss et al. (2021), making it already complicated. However, for classification, there are additional complications on top. We hypothesize that the disederata of Heiss et al. (2021) should not be applied directly to the epistemic uncertainty for classification settings; but, we also hypothesize that the disederata of Heiss et al. (2021) can be applied quite well to the logit-diversity.

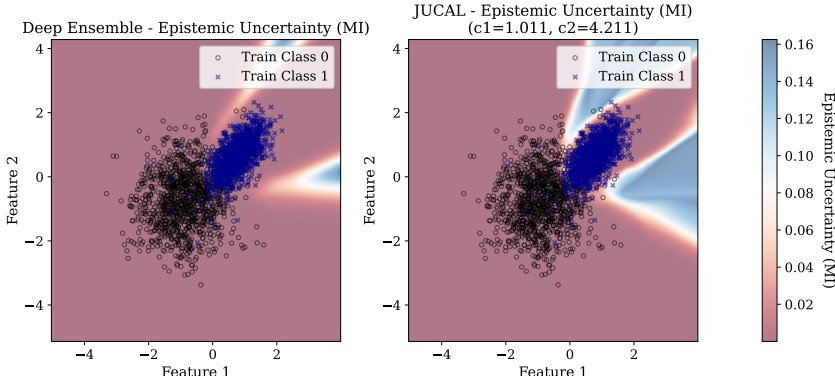

Figure 8: Estimated Epistemic Uncertainty for Figure 1

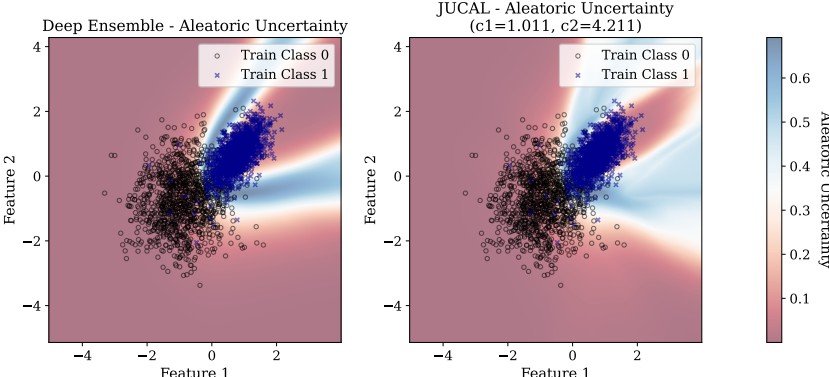

Figure 9: Estimated Aleatoric Uncertainty for Figure 1. In regions of high epistemic uncertainty, one usually does not know if the aleatoric uncertainty is high or low; thus it has the possibility to be high, and averaging over all possibilities can result in quite high estimates of the aleatoric uncertainty.

domain-specific prior knowledge. However, in such a situation, one should usually also not guess minimal epistemic uncertainty, as trends are not always continued in the real world. Intuitively, in a region with many labeled training data points, the epistemic uncertainty should be even lower. On the other hand, for an input $\tilde{x}$ that is as far away from the training data as $x$, but deviates from the training data in a direction $u$ which is orthogonal to $v$, one should intuitively typically estimate more epistemic uncertainty than for $x$. See Figure 8.

*Remark* A.8 (How do different algorithms deal with Example A.7). We expect that for an ensemble of linear logistic regression models trained on the dataset described in Example A.7, the coordinate of the logits corresponding to class A increases linearly as you move in the direction $v$, for each ensemble member. This means that if you move far enough in a direction $v$, both epistemic and aleatoric uncertainty vanish asymptotically. Pool-then-calibrate or calibrate-then-pool can slow down this decrease in uncertainty, but cannot stop this asymptotic behavior in direction $v$ (no matter which finite value you use as a calibration constant). In contrast, JUCAL can change this asymptotic behavior; it can even reverse it: If the slopes of the ensemble members' logits in direction $v$ at least slightly disagree, then this disagreement linearly increases as you move into direction $v$. Thus, for sufficiently large values of $c_2$ the epistemic uncertainty increases as you further move away in the direction $v$ instead of vanishing.[11] Analogous effects are expected for models that extrapolate local trends, such as logistic spline regression. Theoretical results in Heiss et al. (2019; 2023; 2022); Heiss (2024) suggest that ReLU neural networks also extrapolate local trends (or global trends for larger

---

[11]For example, JUCAL can choose a value $c_2$ which is neigther so large that epistemic uncertainty quickly increases in direction $v$ nor a value of $c_2$ so small that epistemic uncertainty quickly vanishing in direction $v$, but rather something in between where the epistemic uncertainty almost stays constant when extrapolating in direction $v$ (while quickly increasing when extrapolating in other orthogonal directions).

regularization), and in our experiments on synthetic datasets, we actually observe such phenomena. See Figure 10 as an example of trained NNs.

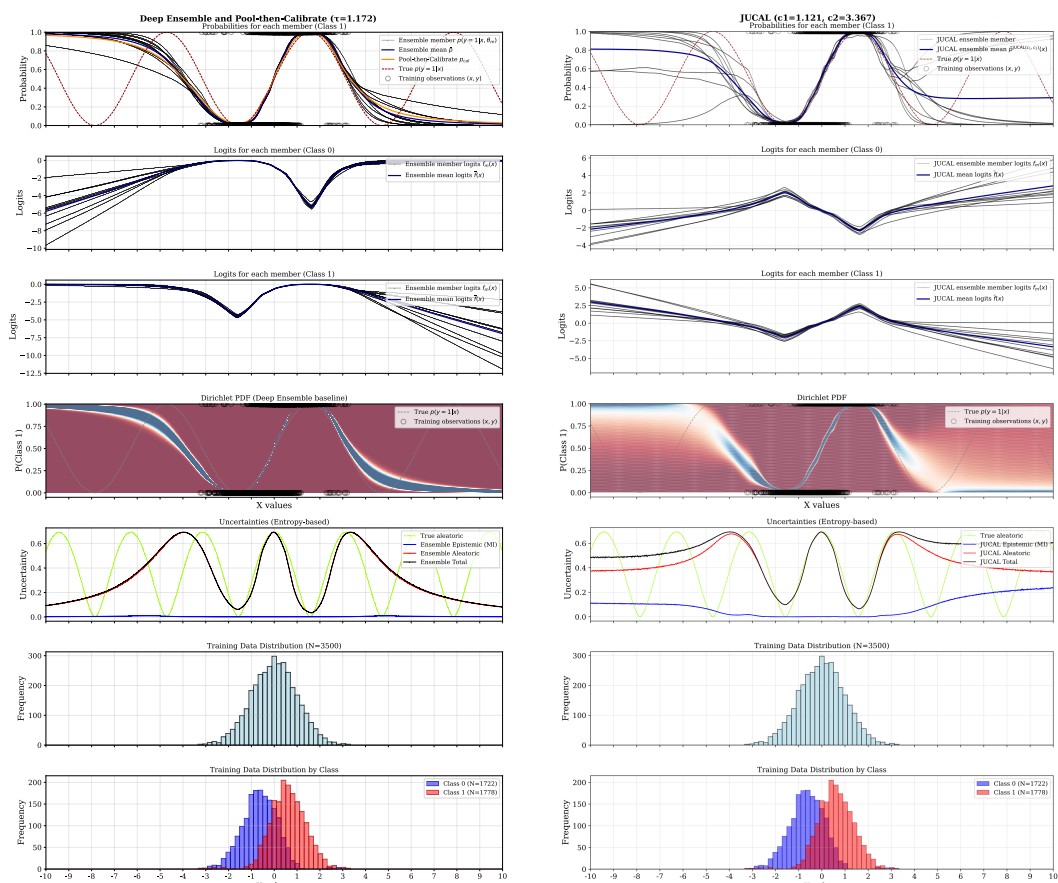

Figure 10: The same ensemble without and with JUCAL calibration. The logit diversity increases as you move further OOD, but the probability-diversity decreases without JUCAL.

If you observe different labels $y_i \neq y_j$ for identical inputs $x_i = x_i$, there has to be some aleatoric uncertainty present. In practice, you rarely observe exactly the same input $x$ more than once, but typical models also estimate large aleatoric uncertainty if the labels vary for almost identical $x$. Intuitively, this is a reasonable, if one assumes that the true conditional distribution does not fluctuate a lot between almost identical inputs $x$.

## A.5 APPLICATIONS OF EPISTEMIC AND ALEATORIC UNCERTAINTY

Aleatoric uncertainty and epistemic uncertainty can play different roles for different applications. For some applications, estimating pure epistemic uncertainty is more relevant, while for other applications, the combined total predictive uncertainty is more relevant.

**Active Learning, Experimental Design, and Efficient Data Collection.** In active learning, ranking the epistemic uncertainty of different input points $x$ can help to prioritize which of them to collect expensive labels for to reduce the overall uncertainty. After having trained a model on a labeled training dataset, comparing the epistemic and aleatoric uncertainty aggregated over some (unseen) dataset can help you to decide whether (a) collecting more labeled training samples, or (b) measuring more covariates per sample has more potential to reduce the overall uncertainty, even before investing anything into conducting (a) or (b): If the estimated epistemic uncertainty dominates the total predictive uncertainty of your current model, then (a) is more promising whereas if vice-versa

the estimated aleatoric uncertainty dominates then (b) has more potential, if there are promising candidates for further covariances.

**Prediction Tasks.** The total predictive uncertainty tries to predict the true label. Both epistemic and aleatoric uncertainty are reasons to be uncertain about your prediction.

## B    CONDITIONAL VS. MARGINAL COVERAGE

This section clarifies the distinction between conditional and marginal coverage in the context of classification. These concepts are closely related to the notions of *relative vs. absolute uncertainty* (Heiss et al., 2021), and also overlap with the terminology of *adaptive vs. calibrated*[12] *uncertainty*.

**Input-conditional coverage** (which we refer to as *conditional coverage*) requires that, for every possible input $x$, the probability that the prediction set $C(x)$ contains the true class is at least $1 - \alpha$:

$$\forall x \in \text{supp}(X) : \mathbb{P}\left[Y_{n+1} \in C(X_{n+1}) \mid X_{n+1} = x\right] \geq 1 - \alpha. \tag{13}$$

This definition is agnostic to the input distribution and instead enforces a per-instance guarantee.

In contrast, **marginal coverage** provides an average-case guarantee across the data distribution:

$$\mathbb{P}\left[Y_{n+1} \in C(X_{n+1})\right] \geq 1 - \alpha. \tag{14}$$

While marginal coverage is easier to attain and is the guarantee provided by standard conformal prediction methods, it can hide undercoverage in specific regions of the input space.

Crucially, conditional coverage implies marginal coverage under any distribution on $X$, but not vice versa. As such, achieving approximate conditional coverage is a desirable but more ambitious goal in practice.

### B.1    RELATIVE VS. ABSOLUTE UNCERTAINTY

To move toward conditional guarantees, two complementary components are needed: (i) a method that ranks uncertainty effectively (relative uncertainty), and (ii) a calibration mechanism to set the correct scale (absolute uncertainty).

**Relative uncertainty** refers to how well the model can identify which instances are more or less uncertain. For classification, this is often expressed through metrics like AOROC and AORAC, which are invariant under monotonic transformations of the confidence scores. Methods with strong relative uncertainty assign higher uncertainty to ambiguous or out-of-distribution samples and lower uncertainty where predictions are more certain and reliable.

**Absolute uncertainty**, on the other hand, involves calibrating the scale of predicted confidence. A model has a poor absolute scale of uncertainty if it is on average overconfident or underconfident averaged over the whole test dataset.

## C    FURTHER RELATED WORK

### C.1    THE PCS FRAMEWORK FOR VERIDICAL DATA SCIENCE

The *Predictability-Computability-Stability* (PCS) framework for veridical data science (Yu, 2020; Yu & Barter, 2024) provides a framework for the whole data-science-life-cycle (DSLC). They argue that uncertainty in each step of the DSLC needs to be considered. These steps include the problem formulation, data collection, exploratory analyses, data pre-processing (e.g., data transformations), data cleaning, modeling, algorithm choices, hyper- parameter tuning, interpretation, and even visualization). They suggest creating an ensemble by applying reasonable perturbations to each judgment call across all steps of the DSLC (Yu & Barter, 2024, Chapter 13). (Yu & Barter, 2024, Chapter 13)

---

[12]When we write about "calibrated uncertainty", we more precisely mean *marginally calibrated uncertainty*, $\mathbb{P}\left[Y_{n+1} \in C(X_{n+1})\right] = 1 - \alpha$, which is orthogonal to adaptivity; in contrast to *input-conditionally calibrated uncertainty*, $\forall x \in \text{supp}(X) : \mathbb{P}\left[Y_{n+1} \in C(x) \mid X_{n+1} = x\right] = 1 - \alpha$, which requires perfect adaptivity.

demonstrates PCS-based uncertainty quantification on a regression problem and poses PCS-based uncertainty quantification for classifications as an open problem.

Agarwal et al. (2025) extend the method from (Yu & Barter, 2024, Chapter 13) to classification and suggest additional improvements: The majority of the calibration literature (including Yu & Barter (2024)) removes part of the training data to leave it as calibration data, whereas Agarwal et al. (2025) give each ensemble member a bootstrap sample of the *whole* training data and only uses out-of-bag data for calibration, leading to improved data efficiency. This approach also increases the amount of data used for calibration. We believe that our method could potentially benefit even more from such an enlarged amount of calibration data, since our method calibrates 2 constants $c_1$ and $c_2$ instead of 1 constant on the calibration data. Therefore, it would be an interesting future work to combine this out-of-bag technique with JUCAL. As JUCAL can be applied to any ensemble of soft classifiers, JUCAL can also be applied to ensembles obtained via the PCS framework (the out-of-bag technique would only require a small change in the code).

We note that while (Yu & Barter, 2024, Chapter 13) and Agarwal et al. (2025) do not explicitly model aleatoric uncertainty for the case of regression, Agarwal et al. (2025) do explicitly model aleatoric uncertainty for classification by directly averaging the *soft* labels. However, they only use one calibration constant to calibrate their predictive sets, which does not allow them to compensate for a possible imbalance between aleatoric and epistemic uncertainty.[13] In contrast, our data-driven joint calibration method decides automatically in a data-driven way how to combine aleatoric and epistemic uncertainty.

Agarwal et al. (2025) conducted a large-scale empirical evaluation, showing the strong empirical performance of PCS-based uncertainty quantification on real-world datasets. For these experiments, they focused only on a smaller part of the DSLC than Yu & Barter (2024), i.e., they did not consider uncertainty from data-cleaning choices and other human judgment calls. In our experiments, we follow the setting from Arango et al. (2024), where some judgement calls (such as the choice over different pre-trained LLMs, different LoRA-ranks, and learning rate) are explicitly considered, while we also ignore other steps of the DSLC. For real-world data-science projects, we recommend combining the full PCS framework (considering all steps of the DSLC) from Yu (2020); Yu & Barter (2024) with the techniques from Agarwal et al. (2025) with JUCAL.[14]

## C.2 UNCERTAINTY CALIBRATION TECHNIQUES IN THE LITERATURE

*CLEAR* (Azizi et al., 2025) uses two constants to calibrate epistemic and aletoric uncertainty for regression tasks, while leaving classification explicitly as open future work. For regression, once you have good uncalibrated estimators for epistemic and aleatoric uncertainty, additively combining is more straightforward than for classification, i.e., they simply add the width of the scaled intervals. JUCAL's defining equation (2) is a non-trivial extension of this, as for classification, we cannot simply add predictive sets or predictive distributions. CLEAR does not give predictive distributions but predictive intervals, using the pinball loss and a constraint on the marginal coverage to calibrate the two constants. In contrast, JUCAL can output both predictive distributions and predictive sets and uses the NLL to calibrate the two constants. CLEAR significantly outperforms recent state-of-the-art models for uncertainty quantification in regression, such as CQR, PCS-UQ, and UACQR, across 17 real-world datasets, demonstrating that the conceptual idea of using two calibration constants to calibrate epistemic and aleatoric uncertainty goes beyond JUCAL's success in classification, suggesting the fundamental importance of correctly combining epistemic and aleatoric uncertainty across various learning problems. In the future, we want to extend JUCAL's concept to LLM chatbots.

---

[13]Through the lens of epistemic and aleatoric uncertainty, (Yu & Barter, 2024, Subsection 13.1.2) only focuses on aleatoric uncertainty when computing the AUROC since they only use the soft labels of a single model, whereas (Yu & Barter, 2024, Subsection 13.2.2) mainly focuses on epistemic uncertainty since they only use the *hard* (i.e., binary) labels of the ensemble members, and Agarwal et al. (2025) combines aleatoric and epistemic uncertainty in the fixed ratio 1:1 since they average the soft labels.

[14]Note that while Yu (2020); Yu & Barter (2024) were very thoroughly vetted across many real-world applications with an actual impact to practice (Wu et al., 2016; Wang et al., 2023; Basu et al., 2018; Dwivedi et al., 2020), Agarwal et al. (2025) and JUCAL are more recent works which have so far only shown their success on benchmark datasets without being vetted in the context of the full data-science-life-cycle. Therefore, the second part of the recommendation should be taken with a grain of salt.

The concept of post-hoc calibration was formalized for binary classification by Platt (1999) with the two-parameter *Platt scaling*. This idea was later adapted for the multi-class setting by Guo et al. (2017), who introduced *temperature scaling*, a simple single-parameter approach. Through a large-scale empirical study, they demonstrated that modern neural networks are often poorly calibrated and showed that this method was highly effective at correcting this. As a result, temperature scaling has become a common baseline for this task. Notably, some modern works still refer to this one-parameter method as Platt scaling, acknowledging its intellectual lineage. Beyond single-model calibration, these techniques are crucial for methods like Deep Ensembles, which improve uncertainty estimates by averaging predictions from multiple models (Lakshminarayanan et al., 2017). For ensembles, a naive approach is to calibrate each model's outputs before averaging them. However, Rahaman et al. (2021) have shown that a *pool-then-calibrate* strategy is more effective.

Ahdritz et al. (2025) suggest a higher-order calibration algorithm for decomposing uncertainty into epistemic and aleatoric uncertainty with provable guarantees. However, in contrast to our algorithm, they assume that multiple labels $y$ per training input point $x$ are available during training. For many datasets, such this is not the case. E.g., for the datasets we used in our experiment, we have only one label per input datapoint $x$.

Javanmardi et al. (2025) assumes that they have access to valid credal sets, i.e., subsets $\tilde{C}(x)$ of the simplex $\triangle_{K-1}$ that definitely contain the true probability vector $p(x)$. Under this assumption, they can trivially obtain predictive sets with a conditional coverage at least as large as the target coverage. However, in practice, without strong assumptions, it is impossible to obtain such valid credal sets $\tilde{C}(x) \subsetneq \triangle_{K-1}$. Furthermore, even if one had access to such credal sets, their predictive sets would be poorly calibrated as they are strongly biased towards over-covering, resulting in large predictive sets (which can be very far from optimal from a Bayesian perspective). They also conduct a few experiments on real-world datasets with approximate credal sets, where they achieve (slightly) higher conditional coverage than other methods, but at the cost of having larger sets than their competitor in every single experiment. They did not show a single real-world experiment where they Pareto-outperform APS in terms of coverage and set size. In contrast, JUCAL Pareto-outperforms both APS and pool-then-calibrate-APS in terms of coverage and set size in 22 out of 24 experiments. Additionally the method proposed by Javanmardi et al. (2025) is computationally much more expensive than JUCAL. In contrast to JUCAL, the method proposed by Javanmardi et al. (2025) does not adequately balance the ratio of epistemic and aleatoric uncertainty. In principle, one could apply the method by Javanmardi et al. (2025) on top of JUCAL.

Rossellini et al. (2024) introduced UACQR, a method that combines aleatoric and epistemic uncertainty in a conformal way by calibrating only the epistemic uncertainty for regression tasks, while keeping classification open for future work. They achieve good empirical results, but are outperformed by CLEAR (Azizi et al., 2025) which achieves even better results.

Cabezas et al. (2025) introduces EPISCORE, a conformal method to combine epistemic and aleatoric uncertainty using Bayesian techniques. However, they focus mainly on regression, where they achieve good results but are outperformed by CLEAR (Azizi et al., 2025). They also extent their method to classification settings and in (Cabezas et al., 2025, Appendix A.2) they also conduct one preliminary experiment for classification, where they achieve better coverage with larger set sizes, but they don't report how much larger the set size is on average.

Karimi & Samavi (2024) introduces a conformal version of Evidential Deep Learning.

### C.3 PRE-CALIBRATED UNCERTAINTY QUANTIFICATION IN THE LITERATURE

*Bayesian neural networks* (BNNs) MacKay (1992); Neal (1996) offer a principled Bayesian framework for quantifying both epistemic and aleatoric uncertainty through the placement of a prior distribution on network weights. However, the ratio of estimated epistemic and aleatoric uncertainty in BNNs is highly sensitive to the choice of prior. Consequently, we advocate applying JUCAL to an already trained BNN, calibrating both uncertainty types via scaling factors $c_1$ and $c_2$ with negligible additional computational overhead. While exact Bayesian inference in large BNNs is computationally intractable, numerous approximation techniques have been proposed, including variational inference (Graves, 2011; Blundell et al., 2015; Gal & Ghahramani, 2015; Bakhouya et al., 2024; Cong et al., 2024; Shen et al., 2024), Laplace approximations (Ritter et al., 2018; Daxberger et al., 2021), probabilistic propagation methods (Hernández-Lobato & Adams, 2015; Nguyen &

Goulet, 2022b;a), and ensembles or heuristics (Lakshminarayanan et al., 2017; Maddox et al., 2019; Heiss et al., 2021), with MCMC methods often serving as a gold standard for evaluation (Neal, 1996; Wenzel et al., 2020).[15] JUCAL can be applied to all these approximated BNNs as a simple post-processing step.

*TabPFN* (Hollmann et al., 2023) and in particular *TabPFN v2* (Hollmann et al., 2025) achieve remarkable results with their predictive uncertainty across a wide range of tabular real-world datasets (Ye et al., 2025). TabPFN (v2) is a fully Bayesian method based on a very well-engineered, highly realistic prior. A few years ago, doing Bayesian inference for such a sophisticated prior would have been considered computationally intractable. However, they managed to train a foundational model that can do such a Bayesian inference at an extremely computational cost within a single forward pass through their transformer. Their method directly outputs predictive uncertainty, which already contains both epistemic and aleatoric uncertainty. Since their prior contains a wide variety of infinitely many different realistic noise structures and function classes, we expect their method to struggle less with imbalances between epistemic and aleatoric uncertainty. Recently TabPFN-TS, a slightly modified version of TabPFN v2 was also able to outperform many state-of-the-art times models (Hoo et al., 2025). However, they come with 2 limitations compared to our method:

1. TabpPFNv2 can only deal with datasets of at most 10,000 samples and 500 features. The limited number of samples was to some extent mitigated by TabPFN v2*-DT (Ye et al., 2025). However, for high dimensional images or language datasets, such as the language datasets from our experimental setting, TabPFN is not applicable. In contrast, our method easily scales up to arbitrarily large models and is compatible with all modalities of input data, no matter if you want to classify videos, text, sound, images, graphs, or whatever.

2. TabPFN directly outputs the total predictive uncertainty without disentangling it into aleatoric and epistemic uncertainty. And we don't see any straightforward way to do so. However, in some applications it is crucial to understand which proportion of the uncertainty is epistemic and how much of it is aleatoric. Our joint calibration method explicitly entangles the predictive distribution into these 2 sources of uncertainty.

Yet another Bayesian deep learning framework is presented by (Kendall & Gal, 2017). Again they place a prior over weights and alter the output of the classification task, such that the network outputs both the mean logits and the aleatoric noise parameter $\hat{z}_t = f_\theta(\boldsymbol{x}) + \sigma_\theta(\boldsymbol{x})\,\varepsilon_t, \quad \varepsilon_t \sim \mathcal{N}(0, I)$. The posterior, being intractable, needs to be approximated. With Monte Carlo integration, the posterior predictive distribution becomes $p(y = c \mid \boldsymbol{x}_{n+1}, \mathcal{X}, \mathcal{Y}) \approx \frac{1}{T}\sum_{t=1}^{T} \mathrm{Softmax}\Big(f_{\theta_t}(\boldsymbol{x}_{n+1}) + \sigma_{\theta_t}(\boldsymbol{x}_{n+1})\,\varepsilon_t\Big)_c$, where each $\theta_t$ is a sample from $q(\theta)$. Aleatoric uncertainty is directly estimated through the fitted $\sigma_\theta$ and epistemic uncertainty through using the posterior distribution. Again this framework does not yield inherently well calibrated results.

Evidential deep learning (EDL) as presented by Sensoy et al. (2018) is a probabilistic framework for quantifying uncertainty in classification task specifically. EDL explicitly models a higher-order distribution, more specifically the Dirichlet distribution, which defines a probability density over the $K$-dimensional unit simplex Lin (2016). EDL directly fits the $\boldsymbol{\alpha}$ parameters of a Dirichlet distribution such that: $\alpha_k = f_k(\mathbf{X} \mid \boldsymbol{\theta}) + 1$, where $f_k$ denotes the output for class $k$, $\mathbf{X}$ is the input, and $\boldsymbol{\theta}$ are the model parameters. Uncertainty can then be estimated utilizing the Dirichlet distribution and its properties. Karimi & Samavi (2024) introduces a conformal version of EDL.

Malinin & Gales (2018) work on Prior Networks (PNs) entangles uncertainty estimation into *data uncertainty*, *model uncertainty*, and *distributional uncertainty*. In most methods for estimating uncertainty, the distributional uncertainty is not explicitly modeled and will also not be explicitly studied in this work. Dirichlet Prior Network (DPN) is one implementation that explicitly models the higher-order distribution as a Dirichlet distribution.

---

[15]Interestingly, there are theoretical (Heiss et al., 2022; Heiss, 2024) and empirical (Wenzel et al., 2020) studies suggesting that some of these approximations might actually provide superior estimates compared to their exact counterparts, due to poor choices of priors, such as i.i.d. Gaussian priors, in certain settings.

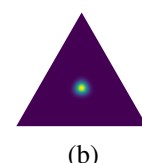 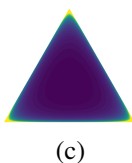 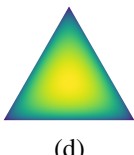

(a)          (b)          (c)          (d)

Figure 11: Desired behavior of a higher-order distribution over the simplex in a ternary classification task. Sub-figure (a) almost **no aleatoric or epistemic uncertainty**, (b) shows almost **only aleatoric uncertainty**, (c) shows almost **only epistemic uncertainty** and (d) shows **both aleatoric and epistemic uncertainty**

# D    MORE INTUITION ON JOINTLY CALIBRATING ALEATORIC AND EPISTEMIC UNCERTAINTY

To address shortcomings in DEs, we suggest a simple yet powerful calibration method that jointly calibrates *aleatoric* and *epistemic* uncertainty. We formulate desiderata for calibrated uncertainty, which motivate the design of our proposed method. Building on these principles, we develop JUCAL (Algorithm 1) as a structured calibration procedure that satisfies the desiderata by utilizing two calibration hyperparameters.

## D.1    DESIDERATA

To describe the desiderata, we consider the Dirichlet distribution as a distribution over the predicted class probabilities $\boldsymbol{p}_i = (p_{0,i}, p_{1,i}, \ldots, p_{K-1,i})$. This provides an interpretable representation, visualized on the 2-dimensional simplex in Figure 11. Calibrated classification methods should satisfy the following desiderata to yield meaningful predictions.

- For *no aleatoric and no epistemic uncertainty:* the model should produce a distribution with all its mass concentrated at one of the corners of the simplex. This corresponds to a confident and sharp prediction (visualized in Figure 11(a)).

- For *non-zero aleatoric uncertainty but zero epistemic uncertainty:* the model should produces a distribution concentrated at the center of the simplex. This corresponds to a sharp but uncertain prediction, indicating that the uncertainty is intrinsic to the data (visualized in Figure 11(b)).

- For *zero aleatoric uncertainty but non-zero epistemic uncertainty:* the model should produces a distribution with mass spread across several corners of the simplex. This reflects uncertainty due to a lack of knowledge and results in a less sharp predictive distribution (visualized in Figure 11(c)).

- For *non-zero aleatoric and non-zero epistemic uncertainty:* the model should produces a distribution that is spread broadly over the entire simplex, corresponding to high overall uncertainty and a flat predictive distribution (visualized in Figure 11(d)).

Figure 12 demonstrates how our proposed method (see Appendix D.2) satisfies these desiderata in a binary classification task.

## D.2    JUCAL

To satisfy the desiderata outlined above and to provide high-quality, point-wise predictions along with calibrated uncertainty estimates, we introduce JUCAL, summarized in Algorithm 1. Note that our actual implementation of JCUAL (Algorithm 2) is slightly more advanced than Algorithm 1. Instead of the naive grid search, we first optimize over a coarse grid and then optimize over a finer grid locally around the winner of the first grid search.

JUCAL takes as input a set of trained ensemble members $f_m \in \mathcal{E}$ and a validation set $\mathcal{D}_{\mathrm{val}}$, and returns the optimal calibration hyperparameters $(c_1^*, c_2^*)$. The implementation presented in Algorithm 1 is

based on grid search and additionally requires candidate values for the calibration hyperparameters $c_1$ and $c_2$.[16]

For inference, JUCAL computes calibrated predictive probabilities using:

$$\bar{p}(y \mid x; c_1^*, c_2^*) = \frac{1}{M} \sum_{m=1}^{M} \text{Softmax} \left( (1 - c_2^*) \cdot \frac{1}{M} \sum_{m'=1}^{M} \frac{f_{m'}(x)}{c_1^*} + c_2^* \cdot \frac{f_m(x)}{c_1^*} \right). \tag{15}$$

See Figure 13 for more intuition on how JUCAL works.

JUCAL (Algorithm 1) requires the outputs of a trained deep ensemble. If such members are not already available, a DE can be trained following the procedure described by (Lakshminarayanan et al., 2017). Optionally, ensemble member selection can be performed on the validation or calibration set, as detailed in Algorithm 3. Notably, our joint calibration method does not require access to the model parameters or training inputs, it only relies on the softmax outputs of the ensemble members and the corresponding labels on the validation and test sets.

Different values for the calibration parameters $c_1$ and $c_2$ affect the calibration in different ways. When $c_1 = 1$ and $c_2 = 1$, the distribution remains unchanged. When $c_1 < 1$, the adjusted Dirichlet distribution should concentrate more mass toward the corners of the simplex, thereby reducing *aleatoric* uncertainty. In contrast, when $c_1 > 1$, the adjusted Dirichlet distribution should shift toward the center of the simplex.

The parameter $c_2$ models the variability across the ensemble members. When $c_2 > 1$ the adjusted Dirichlet distribution should increase its variance spread mass across multiple corners of the simplex, reflecting higher *epistemic* uncertainty. In contrast, when $c_2 < 1$ the *epistemic* uncertainty decreases. There are cases where changing $c_2$ does not affect the higher-order distribution: When all ensemble members produce identical logits, the output remains a Dirac delta.

In Figure 14, we empirically compute the influence of $c_1$ and $c_2$. In Figure 14, we see in the second row of subplots that the (average) aleatoric uncertainty is monotonically increasing with $c_1$ and that large values of $c_2$ can reduce the aleatoric uncertainty. In the third row of subplots, we can see that the (average) epistemic uncertainty is monotonically increasing with $c_2$ and that large values of $c_1$ can reduce the epistemic uncertainty. In the fourth row of subplots, we can see that the (average) total uncertainty is monotonically increasing in $c_1$ and $c_2$. When jointly studying the last three rows of sublots, we can see that we can change the ratio of epistemic and aleatoric uncertainty (even without changing the total uncertainty) when increasing one of the two constants while decreasing the other one.

---

[16]While Algorithm 1 uses grid search for clarity and reproducibility, the parameters $(c_1^*, c_2^*)$ can alternatively be found via a two-stage grid search (Algorithm 2), gradient-based optimization methods, or any other optimization algorithm.

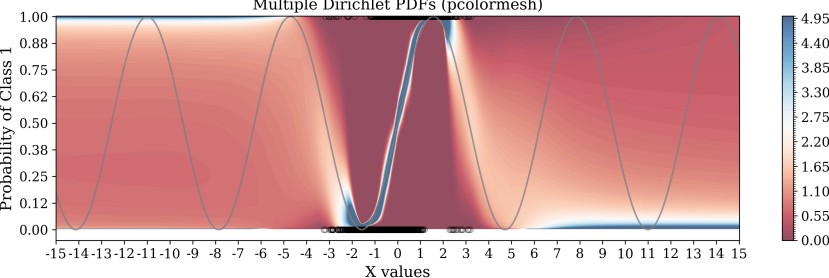

Figure 12: **Illustrating the point-wise predicted Dirichlet distributions** in a 1D binary classification task with class probabilities defined by $p(y = 1 \mid x) = 0.5 + 0.5 \sin(x)$, where $x \sim \mathcal{N}(0, 1)$ (visualized as green line) and $y \sim \text{Bernoulli}(p(y = 1 \mid x))$. For each value $x \in [-15, 15]$, we visualize the density of the corresponding Dirichlet distribution over the interval $[0, 1]$, with black circles indicating the training data.

**Algorithm 2:** JUCAL (coarse-to-fine grid search). See Algorithm 1 for a simplified version.

**Input** : Ensemble $\mathcal{E} = (f_1, \ldots, f_M)$, calibration set $\mathcal{D}_{\text{cal}}$ (e.g., $\mathcal{D}_{\text{cal}} = \mathcal{D}_{\text{val}}$), **coarse grid** $C^{\text{coarse}}$ for candidate values $(c_1, c_2)$, **fine grid size** $K$

**1** Initialize best_NLL$^{\text{coarse}} \leftarrow \infty$ and $(\hat{c}_1, \hat{c}_2)$ arbitrarily

**2 foreach** $(c_1, c_2) \in C^{\text{coarse}}$ **do**

**3**     current_NLL $\leftarrow 0$

**4**     **foreach** $(x, y) \in \mathcal{D}_{\text{cal}}$ **do**

**5**        **foreach** $m = 1, \ldots, M$ **do**

**6**           $f_m^{\text{TS}}(x) \leftarrow f_m(x)/c_1$                      ▷ Temperature scaling

**7**        **foreach** $m = 1, \ldots, M$ **do**

**8**           $f_m^{\text{JUCAL}}(x) \leftarrow (1 - c_2) \cdot \frac{1}{M} \sum_{m'=1}^{M} f_{m'}^{\text{TS}}(x) + c_2 \cdot f_m^{\text{TS}}(x)$    ▷ Diversity adjustment

**9**        $\bar{p}^{\text{JUCAL}}(x) \leftarrow \frac{1}{M} \sum_{m=1}^{M} \text{Softmax}(f_m^{\text{JUCAL}}(x))$

**10**        current_NLL $\leftarrow$ current_NLL + NLL$(\bar{p}^{\text{JUCAL}}(x), y)$

**11**     **if** *current_NLL $<$ best_NLL$^{\text{coarse}}$* **then**

**12**        best_NLL$^{\text{coarse}} \leftarrow$ current_NLL

**13**        $(\hat{c}_1, \hat{c}_2) \leftarrow (c_1, c_2)$

**14** Let $c_{1,\min}$ be the minimum $c_1$ in $C^{\text{coarse}}$ and $c_{2,\min}$ the minimum $c_2$ in $C^{\text{coarse}}$.

**15** $c_1^{\text{low}} \leftarrow \max\{\hat{c}_1 - 0.2\,\hat{c}_1,\ c_{1,\min}\}, \quad c_1^{\text{high}} \leftarrow \hat{c}_1 + 0.2\,\hat{c}_1$

**16** $c_2^{\text{low}} \leftarrow \max\{\hat{c}_2 - 0.2\,\hat{c}_2,\ c_{2,\min}\}, \quad c_2^{\text{high}} \leftarrow \hat{c}_2 + 0.2\,\hat{c}_2$

**17** Define $c_1^{\text{fine}}$ as $K$ evenly spaced values in $\left[c_1^{\text{low}},\ c_1^{\text{high}}\right]$ and $c_2^{\text{fine}}$ as $K$ evenly spaced values in $\left[c_2^{\text{low}},\ c_2^{\text{high}}\right]$.

**18** Initialize best_NLL $\leftarrow \infty$ and $(c_1^*, c_2^*)$ arbitrarily

**19 foreach** $c_1 \in c_1^{\text{fine}}$ **do**

**20**     **foreach** $c_2 \in c_2^{\text{fine}}$ **do**

**21**        current_NLL $\leftarrow 0$

**22**        **foreach** $(x, y) \in \mathcal{D}_{\text{cal}}$ **do**

**23**           **foreach** $m = 1, \ldots, M$ **do**

**24**              $f_m^{\text{TS}}(x) \leftarrow f_m(x)/c_1$

**25**           **foreach** $m = 1, \ldots, M$ **do**

**26**              $f_m^{\text{JUCAL}}(x) \leftarrow (1 - c_2) \cdot \frac{1}{M} \sum_{m'=1}^{M} f_{m'}^{\text{TS}}(x) + c_2 \cdot f_m^{\text{TS}}(x)$

**27**           $\bar{p}^{\text{JUCAL}}(x) \leftarrow \frac{1}{M} \sum_{m=1}^{M} \text{Softmax}(f_m^{\text{JUCAL}}(x))$

**28**           current_NLL $\leftarrow$ current_NLL + NLL$(\bar{p}^{\text{JUCAL}}(x), y)$

**29**        **if** *current_NLL $<$ best_NLL* **then**

**30**           best_NLL $\leftarrow$ current_NLL

**31**           $(c_1^*, c_2^*) \leftarrow (c_1, c_2)$

**return :** $(c_1^*, c_2^*)$

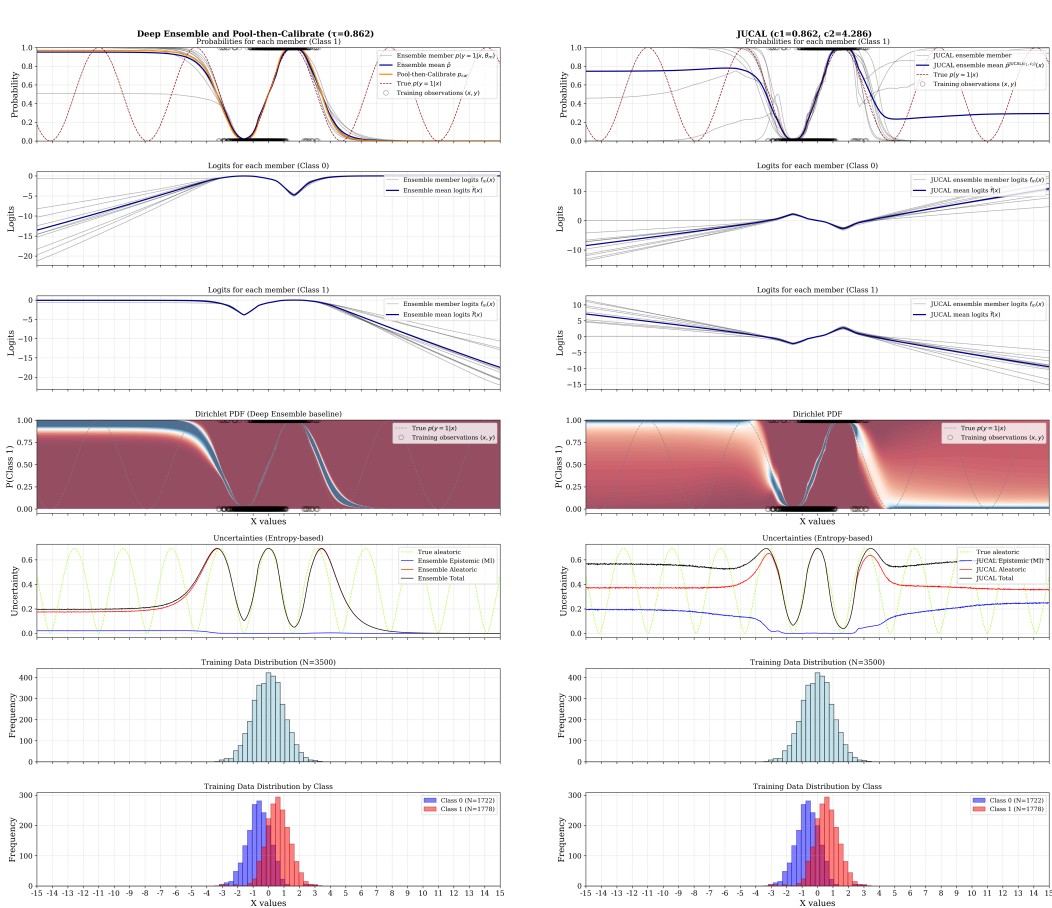

Figure 13: The same ensemble without and with JUCAL calibration. The logit diversity increases as you move further OOD, but the probability-diversity can simultaneously decrease if the logit diversity does not grow fast enough. JUCAL can scale the logit-diversity via $c_2$ to prevent this.

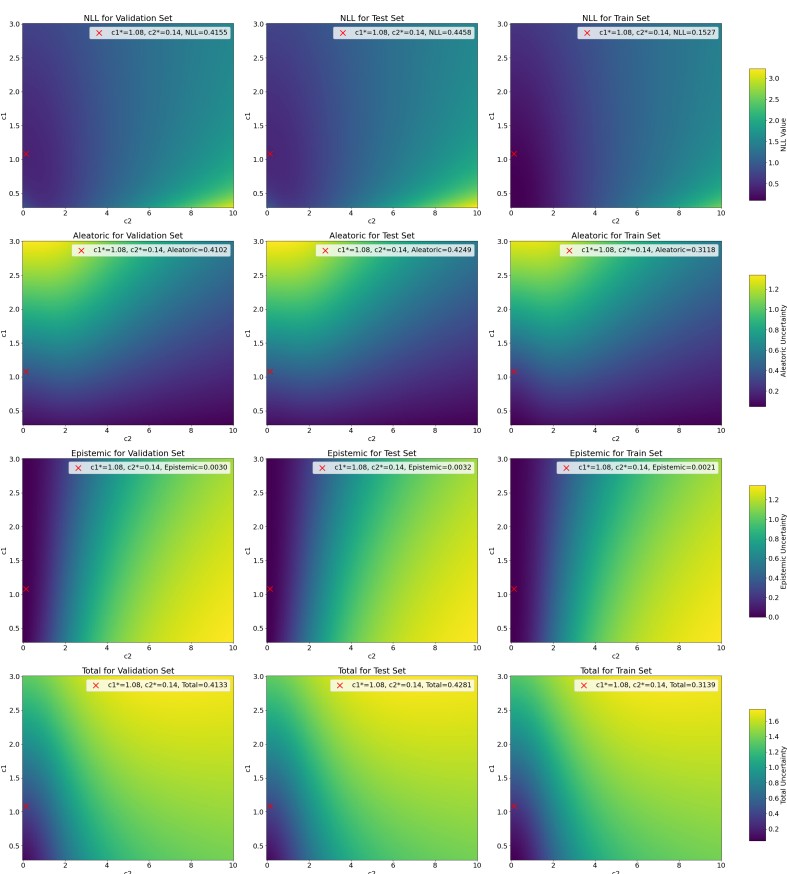

Figure 14: For an ensemble consisting of 5 CNNs trained on CIFAR-10, we compute multiple quantities on the training, validation, and test datasets for multiple different values of $c_1$ and $c_2$. We used Equations (6) and (7) from Appendix A.2.1 to compute aleatoric, epistemic, and total uncertainty, while there would be other alternatives too (see Appendix A.2).

# E    EXTENDED VERSIONS OF METHOD

## E.1    IMPLEMENTATION OF JUCAL WITH REDUCED COMPUTATIONAL COSTS

Since the computational costs of JUCAL are already almost negligible compared to the training costs (even compared to the LoRA-fine-tuning costs) (see Appendix H), one could simply implement JUCAL as suggested in Algorithm 1. However, we implemented a computationally even cheaper version of JUCAL in Algorithm 2, where we, in a first step, optimize $c_1, c_2$ on a coarse grid, and then, in a second step, locally refine $c_1, c_2$ by optimizing them again over a finer grid locally around the solution from the first step.

## E.2    ENSEMBLE SLECTION

Within the PCS framework (Yu, 2020; Yu & Barter, 2024), model selection techniques support the *Predictability* principle, serving as a statistical reality check to ensure that the selected ensemble is well-aligned with empirical results. It follows from common sense that we only want to add ensemble members who positively contribute to the repetitive performance of our ensemble. For example, Yu (2020); Yu & Barter (2024); Agarwal et al. (2025) suggest removing all the ensemble members with hyperparameters that result in poor predictive validation performance. Also, the experiments Arango et al. (2024) empirically suggest that using only the top $M$ ensemble members from the validation dataset typically performs better on the test dataset than using all ensemble members or only the top 1 ensemble members. However, Arango et al. (2024) also empirically show that Greedy-50, as suggested by Caruana et al. (2004; 2006), achieves the best test-NLL across all 12 LLM-datasets among multiple considered ensembling strategies (Single-Best, Random-5, Random-50, Top-5, Top-50, Model Average, Greedy-5, and Greedy-50). Therefore, we used Greedy-50 and Greedy-5 for ensemble selection for the experiments in Section 5. In Section 5, we applied JUCAL directly on the ensembles selected by Greedy-50 and Greedy-5. In the following, we propose we propose three modifications of Greedy-$M$.

Algorithm 3 presents a calibration-aware greedy ensemble selection strategy that incrementally constructs an ensemble to minimize the mean negative log-likelihood ($\text{NLL}_{\text{mean}}$). Starting from a temperature-scaled set of individually strong models, the algorithm selects an initial subset based on their individual validation-NLL performance, then applies the JUCAL procedure to jointly calibrate this subset by optimizing $(c_1, c_2)$. New members are greedily added based on their marginal improvement to ensemble-level $\text{NLL}_{\text{mean}}$, with optional recalibration after each addition when mode = "r.c." is enabled. We call this algorithm *Greedy-M re-calibrate once* (G$M$ r.c.o.) if mode = "r.c.o." is selected and *Greedy-M re-calibrate* (G$M$ r.c.) if mode = "r.c." is selected. This process encourages the construction of a diverse yet sharp ensemble, with calibration tightly integrated into the selection loop.

We designed this ensembling strategy to improve upon our main implementation of JUCAL (Algorithm 1). The key motivation for Algorithm 3 is the following: Plain Greedy-$M$ selects the ensemble such that it minimizes the validation-NLL for $c_1 = 1, c_2 = 1$, but JUCAL will change $c_1, c_2$ afterwards. Therefore Algorithm 3 attempts to approximately account already to some extent for the fact that $c_1, c_2$ can be different from one, when JUCAL is applied. In Appendix F we empirically compare both versions of Algorithm 3 to Greedy-$M$. Algorithm 3 can partially even further improve JUCAL's results; however, the slightly refined ensemble selections seem rather negligible compared to the magnitude of improvement from JUCAL itself. It would be interesting future work to apply JUCAL also every time directly after Line 20 in Algorithm 3 to fully adjust the ensemble selection to JUCAL.

Furthermore, we also propose a simple yet slightly different selection strategy in comparison to *Greedy-50* to select *Greedy-5 (unique)*. Algorithm 4 presents how *Greedy-5 (unique)* members are selected by first initializing an empty ensemble and then iteratively adding the model that yields the greatest reduction in mean negative log-likelihood (NLL) on the validation set. This process continues until five *unique* ensemble members have been selected, regardless of the total number of additions. In contrast to *Greedy-50*, which continues for a fixed total number of $M^*$ selections, *Greedy-5 (unique)* terminates early once the target number of distinct models is reached, but we have not included it in our experiments.

---

**Algorithm 3:** Greedy-$M$ re-calibrated (once) ensemble selection based on JUCAL

---

**Input** : Ensemble $\mathcal{E} = \{f_1, \ldots, f_M\}$, validation set $\mathcal{D}_{\text{val}}$, target size $M^*$, $N_{\text{init}}$,
$\quad\quad\quad$ mode $\in \{$"r.c.", "r.c.o."$\}$

1 Initialize best NLL $\leftarrow \infty$ and $c_1'^* \leftarrow$ arbitrary $\quad\quad\quad\quad\quad\quad\quad\quad$ ▷ Temperature scaling

2 **foreach** $c_1'$ *in grid* **do**

3 $\quad$ Set current NLL $\leftarrow 0$

4 $\quad$ **foreach** $(x, y) \in \mathcal{D}_{\text{val}}$ **do**

5 $\quad\quad$ **foreach** $m = 1, \ldots, M$ **do**

6 $\quad\quad\quad$ Compute $f_m^{\text{TS}}(x) \leftarrow f_m(x)/c_1'$

7 $\quad\quad$ Compute $\bar{p}(x; c_1') \leftarrow \frac{1}{M} \sum_{m=1}^{M} \text{Softmax}(f_m^{\text{TS}}(x))$

8 $\quad\quad$ current NLL $\leftarrow$ current NLL + NLL$(\bar{p}(x; c_1'), y)$

9 $\quad$ **if** *current NLL < best NLL* **then**

10 $\quad\quad$ Update best NLL $\leftarrow$ current NLL and $c_1'^* \leftarrow c_1'$

11 Select top $N_{\text{init}}$ models with lowest NLL to form $\mathcal{E}_{\text{init}}$ $\quad\quad\quad$ ▷ Initial ensemble selection

12

13 Apply Algorithm 1 to $\mathcal{E}_{\text{init}} \rightarrow$ obtain $(c_1^*, c_2^*)$ $\quad\quad\quad\quad\quad$ ▷ Run JUCAL on initial subset

14

15 Initialize $\mathcal{E} \leftarrow \mathcal{E}_{\text{init}}$ and best NLL $\leftarrow$ NLL$_{\text{mean}}(\mathcal{E}; c_1^*, c_2^*)$ $\quad\quad$ ▷ Greedy forward selection

16 **while** $|\mathcal{E}| < M^*$ **do**

17 $\quad$ **foreach** $f_m \in \{f_1, \ldots, f_M\} \setminus \mathcal{E}$ **do**

18 $\quad\quad$ Let $\mathcal{E}' \leftarrow \mathcal{E} \cup \{f_m\}$

19 $\quad\quad$ **foreach** $(x, y) \in \mathcal{D}_{\text{val}}$ **do**

20 $\quad\quad\quad$ **foreach** $f_m \in \mathcal{E}'$ **do**

21 $\quad\quad\quad\quad$ Compute $f_m^{\text{TS}}(x) \leftarrow f_m(x)/c_1^*$

22 $\quad\quad\quad\quad$ Compute $f_m^{\text{JUCAL}}(x) \leftarrow (1 - c_2^*) \cdot \frac{1}{|\mathcal{E}'|} \sum f_{m'}^{\text{TS}}(x) + c_2^* \cdot f_m^{\text{TS}}(x)$

23 $\quad\quad\quad$ Compute $\bar{p}(x; c_1^*, c_2^*) \leftarrow \frac{1}{|\mathcal{E}'|} \sum \text{Softmax}(f_m^{\text{JUCAL}}(x))$

24 $\quad\quad\quad$ Accumulate NLL$(\bar{p}(x; c_1^*, c_2^*), y)$

25 $\quad\quad$ Store NLL$_{\text{mean}}(\mathcal{E}')$

26 $\quad$ Identify $f_{m^*}$ giving lowest NLL$_{\text{mean}}$

27 $\quad$ **if** *NLL improves* **then**

28 $\quad\quad$ $\mathcal{E} \leftarrow \mathcal{E} \cup \{f_{m^*}\}$

29 $\quad\quad$ Update best NLL $\leftarrow$ NLL$_{\text{mean}}(\mathcal{E}; c_1^*, c_2^*)$

30 $\quad\quad$ **if** mode $=$ *"r.c."* **then**

31 $\quad\quad\quad$ Apply Algorithm 1 to $\mathcal{E} \rightarrow$ obtain $(c_1^*, c_2^*)$ $\quad\quad$ ▷ Run JUCAL on updated subset

32 $\quad$ **else**

33 $\quad\quad$ **break** $\quad\quad\quad\quad\quad\quad\quad\quad\quad\quad\quad\quad\quad\quad$ ▷ No further improvement

**return** : Ensemble set $\mathcal{E}$

---

**Algorithm 4:** Greedy-5 (unique) ensemble selection with unique members (simple extension of Greedy-M in (Arango et al., 2024)).

**Input** : Ensemble $\mathcal{E} = \{f_1, \ldots, f_M\}$, validation set $\mathcal{D}_{\text{val}}$, $m = 5$

1  Initialize $\mathcal{E} \leftarrow \emptyset$, $\text{NLL}_{\text{best}} \leftarrow \infty$

2  **for** $t = 1$ *to* $T \gg t$ **do**

3      **if** $|\mathcal{E}| \geq m$ **then**

4          **break**

5      $f_{\text{best}} \leftarrow \text{None}$

6      **foreach** $f_j \in \mathcal{R}$ **do**

7          $\mathcal{E}' \leftarrow \mathcal{E} \cup \{j\}$

8          Compute $\bar{p}(x) \leftarrow \frac{1}{|\mathcal{E}'|} \sum\limits_{j' \in \mathcal{E}'} \text{Softmax}(f_{j'}(x))$

9          Compute $\text{NLL} \leftarrow -\frac{1}{|\mathcal{D}_{\text{val}}|} \sum\limits_{(x,y) \in \mathcal{D}_{\text{val}}} \log \bar{p}_y(x)$

10         **if** *NLL* < *NLL*$_{best}$ **then**

11             $\text{NLL}_{\text{best}} \leftarrow \text{NLL}, \quad f_{\text{best}} \leftarrow j$

**return** : Ensemble set $\mathcal{E}$

# F  TABLES AND FIGURES

## F.1  TABLES WITH DETAILED RESULTS

Tables 2, 4 to 6, 8 and 9 present the experimental results for JUCAL (Algorithm 1) and its extensions, using Algorithms 3 and 4. Here, G5 denotes *Greedy-5* and G50 denotes *Greedy-50*. When an ensemble strategy is followed by *t.s.*, it indicates temperature scaling via the *pool-then-calibrate* approach. The abbreviation *r.c.o.* stands for *re-calibrated once*, where Algorithm 3 is applied with mode = "r.c.o.". In contrast, *r.c.* refers to *re-calibrated*, where Algorithm 3 is used with mode = "r.c.".

Table 2: FTC-metadataset full: Negative log-likelihood (NLL$_{mean}$ over data splits; mean ± 95% confidence interval half-width) on the full dataset (100%). The best mean is shown in bold, and methods not significantly different from the best (paired test, $\alpha = 0.05$) are shaded.

| Ensemble Type | DBpedia | News | SST-2 | SetFit | Tweet | IMDB |
|---|---|---|---|---|---|---|
| G5 | 0.0376 ± 0.0005 | 0.1682 ± 0.0048 | 0.1359 ± 0.0051 | 0.5465 ± 0.0033 | 0.5095 ± 0.0089 | 0.1171 ± 0.0028 |
| G5 p.t.c. | 0.0348 ± 0.0007 | 0.1618 ± 0.0052 | 0.1208 ± 0.0040 | 0.5431 ± 0.0019 | 0.5012 ± 0.0052 | 0.1018 ± 0.0022 |
| G5 JUCAL | 0.0290 ± 0.0004 | 0.1479 ± 0.0023 | 0.1143 ± 0.0032 | 0.4965 ± 0.0013 | 0.4772 ± 0.0028 | 0.1005 ± 0.0018 |
| G50 | 0.0349 ± 0.0005 | 0.1541 ± 0.0043 | 0.1137 ± 0.0039 | 0.531 ± 0.0016 | 0.4763 ± 0.0052 | 0.1050 ± 0.0026 |
| G50 p.t.c. | 0.0331 ± 0.0003 | 0.1510 ± 0.0037 | 0.1130 ± 0.0035 | 0.5309 ± 0.0016 | 0.4758 ± 0.0049 | 0.1042 ± 0.0019 |
| G50 JUCAL | **0.0288 ± 0.0004** | **0.1423 ± 0.0024** | 0.1090 ± 0.0032 | 0.4972 ± 0.0018 | 0.4680 ± 0.0045 | 0.0983 ± 0.0017 |
| G50 r.c.o. JUCAL | 0.0291 ± 0.0004 | 0.1425 ± 0.0032 | 0.1087 ± 0.0031 | **0.4909 ± 0.0012** | 0.4594 ± 0.0051 | 0.0974 ± 0.0017 |
| G50 r.c. JUCAL | 0.0290 ± 0.0005 | 0.1433 ± 0.0029 | **0.1075 ± 0.0035** | 0.4938 ± 0.0014 | **0.4594 ± 0.0051** | **0.0970 ± 0.0013** |

Table 3: FTC-metadataset full: Area Under the Rejection-Accuracy Curve (AURAC) over data splits; mean ± 95% confidence interval half-width) on the full dataset (100%). The best mean is shown in bold, and methods not significantly different from the best (paired test, $\alpha = 0.05$) are shaded.

| Ensemble Type | DBpedia | News | SST-2 | SetFit | Tweet | IMDB |
|---|---|---|---|---|---|---|
| G5 | 0.9895 ± 0.0 | 0.981 ± 0.0011 | 0.984 ± 0.0005 | 0.8915 ± 0.0008 | 0.9103 ± 0.0028 | **0.9859 ± 0.0002** |
| G5 p.t.c. | 0.9895 ± 0.0 | 0.981 ± 0.0011 | 0.984 ± 0.0005 | 0.8915 ± 0.0008 | 0.9103 ± 0.0027 | 0.9859 ± 0.0002 |
| G5 JUCAL | 0.9897 ± 0.0 | 0.9829 ± 0.0005 | 0.9842 ± 0.0005 | 0.924 ± 0.0006 | 0.9211 ± 0.0006 | 0.9858 ± 0.0002 |
| G50 | 0.9895 ± 0.0 | 0.981 ± 0.0008 | 0.9833 ± 0.0005 | 0.9023 ± 0.0006 | 0.9157 ± 0.0021 | 0.9838 ± 0.0003 |
| G50 p.t.c. | 0.9895 ± 0.0 | 0.981 ± 0.0008 | 0.9833 ± 0.0005 | 0.9023 ± 0.0006 | 0.9158 ± 0.0021 | 0.9838 ± 0.0003 |
| G50 JUCAL | **0.9897 ± 0.0** | 0.9835 ± 0.0005 | 0.9849 ± 0.0005 | 0.9237 ± 0.0007 | 0.9236 ± 0.0014 | 0.9855 ± 0.0002 |
| G50 r.c.o. JUCAL | 0.9897 ± 0.0 | **0.9837 ± 0.0005** | 0.985 ± 0.0004 | **0.9252 ± 0.0005** | **0.9249 ± 0.0013** | 0.9859 ± 0.0002 |
| G50 r.c. JUCAL | 0.9897 ± 0.0 | 0.9837 ± 0.0005 | **0.985 ± 0.0005** | 0.9226 ± 0.0005 | 0.9244 ± 0.0015 | 0.9859 ± 0.0002 |

Table 4: FTC-metadataset full: Area under the ROC (AUROC over data splits; mean ± 95% confidence interval half-width) on the full dataset (100%). The best mean is shown in bold, and methods not significantly different from the best (paired test, $\alpha = 0.05$) are shaded.

| Ensemble Type | DBpedia | News | SST-2 | SetFit | Tweet | IMDB |
|---|---|---|---|---|---|---|
| G5 | 0.9998312 ± 0.0 | 0.9929 ± 0.0007 | 0.9907 ± 0.0007 | 0.9144 ± 0.0008 | 0.9316 ± 0.0019 | **0.9934 ± 0.0003** |
| G5 p.t.c. | 0.9998311 ± 0.0 | 0.9929 ± 0.0007 | 0.9907 ± 0.0007 | 0.9144 ± 0.0008 | 0.9316 ± 0.0018 | 0.9934 ± 0.0003 |
| G5 JUCAL | 0.9998758 ± 0.0 | 0.9943 ± 0.0003 | 0.9912 ± 0.0006 | 0.9377 ± 0.0004 | 0.9383 ± 0.0010 | 0.9934 ± 0.0002 |
| G50 | 0.9998198 ± 0.0 | 0.9931 ± 0.0005 | 0.9898 ± 0.0007 | 0.9229 ± 0.0005 | 0.9369 ± 0.0014 | 0.9911 ± 0.0003 |
| G50 p.t.c. | 0.9998199 ± 0.0 | 0.9931 ± 0.0005 | 0.9898 ± 0.0007 | 0.9229 ± 0.0005 | 0.9369 ± 0.0014 | 0.9911 ± 0.0003 |
| G50 JUCAL | **0.9998785 ± 0.0** | **0.9948 ± 0.0004** | 0.9917 ± 0.0007 | 0.9371 ± 0.0006 | 0.9405 ± 0.0014 | 0.9930 ± 0.0004 |
| G50 r.c.o. JUCAL | 0.9998632 ± 0.0 | 0.9947 ± 0.0003 | 0.9918 ± 0.0006 | **0.9386 ± 0.0003** | **0.9408 ± 0.0013** | 0.9934 ± 0.0002 |
| G50 r.c. JUCAL | 0.9998660 ± 0.0 | 0.9947 ± 0.0003 | **0.9919 ± 0.0007** | 0.9362 ± 0.0003 | 0.9405 ± 0.0013 | 0.9933 ± 0.0002 |

Table 5: FTC-metadataset full: Set size over data splits; mean ± 95% confidence interval half-width) on the full dataset (100%). The best mean is shown in bold, and methods not significantly different from the best (paired test, $\alpha = 0.05$) are shaded. Here the coverage threshold is 99% for all but DBpedia where it is 99.9%

| Ensemble Type | DBpedia | News | SST-2 | SetFit | Tweet | IMDB |
|---|---|---|---|---|---|---|
| G5 | 1.2941 ± 0.0395 | 1.3517 ± 0.0385 | 1.1544 ± 0.0097 | 2.6642 ± 0.0228 | 2.3281 ± 0.0963 | 1.0996 ± 0.0065 |
| G5 p.t.c. | 1.3008 ± 0.0484 | 1.3591 ± 0.0424 | 1.1550 ± 0.0107 | 2.6567 ± 0.0209 | 2.3313 ± 0.0993 | 1.1003 ± 0.0062 |
| G5 JUCAL | 1.2270 ± 0.0438 | 1.2490 ± 0.0161 | 1.1459 ± 0.0116 | 2.2368 ± 0.0231 | 2.1286 ± 0.0722 | 1.1004 ± 0.0039 |
| G50 | 1.3516 ± 0.0428 | 1.3436 ± 0.0313 | 1.1617 ± 0.0148 | 2.6519 ± 0.0237 | 2.2280 ± 0.0507 | 1.1135 ± 0.0070 |
| G50 p.t.c. | 1.3534 ± 0.0398 | 1.3517 ± 0.0226 | 1.1621 ± 0.0175 | 2.6514 ± 0.0490 | 2.2261 ± 0.0476 | 1.1140 ± 0.0092 |
| G50 JUCAL | **1.2072 ± 0.0358** | **1.2228 ± 0.0244** | 1.1385 ± 0.0094 | **2.2334 ± 0.0199** | 2.0633 ± 0.0291 | 1.1005 ± 0.0051 |
| G50 r.c.o. JUCAL | 1.2355 ± 0.0554 | 1.2350 ± 0.0213 | 1.1397 ± 0.0112 | 2.2431 ± 0.0200 | 2.0596 ± 0.0411 | 1.0995 ± 0.0020 |
| G50 r.c. JUCAL | 1.2259 ± 0.0382 | 1.2429 ± 0.0215 | **1.1317 ± 0.0113** | 2.2766 ± 0.0279 | **2.0475 ± 0.0328** | 1.0988 ± 0.0023 |

Table 6: FTC-metadataset mini (10%): Negative log-likelihood (NLL$_{mean}$ over data splits; mean ± 95% confidence interval half-width) on the full dataset (100%). The best mean is shown in bold, and methods not significantly different from the best (paired test, $\alpha = 0.05$) are shaded.

| Ensemble Type | DBpedia | News | SST-2 | SetFit | Tweet | IMDB |
|---|---|---|---|---|---|---|
| G5 | 0.0432 ± 0.0012 | 0.2321 ± 0.0031 | 0.1534 ± 0.0044 | 0.4067 ± 0.002 | 0.5311 ± 0.0065 | 0.1334 ± 0.0064 |
| G5 p.t.c. | 0.0341 ± 0.0008 | 0.2050 ± 0.0026 | 0.1472 ± 0.0020 | 0.4051 ± 0.0018 | 0.5294 ± 0.0062 | 0.1314 ± 0.0043 |
| G5 JUCAL | 0.0326 ± 0.0008 | 0.1966 ± 0.0026 | 0.1396 ± 0.002 | 0.3684 ± 0.0018 | 0.5205 ± 0.0059 | 0.1303 ± 0.0034 |
| G50 | 0.0352 ± 0.0009 | 0.1967 ± 0.0032 | 0.1320 ± 0.0035 | 0.3594 ± 0.0014 | 0.4980 ± 0.0063 | 0.1258 ± 0.0020 |
| G50 p.t.c. | 0.0346 ± 0.0004 | 0.1964 ± 0.0031 | 0.1320 ± 0.0034 | 0.3594 ± 0.0014 | 0.4979 ± 0.0061 | 0.1255 ± 0.0014 |
| G50 JUCAL | **0.0305 ± 0.0008** | **0.1899 ± 0.0028** | 0.1309 ± 0.0034 | **0.3480 ± 0.0013** | **0.4979 ± 0.0059** | 0.1257 ± 0.0018 |
| G50 r.c.o. JUCAL | 0.0309 ± 0.0007 | 0.1911 ± 0.0035 | 0.1335 ± 0.0025 | 0.3602 ± 0.0023 | 0.5038 ± 0.0048 | 0.1249 ± 0.0020 |
| G50 r.c. JUCAL | 0.0308 ± 0.0007 | 0.1904 ± 0.0033 | 0.1345 ± 0.0028 | 0.3516 ± 0.0012 | 0.4997 ± 0.0059 | **0.1248 ± 0.0018** |

Table 7: FTC-metadataset mini (10%): Area Under the Rejection-Accuracy Curve (AURAC) over data splits; mean ± 95% confidence interval half-width) on the full dataset (100%). The best mean is shown in bold, and methods not significantly different from the best (paired test, $\alpha = 0.05$) are shaded.

| Ensemble Type | DBpedia | News | SST-2 | SetFit | Tweet | IMDB |
|---|---|---|---|---|---|---|
| G5 | 0.9895 ± 0.0001 | 0.9769 ± 0.0002 | 0.979 ± 0.0008 | 0.9406 ± 0.0005 | 0.8982 ± 0.0026 | 0.9809 ± 0.0006 |
| G5 p.t.c. | 0.9895 ± 0.0001 | 0.9769 ± 0.0003 | 0.979 ± 0.0008 | 0.9407 ± 0.0005 | 0.8981 ± 0.0026 | 0.9809 ± 0.0006 |
| G5 JUCAL | 0.9895 ± 0.0001 | 0.9779 ± 0.0003 | 0.9817 ± 0.0004 | 0.95 ± 0.0005 | 0.9025 ± 0.0018 | 0.9819 ± 0.0005 |
| G50 | 0.9893 ± 0.0001 | 0.9748 ± 0.0005 | 0.9822 ± 0.0005 | 0.9503 ± 0.0003 | 0.9091 ± 0.0018 | 0.9821 ± 0.0004 |
| G50 p.t.c. | 0.9893 ± 0.0001 | 0.9748 ± 0.0005 | 0.9822 ± 0.0005 | 0.9503 ± 0.0003 | 0.9091 ± 0.0018 | 0.9821 ± 0.0004 |
| G50 JUCAL | 0.9896 ± 0.0 | 0.978 ± 0.0005 | **0.9828 ± 0.0005** | **0.9554 ± 0.0002** | **0.9099 ± 0.0017** | **0.9822 ± 0.0005** |
| G50 r.c.o. JUCAL | **0.9896 ± 0.0001** | **0.9783 ± 0.0006** | 0.982 ± 0.0004 | 0.9531 ± 0.0003 | 0.9094 ± 0.002 | 0.9819 ± 0.0003 |
| G50 r.c. JUCAL | 0.9896 ± 0.0001 | 0.9781 ± 0.0006 | 0.9821 ± 0.0002 | 0.9544 ± 0.0002 | 0.9097 ± 0.0014 | 0.9815 ± 0.0006 |

Table 8: FTC-metadataset mini (10%): Are under the ROC (AUROC over data splits; mean ± 95% confidence interval half-width) on the full dataset (100%). The best mean is shown in bold, and methods not significantly different from the best (paired test, $\alpha = 0.05$) are shaded.

| Ensemble Type | DBpedia | News | SST-2 | SetFit | Tweet | IMDB |
|---|---|---|---|---|---|---|
| G5 | 0.9998 ± 0.0 | 0.9899 ± 0.0002 | 0.9853 ± 0.0007 | 0.9539 ± 0.0004 | 0.9226 ± 0.0015 | 0.9872 ± 0.0007 |
| G5 p.t.c. | 0.9998 ± 0.0 | 0.9899 ± 0.0001 | 0.9853 ± 0.0007 | 0.9539 ± 0.0004 | 0.9225 ± 0.0015 | 0.9872 ± 0.0007 |
| G5 JUCAL | 0.9998 ± 0.0 | 0.9905 ± 0.0002 | 0.9874 ± 0.0005 | 0.9620 ± 0.0004 | 0.9253 ± 0.0019 | 0.9883 ± 0.0006 |
| G50 | 0.9997 ± 0.0 | 0.9889 ± 0.0005 | 0.9878 ± 0.0008 | 0.9632 ± 0.0002 | 0.9302 ± 0.0013 | 0.9886 ± 0.0001 |
| G50 p.t.c. | 0.9997 ± 0.0 | 0.9889 ± 0.0005 | 0.9878 ± 0.0008 | 0.9632 ± 0.0002 | 0.9302 ± 0.0013 | 0.9886 ± 0.0001 |
| G50 JUCAL | 0.9998 ± 0.0 | **0.9907 ± 0.0003** | **0.9885 ± 0.0008** | **0.9667 ± 0.0001** | **0.9306 ± 0.0012** | 0.9886 ± 0.0002 |
| G50 r.c.o. JUCAL | **0.9999 ± 0.0** | 0.9906 ± 0.0004 | 0.9879 ± 0.0005 | 0.9649 ± 0.0007 | 0.9298 ± 0.0007 | **0.9891 ± 0.0005** |
| G50 r.c. JUCAL | 0.9998 ± 0.0 | 0.9907 ± 0.0003 | 0.9878 ± 0.0005 | 0.9658 ± 0.0002 | 0.9303 ± 0.0011 | 0.9890 ± 0.0005 |

Table 9: FTC-metadataset mini (10%): Set size over data splits; mean ± 95% confidence interval half-width) on the full dataset (100%). The best mean is shown in bold, and methods not significantly different from the best (paired test, $\alpha = 0.05$) are shaded. Here the coverage threshold is 99% for all but DBpedia where it is 99.9%

| Ensemble Type | DBpedia | News | SST-2 | SetFit | Tweet | IMDB |
|---|---|---|---|---|---|---|
| G5 | 1.3673 ± 0.0702 | 1.4414 ± 0.0167 | 1.2467 ± 0.0135 | 2.2989 ± 0.0027 | 2.2997 ± 0.0356 | 1.2392 ± 0.0099 |
| G5 p.t.c. | 1.4313 ± 0.0695 | 1.4475 ± 0.0184 | 1.2504 ± 0.0149 | 2.3124 ± 0.0136 | 2.3028 ± 0.0366 | 1.2347 ± 0.0158 |
| G5 JUCAL | 1.4522 ± 0.0567 | **1.4131 ± 0.0276** | 1.2091 ± 0.0115 | 1.9976 ± 0.0254 | 2.2110 ± 0.0277 | 1.2148 ± 0.0115 |
| G50 | 1.6459 ± 0.0546 | 1.7193 ± 0.0952 | 1.1918 ± 0.0132 | 2.1899 ± 0.0061 | 2.1735 ± 0.0475 | 1.1821 ± 0.0043 |
| G50 p.t.c. | 1.6453 ± 0.0563 | 1.7274 ± 0.0792 | 1.1933 ± 0.0119 | 2.2008 ± 0.0059 | 2.1684 ± 0.0414 | 1.1831 ± 0.0092 |
| G50 JUCAL | **1.3105 ± 0.0232** | 1.4389 ± 0.0334 | **1.1862 ± 0.0120** | **1.8980 ± 0.0086** | **2.1470 ± 0.0444** | 1.1819 ± 0.0126 |
| G50 r.c.o. JUCAL | 1.3552 ± 0.0575 | 1.4243 ± 0.0345 | 1.1874 ± 0.0068 | 1.9958 ± 0.0303 | 2.2385 ± 0.0414 | 1.1698 ± 0.0045 |
| G50 r.c. JUCAL | 1.339 ± 0.0478 | 1.4384 ± 0.0154 | 1.1956 ± 0.0081 | 1.9198 ± 0.0240 | 2.2213 ± 0.0255 | **1.1677 ± 0.0064** |

### F.2 RESULTS ON EXPECTED CALIBRATION ERROR (ECE)

Note that the ECE suffers from severe limitations as an evaluation metric. In contrast to the NLL and the Brier Score displayed in Figures 4 and 5, the ECE is not a strictly proper scoring rule (see Appendix I.2 for more details on the theoretical properties of strictly proper scoring rules).

The Expected Calibration Error (ECE) is calculated by partitioning the predictions into $M = 15$ equally spaced bins. Let $B_m$ be the set of indices of samples whose prediction confidence falls into the $m$-th bin. The ECE is defined as the weighted average of the absolute difference between the accuracy and the confidence of each bin:

$$\text{ECE} = \sum_{m=1}^{M} \frac{|B_m|}{n} \left| \text{acc}(B_m) - \text{conf}(B_m) \right|, \tag{16}$$

where $n$ is the total number of samples, $\text{acc}(B_m)$ is the average accuracy, and $\text{conf}(B_m)$ is the average confidence within bin $B_m$.

Because it is not a proper scoring rule, it can be trivially minimized by non-informative models. For example, a classifier that ignores the input features $x$ and assigns the same marginal class probabilities to every datapoint can achieve a perfect ECE of zero, despite having no discriminatory power.

Furthermore, one can artificially minimize ECE without improving the model's utility. Consider a method that replaces the top predicted probability for every datapoint with the model's overall average accuracy, while assigning random, smaller probabilities to the remaining classes. This "absurd" modification results in a perfectly calibrated model (ECE = 0) and maintains the original accuracy, yet it completely discards the useful, instance-specific uncertainty quantification required for safety-critical applications.

However, very high values of ECE indicate inaccurate uncertainty quantification. See Figures 15 and 16 for our ECE results. Note that while *calibrate-then-pool* overall achieved the 2nd best results after JUCAL in all metrics, *calibrate-then-pool* is one of the worst methods for ECE. JUCAL performs as good or better than *calibrate-then-pool* on all 24 LLM experiments and on 6 CNN experiments.

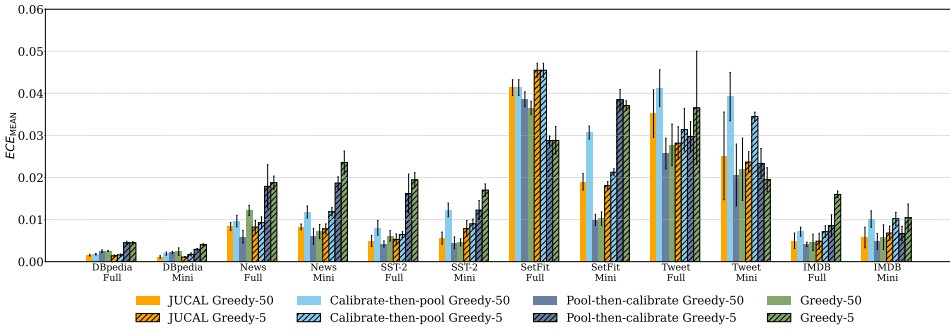

Figure 15: **ECE Results for Text Classification.** For the ECE, lower values (displayed on the y-axis) are better. On the x-axis, we list 12 text classification datasets (a 10%-mini and a 100%-full version of 6 distinct datasets). The striped bars correspond to ensemble size $M = 5$, while the non-striped bars correspond to $M = 50$. JUCAL's results are yellow. We show the average ECE and $\pm 1$ standard deviation across 5 random validation-test splits.

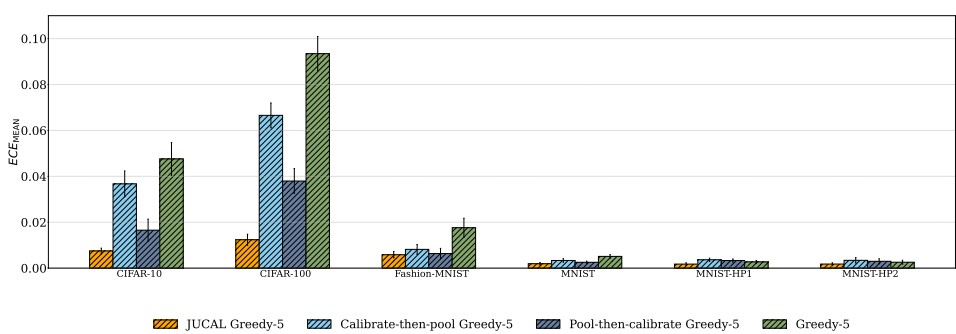

Figure 16: **ECE Results for Image Classification.** For the ECE, lower values (displayed on the y-axis) are better. On the x-axis, we list distinct image classification datasets (and two hyperparameter-ablation studies for MNIST). JUCAL's results are yellow. We show the average ECE and $\pm 1$ standard deviation across 10 random train-validation-test splits.

### F.3 RESULTS ON CONFORMAL PREDICTION SETS

Note that JUCAL does not need a conformal unseen calibration dataset, as JUCAL only reuses the already seen validation dataset. JUCAL outputs predictive distributions that can be conformalized in a separate step using an unseen calibration dataset. In this subsection, we compare APS-conformalized JUCAL against APS-conformalized versions of its competitors, where we apply APS-conformalization on the same unseen calibration dataset for all competitors using the predictive probabilities of each competitor to compute their APS-conformity scores (Romano et al., 2020). JUCAL shows as good or better overall performance than all considered competitors across all considered conformal metrics (average set size and average logarithm of the set size; see Figures 17 to 22). For multiple datasets, JUCAl simultaneously achieves smaller set sizes and slightly higher coverage than its competitors. Due to conformal guarantees, all conformalized methods achieve approximately the same marginal coverage on the test dataset (see Figures 21 and 22). In Appendix I.1.1, we discuss multiple limitations of conformal guarantees.

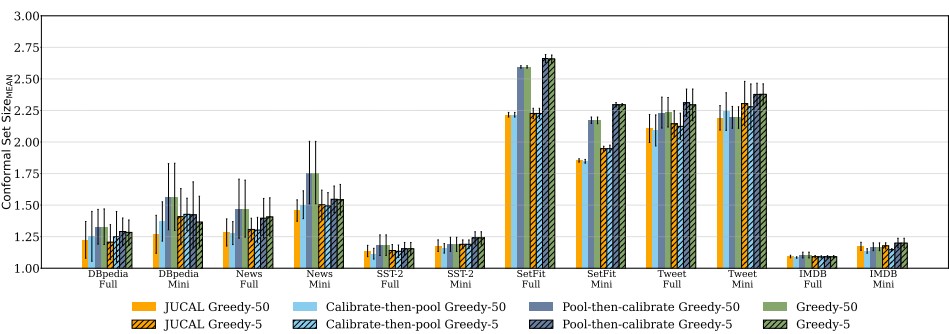

Figure 17: **Conformal Set Size Results for Text Classification.** For the conformal set size, lower values (displayed on the y-axis) are better. On the x-axis, we list 12 text classification datasets (a 10%-mini and a 100%-full version of 6 distinct datasets). The striped bars correspond to ensemble size $M = 5$, while the non-striped bars correspond to $M = 50$. JUCAL's results are yellow. We show the average conformal prediction set size (for the conformal target coverage threshold of 99.9% for *DBpedia* (Full and Mini) and 99% for all other datasets) and $\pm 1$ standard deviation across 5 random validation-test splits.

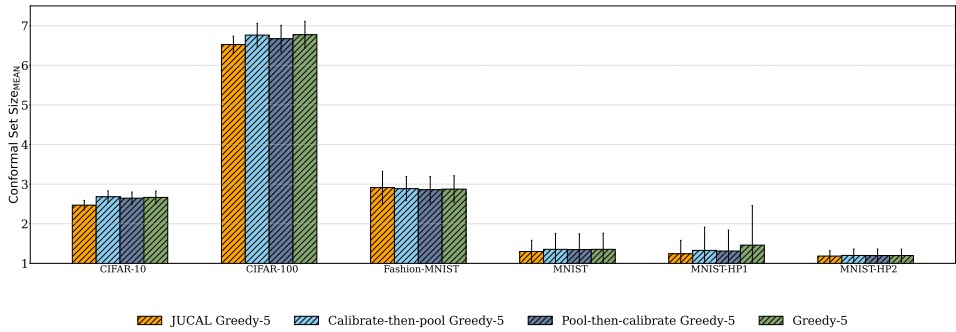

Figure 18: **Conformal Set Size Results for Image Classification.** For the conformal set size, lower values (displayed on the y-axis) are better. On the x-axis, we list distinct image classification datasets (and two hyperparameter-ablation studies for MNIST). JUCAL's results are yellow. We show the average conformal prediction set size (for the conformal target coverage threshold of 99% for *CIFAR-10*, 90% for *CIFAR-100*, and 99.9% for al variants of *MNIST* and *Fashion-MNIST*) and $\pm 1$ standard deviation across 10 random train-validation-test splits.

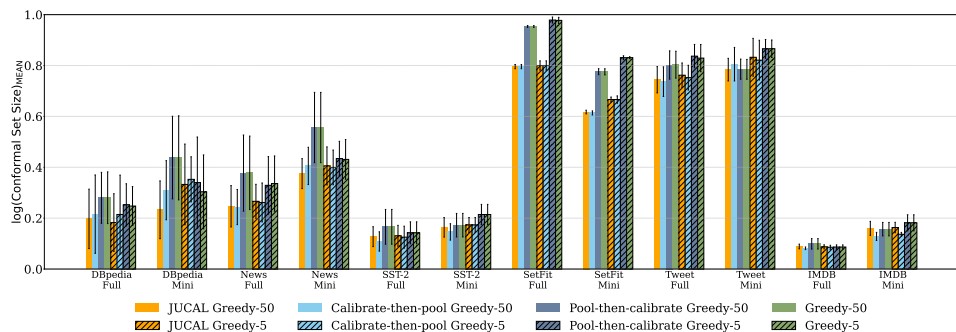

Figure 19: **Conformal Log Set Size Results for Text Classification.** For the conformal log set size, lower values (displayed on the y-axis) are better. On the x-axis, we list 12 text classification datasets (a 10%-mini and a 100%-full version of 6 distinct datasets). The striped bars correspond to ensemble size $M = 5$, while the non-striped bars correspond to $M = 50$. JUCAL's results are yellow. We show the average of the logarithm of the conformal prediction set size (for the conformal target coverage threshold of 99.9% for *DBpedia* (Full and Mini) and 99% for all other datasets) and $\pm 1$ standard deviation across 5 random validation-test splits.

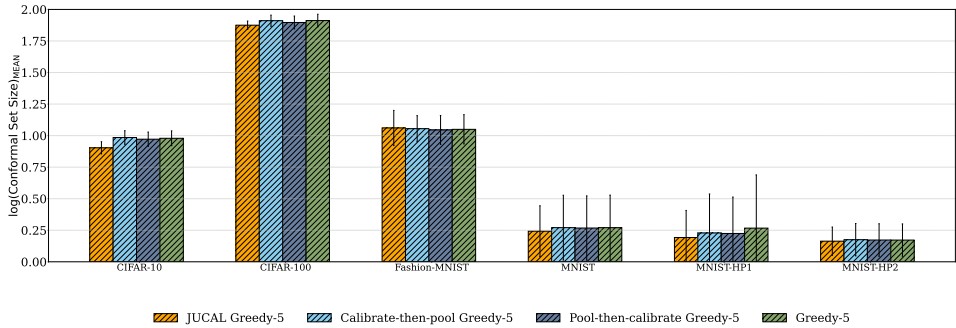

Figure 20: **Conformal Log Set Size Results for Image Classification.** For the conformal log set size, lower values (displayed on the y-axis) are better. On the x-axis, we list distinct image classification datasets (and two hyperparameter-ablation studies for MNIST). JUCAL's results are yellow. We show the average logarithmic conformal prediction set size (for the conformal target coverage threshold of 99% for *CIFAR-10*, 90% for *CIFAR-100*, and 99.9% for al variants of *MNIST* and *Fashion-MNIST*) and $\pm 1$ standard deviation across 10 random train-validation-test splits.

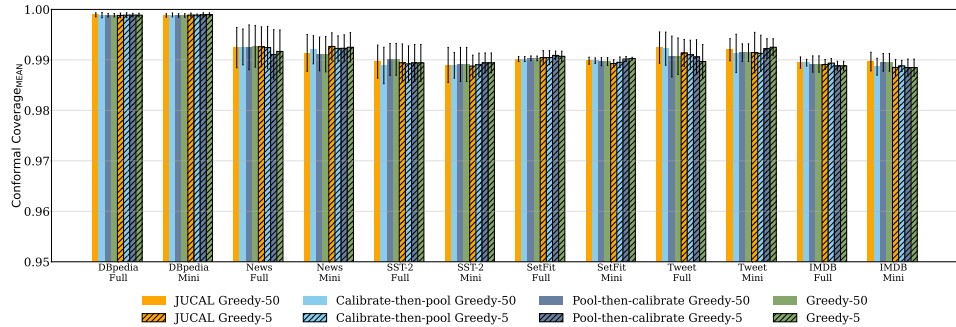

Figure 21: **Conformal Coverage Results for Text Classification.** For the conformal coverage, values near the target coverage indicate better calibration. Larger values of coverage are more desirable than smaller values of coverage (unless larger coverage leads to larger set sizes). On the x-axis, we list 12 text classification datasets (a 10%-mini and a 100%-full version of 6 distinct datasets). The striped bars correspond to ensemble size $M = 5$, while the non-striped bars correspond to $M = 50$. JUCAL's results are yellow. We show the average test-coverage (for the conformal target coverage threshold of 99.9% for *DBpedia* (Full and Mini) and 99% for all other datasets), and $\pm 1$ standard deviation across 5 random validation-test splits.

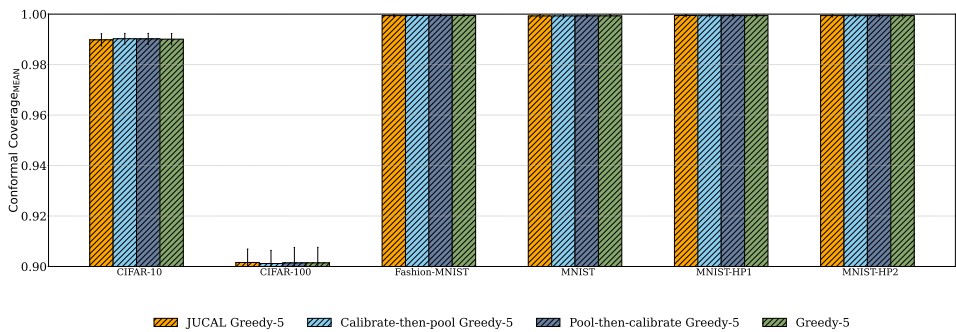

Figure 22: **Conformal Coverage Results for Image Classification.** For the conformal coverage, values near the target coverage indicate better calibration. Larger values of coverage are more desirable than smaller values of coverage (unless larger coverage leads to larger set sizes). On the x-axis, we list distinct image classification datasets (and two hyperparameter-ablation studies for MNIST). JUCAL's results are yellow. We show the average test-coverage (for the conformal target coverage threshold of 99% for *CIFAR-10*, 90% for *CIFAR-100*, and 99.9% for al variants of *MNIST* and *Fashion-MNIST*) and $\pm 1$ standard deviation across 10 random train-validation-test splits.

## F.4 FURTHER INTUITIVE LOW-DIMENSIONAL PLOTS

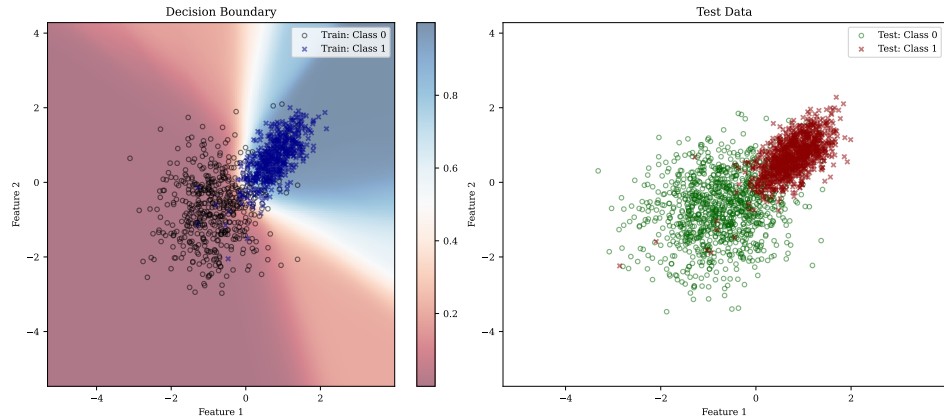

Figure 23: Softmax outputs visualizing the estimated predictive probabilities calibrated by JUCAL for a synthetic 2D binary classification task.

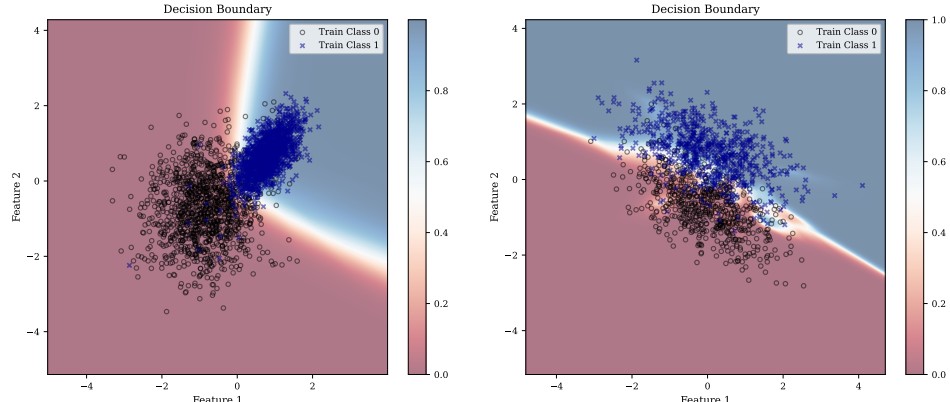

Figure 24: Softmax outputs visualizing the estimated predictive probabilities from a single neural network trained on two dataset configurations of ta synthetic 2D binary classification task.

## G   DETAILED DESCRIPTION OF METADATASET

The metadataset presented by Arango et al. (2024) and used in our study is designed to support analysis of uncertainty and calibration methods in text classification. It comprises model predictions across six diverse datasets, covering domains such as movie reviews, tweets, encyclopedic content, and news. Each dataset involves classification tasks with varying numbers of classes (details provided in Table 10).

The datasets include IMDB for sentiment analysis (Maas et al., 2011), Tweet Sentiment Extraction (Maggie & Culliton), AG News and DBpedia (Zhang et al., 2015), SST-2 (Socher et al., 2013), and SetFit (Tunstall et al., 2021). For each dataset, Arango et al. (2024) construct two versions: one trained with the full training split (100%), and another trained on a smaller subset comprising 10% of the original training data. All models are fine-tuned separately for each configuration.

Predictions are saved on validation and test splits to enable controlled evaluation of ensemble and calibration strategies. The validation split corresponds to 20% of the training data. For SST-2 and SetFit, where either test labels are not publicly released or are partially hidden, Arango et al. (2024) instead allocate 20% of the remaining training data to simulate a test set.

This setup allows for consistent comparison across tasks and supervision levels, facilitating the study of uncertainty estimation under varying domain and data conditions.

| Dataset | Classes | Members | Train Size | Valid Size | Test Size |
|---|---|---|---|---|---|
| DBpedia Full | 14 | 25 | 448,000 | 112,000 | 70,000 |
| DBpedia Mini | 14 | 65 | 44,800 | 112,000 | 70,000 |
| News Full | 4 | 99 | 96,000 | 24,000 | 7,600 |
| News Mini | 4 | 120 | 9,600 | 24,000 | 7,600 |
| SST-2 Full | 2 | 125 | 43,103 | 13,470 | 10,776 |
| SST-2 Mini | 2 | 125 | 4,310 | 13,470 | 10,776 |
| SetFit Full | 3 | 25 | 393,116 | 78,541 | 62,833 |
| SetFit Mini | 3 | 100 | 39,312 | 78,541 | 62,833 |
| Tweet Full | 3 | 100 | 27,485 | 5,497 | 3,534 |
| Tweet Mini | 3 | 100 | 2,748 | 5,497 | 3,534 |
| IMDB Full | 2 | 125 | 20,000 | 5,000 | 25,000 |
| IMDB Mini | 2 | 125 | 2,000 | 5,000 | 25,000 |

Table 10: Summary of the underlying datasets from which the FTC-metadataset is constructed by (Arango et al., 2024).

## H   COMPUTATIONAL COSTS

The computational costs of applying JUCAL to an already trained ensemble of classifiers are negligible: While training the ensemble members costs hundreds of GPU-hours (Arango et al., 2024, Table 6), the computational costs of JUCAL are only hundreds of CPU-seconds (see Table 11).

Note that our actual implementation of JCUAL (Algorithm 2) is slightly more advanced than Algorithm 1. Instead of the naive grid search, we first optimize over a coarse grid and then optimize over a finer grid locally around the winner of the first grid search.

We want to emphasize that JUCAL is highly scalable and parallelizable. Since the computational costs are already below 13 CPU-minutes even for the largest datasets we considered (112,000 validation datapoints), we did not use parallelization to obtain the computational times in Table 11. However, for even larger calibration datasets or in settings where one does not want to wait for 13 minutes, it would be very straightforward to parallelize over multiple CPUs, or even over multiple distributed servers (across grid points), or to use GPU acceleration (vectorizing across validation data points).

For these reasons, the computational costs of JUCAL are practically negligible, if one already has access to an already trained ensemble. However, training (or fine-tuning) an ensemble can be computationally very expensive, but there are multiple techniques to reduce these costs (Kendall & Gal, 2017; Gal & Ghahramani, 2015; Wen et al., 2020; Havasi et al., 2021; Rossellini et al., 2024; Chan et al., 2025; Agarwal et al., 2025). In many practical settings, one has to train multiple models for hyperparameter optimization anyway. Then methods such as Greedy-5 can be used to obtain an ensemble from these different candidate models, as in our paper, basically for free.

| Dataset | Ensemble Method | Ensemble Selection Time (s) | Calibration Time (s) |
|---|---|---|---|
| DBpedia Full | JUCAL Greedy-50 | $17.6798 \pm 0.5566$ | $680.2392 \pm 9.9481$ |
| DBpedia Full | JUCAL Greedy-5 | $0.6779 \pm 0.5347$ | $92.5821 \pm 7.4349$ |
| DBpedia Mini | JUCAL Greedy-50 | $51.0481 \pm 5.3242$ | $764.0273 \pm 26.3293$ |
| DBpedia Mini | JUCAL Greedy-5 | $0.8412 \pm 0.0215$ | $99.8790 \pm 12.2445$ |
| News Full | JUCAL Greedy-50 | $8.8411 \pm 0.2699$ | $78.3229 \pm 0.8914$ |
| News Full | JUCAL Greedy-5 | $0.6653 \pm 0.5407$ | $11.2228 \pm 0.3444$ |
| News Mini | JUCAL Greedy-50 | $5.8553 \pm 0.0659$ | $78.2003 \pm 2.4816$ |
| News Mini | JUCAL Greedy-5 | $0.2189 \pm 0.0079$ | $8.3616 \pm 0.4244$ |
| SST-2 Full | JUCAL Greedy-50 | $4.1086 \pm 0.0714$ | $28.3639 \pm 0.3088$ |
| SST-2 Full | JUCAL Greedy-5 | $1.0158 \pm 1.9079$ | $5.2648 \pm 0.3565$ |
| SST-2 Mini | JUCAL Greedy-50 | $2.3958 \pm 0.0370$ | $21.9965 \pm 0.0998$ |
| SST-2 Mini | JUCAL Greedy-5 | $0.1430 \pm 0.0531$ | $3.7017 \pm 0.0385$ |
| SetFit Full | JUCAL Greedy-50 | $4.0146 \pm 0.2551$ | $211.6587 \pm 1.5352$ |
| SetFit Full | JUCAL Greedy-5 | $0.1287 \pm 0.0044$ | $26.0378 \pm 0.7412$ |
| SetFit Mini | JUCAL Greedy-50 | $14.0813 \pm 1.9794$ | $206.9967 \pm 12.2427$ |
| SetFit Mini | JUCAL Greedy-5 | $0.4324 \pm 0.2804$ | $20.2981 \pm 0.8983$ |
| Tweet Full | JUCAL Greedy-50 | $2.1564 \pm 0.0246$ | $16.4324 \pm 1.4845$ |
| Tweet Full | JUCAL Greedy-5 | $1.2017 \pm 2.4726$ | $3.7575 \pm 0.2718$ |
| Tweet Mini | JUCAL Greedy-50 | $1.5102 \pm 0.4339$ | $12.1769 \pm 0.1192$ |
| Tweet Mini | JUCAL Greedy-5 | $0.0996 \pm 0.0660$ | $3.0343 \pm 1.3491$ |
| IMDB Full | JUCAL Greedy-50 | $1.8614 \pm 0.2478$ | $11.6475 \pm 0.3522$ |
| IMDB Full | JUCAL Greedy-5 | $0.5458 \pm 1.1910$ | $2.4718 \pm 0.4158$ |
| IMDB Mini | JUCAL Greedy-50 | $1.3032 \pm 0.0351$ | $9.4108 \pm 0.5836$ |
| IMDB Mini | JUCAL Greedy-5 | $0.0827 \pm 0.0119$ | $1.9897 \pm 0.1802$ |

Table 11: Ensemble selection and calibration time (mean $\pm$ std in seconds) for JUCAL on Greedy-50 and Greedy-5 across all datasets (Full vs Mini).

While the training of models is a one-time investment, in some applications, reducing the prediction costs (i.e., forward passes through the model) for new test observations is more relevant. These costs are linear in the number of ensemble members $M$. The experiments of Arango et al. (2024) (which we reproduced) show clearly that Greedy-50 has a significantly better performance than Greedy-5, while being approximately 10 times more expensive (in terms of forward passes). However, applying JUCAL to Greedy-5 often results in even better performance than standard Greedy-50 (and sometimes even almost as good as applying JUCAL to Greedy-50). At the same time, Greedy-5 (JUCAL) requires approximately 10 times fewer forward passes than Greedy-50 (JUCAL). This makes Greedy-5 (JUCAL) a very powerful choice for real-time applications such as self-driving cars or robotics, where minimizing the number of forward passes is crucial for enabling efficient on-device inference on resource-constrained edge devices.

# I THEORY

## I.1 FINITE-SAMPLE CONFORMAL MARGINAL COVERAGE GUARANTEE

If a conformal marginal coverage guarantee under the exchangability assumption is desired, one can use conformal methods, such as APS, with an unseen exchangeable calibration dataset on top of JUCAL. Note that plain JUCAL does not require any new calibration dataset, as we have reused the validation dataset (already used for ensemble selection) as JUCAL's calibration dataset, which was already sufficient to outperform the baselines. However, for conformalizing JUCAL, a new unseen calibration dataset is required, as for any other conformal method.

### I.1.1 LIMITATIONS OF CONFORMAL MARGINAL COVERAGE GUARANTEES

The conformal theory heavily relies on the assumption of **exchangeability**. Exchangeability means that the joint distribution of calibration and test observations is invariant to permutations (e.g., i.i.d. observations satisfy this assumption).

While exchangeability is theoretically convenient, it is unrealistic in many real-world settings. Models are typically trained on past data and deployed in the future, where the distribution of $X_{new}$ usually shifts, i.e., $\mathbb{P}[X_{new}] \neq \mathbb{P}[X]$. Even if the conditional distribution $\mathbb{P}[Y_{new}|X_{new}] = \mathbb{P}[Y|X]$ remains fixed, such marginal shifts in $X_{new}$ can cause conformal methods to catastrophically fail to

provide valid marginal coverage. In situations such as Figure 3, **JUCAL** intuitively remains more robust, while standard (Conformal) Prediction that do not explicitly model epistemic uncertainty sufficiently well can fail more severely under distribution shifts in $X_{\text{new}}$. E.g., Figure 3, suggests $\mathbb{P}[Y_{\text{new}} \in C_{\text{APS-DE}}(X_{\text{new}}) \mid |X_{\text{new}}| < 7] \ll 99\% = 1 - \alpha$, as $C_{\text{APS-DE}}(X_{\text{new}}) = \{1\}$ would be a singleton in the situation of Figure 3, thus a marginal distribution shift of $X_{\text{new}}$ that strongly increases the probability of $|X_{\text{new}}| < 7$, would lead to a large drop of marginal coverage for $(X_{\text{new}}, Y_{\text{new}})$. JUCAL likewise lacks formal guarantees under extreme shifts, but good estimates of epistemic uncertainty should at least prevent you from being extremely overconfident in out-of-sample regions. Caution is required when trusting conformal guarantees, as the assumption of exchangeability is often not met in practice, and some conformal methods catastrophically fail for slight deviations from this assumption.

Even under the assumption of exchangeability, conformal guarantees have further weaknesses:

1. The conformal marginal coverage guarantee
$$\mathbb{P}[Y_{\text{new}} \in C(X_{\text{new}})] = \mathbb{E}_{\mathcal{D}_{\text{train}}, \mathcal{D}_{\text{cal}}}[\mathbb{P}[Y_{\text{new}} \in C(X_{\text{new}})|\mathcal{D}_{\text{train}}, \mathcal{D}_{\text{cal}}]] \geq 1 - \alpha$$
   does not imply that $\mathbb{P}[Y_{\text{new}} \in C(X_{\text{new}})|\mathcal{D}_{\text{train}}, \mathcal{D}_{\text{cal}}] \geq 1 - \alpha$ for a fixed realization of the calibration set $\mathcal{D}_{\text{cal}}$. If the calibration non-conformity scores are small by chance, conformal prediction sets may be too small (i.e., contain too few classes), especially with small calibration datasets. Reliable calibration is generally unattainable with small calibration datasets: Even if the exchangeability assumption is satisfied, even methods with conformal guarantees often strongly undercover, i.e., $\mathbb{P}_{\mathcal{D}_{\text{train}}, \mathcal{D}_{\text{cal}}}[\mathbb{P}[Y_{\text{new}} \in C_{\text{conformal}}(X_{\text{new}})|\mathcal{D}_{\text{train}}, \mathcal{D}_{\text{cal}}] \ll 1 - \alpha] \gg 0$.

2. Beyond marginal coverage, JUCAL is designed to improve *conditional calibration*: $\mathbb{P}[Y_{\text{new}} \in C(X_{\text{new}})|X_{\text{new}}] \approx 1 - \alpha$. This is crucial in human-in-the-loop settings, where interventions are prioritized based on an accurate *ranking* of predictive uncertainty across data points (see Appendix B). Marginal coverage guarantees offer no assurances for such rankings nor for conditional coverage. A method could have perfect marginal coverage but rank uncertainties arbitrarily. In other words, marginal coverage guarantees address only one specific metric (marginal coverage), while ignoring many other metrics that are often more important in practice.

To summarize, conformal marginal coverage guarantees say very little about the overall quality of an uncertainty quantification method. Conformal marginal coverage guarantees only shed light on a very specific aspect of uncertainty quantification and only under the quite unrealistic assumption of exchangeability.

## I.2    PROPERTIES OF THE NEGATIVE LOG-LIKELIHOOD

We define the $\text{NLL}(\mathcal{D}, \hat{p}) := \frac{1}{|\mathcal{D}|} \sum_{(x,y) \in \mathcal{D}} [-\log \hat{p}(y|x)]$ (where $y$ is the true class, and $\hat{p}(y|x)$ denotes the model's predicted probability mass for the true class $y$). The NLL is a standard and widely accepted metric, also known as the *log-loss* or *Cross-Entropy loss*.

We use the NLL for three different purposes in this paper:

1. Most classification methods use the NLL to train or fine-tune their models.
2. JUCAL minimize the NLL on the calibration dataset to determine $c_1^\star$ and $c_2^\star$.
3. We use the NLL as an evaluation metric on the test dataset $\mathcal{D}_{\text{test}}$.

### I.2.1    INTUITION BEHIND THE NEGATIVE LOG-LIKELIHOOD

Traditional classification metrics, such as accuracy or coverage, treat outcomes as binary (correct/incorrect or covered/not-covered). The NLL, however, offers a more nuanced evaluation by penalizing the magnitude of the model's confidence in its incorrect predictions. Specifically, since $-\log \hat{p}(y|x)$, the penalty for a misprediction is not simply a constant (as in 0/1 loss) but scales with the model's confidence in the true class $y$:

- **Severe Penalty for Overconfidence in Error:** The NLL applies a harsh penalty if the model assigns a very low probability $\hat{p}(y|x)$ to the true class $y$.

- **Incentive for Conditional Mass Accuracy:** This structure incentivizes the predicted distribution $\hat{p}(\cdot|x)$ to accurately reflect the conditional probability mass function $\mathbb{P}(Y|x)$.

This property simultaneously encourages good conditional calibration (i.e., that $\hat{p}$ closely approximates $\mathbb{P}$) and thus also encourages marginal calibration.

### I.2.2 THE NEGATIVE LOG-LIKELIHOOD MEASURES INPUT-CONDITIONAL CALIBRATION

The NLL is a *strictly proper scoring rule* for a predictive probability distribution $\hat{p}$ relative to the true conditional distribution $\mathbb{P}[Y|X]$ (Gneiting & Raftery, 2007). This means that the true conditional distribution $\mathbb{P}[Y|X]$ minimizes the expected NLL:

$$\mathbb{P}[Y|X] \in \arg\min_{\hat{p}} \mathbb{E}_{(X_{\text{new}}, Y_{\text{new}})} \left[ \text{NLL}\big(\{(X_{\text{new}}, Y_{\text{new}})\}, \hat{p}\big) \right]. \tag{17}$$

The expected NLL is minimized uniquely (a.s.) when $\hat{p}(y|x) = \mathbb{P}(Y = y|x)$. Any deviation from the true conditional distribution is penalized. In practice, evaluating the NLL on a finite dataset $\mathcal{D}$ provides a Monte-Carlo estimate of the expected NLL in (17).

Furthermore, unlike evaluating methods based on achieving marginal coverage and then minimizing a secondary metric like *Mean Set Size (MSS)*, the NLL is not susceptible to incentivizing deviations from conditional calibration. While MSS can prefer models that over-cover low-uncertainty regions and under-cover high-uncertainty ones (to reduce average size under a marginal coverage constraint), the NLL is minimized exclusively when the model reports the true conditional distribution $\mathbb{P}[Y|X]$, thereby naturally prioritizing conditional calibration. For more intuition, see Example I.1.

*Example* I.1 (Classification with Unbalanced Groups). Let the set of classes be $Y \in \{0, 1, \dots, 99\} =: \mathcal{Y}$. Let the input be $X \in \{1, 2\}$, with the low-uncertainty group being much more common: $\mathbb{P}[X = 1] = 0.8$ and $\mathbb{P}[X = 2] = 0.2$.

The true conditional probabilities $\mathbb{P}[Y|X]$ are:

- $X = 1$ (Low Uncertainty): $\mathbb{P}(Y = 0|X = 1) = 0.9$, $\mathbb{P}(Y = 1|X = 1) = 0.1$, and $\mathbb{P}(Y = k|X = 1) = 0$ for $k > 1$.

- $X = 2$ (High Uncertainty): $\mathbb{P}(Y = k|X = 2) = 0.02$ for $k = 0, \dots, 44$ (i.e., the first 45 classes together cover 90%), and $\mathbb{P}(Y = k|X = 2) = \frac{0.1}{55} \approx 0.0018$ for $k = 45, \dots, 99$.

Let $\hat{p}_{\text{true}}(y|x) = \mathbb{P}[Y = y|X = x]$. For a target coverage of $1 - \alpha = 0.9$, the *conditionally calibrated* method (which reports the smallest sets $C(x)$ based on $\hat{p}_{\text{true}}$ such that $\mathbb{P}[Y \in C(x)|X = x] \geq 0.9$) would produce:

- When $X = 1$: $C(1) = \{0\}$ (Set Size=1, Coverage=0.9)

- When $X = 2$: $C(2) = \{0, \dots, 44\}$ (Set Size=45, Coverage=0.9)

The marginal coverage is $\mathbb{P}[\text{covered}] = 0.8 \times 0.9 + 0.2 \times 0.9 = 0.9$. The Mean Set Size (MSS) is $\mathbb{E}[\text{size}] = 0.8 \times 1 + 0.2 \times 45 = 0.8 + 9.0 = 9.8$. The expected NLL of the true model is

$$\mathbb{E}[\text{NLL}(\hat{p}_{\text{true}})] = 0.8 \times (-0.9 \ln 0.9 - 0.1 \ln 0.1) + 0.2 \times (-0.9 \ln 0.02 - 0.1 \ln(0.1/55)) \approx 1.09.$$

Now, consider an *alternative method* that sacrifices conditional calibration to minimize MSS, while ensuring the *marginal* coverage is still exactly $0.9$. This method could report:

- When $X = 1$: $C'(1) = \{0, 1\}$ (Set Size=2, Coverage=1.0)

- When $X = 2$: $C'(2) = \{0, \dots, 24\}$ (Set Size=25, Coverage=0.5)

The marginal coverage of this method is $\mathbb{P}[\text{covered}] = 0.8 \times 1.0 + 0.2 \times 0.5 = 0.9$. The Mean Set Size (MSS) is $\mathbb{E}[\text{size}] = 1.6 + 5.0 = 6.6$.

Since $6.6 < 9.8$, this second method is strongly preferred by the (Marginal Coverage, MSS) metric. It achieves this by over-covering the common group ($X = 1$) and severely under-covering the rare, high-uncertainty group ($X = 2$). This alternative sets $C'$ can be obtained from a model that reports *untruthful* predicted probabilities, $\hat{p}'(y|x)$. For example, such a model might report:

- $\hat{p}'(0|X=1) = 0.5$, $\hat{p}'(1|X=1) = 0.4$, $\hat{p}'(2|X=1) = 0.1$. (This is *under-confident*).

- $\hat{p}'(k|X=2) = \frac{0.9}{25} = 0.036$ for $k = 0, \ldots, 24$, and $\hat{p}'(k|X=2) = \frac{0.1}{75} \approx 0.00133$ for $k = 25, \ldots, 99$. (This is *wildly over-confident* on the first 25 classes).

Note that this untruthful $\hat{p}'$ has the same top-1 accuracy as the true conditional probabilities, and yields predictive sets $C'(x) = \arg\min_{S \subseteq 2^{\mathcal{Y}} : \sum_{y \in S} \hat{p}'(y|X=x) \geq 0.9} |S|$ with the same marginal coverage and smaller average set size than $C(x) = \arg\min_{S \subseteq 2^{\mathcal{Y}} : \mathbb{P}[Y \in S|X=x] \geq 0.9} |S|$. However, the NLL, being a strictly proper scoring rule, is minimized *only* by the true distribution $\mathbb{P}[Y|X]$ (Gneiting & Raftery, 2007). This untruthful $\hat{p}'$ would incur a very high NLL

$$\mathbb{E}[\mathrm{NLL}(\hat{p}')] = 0.8 \times (-0.9 \ln 0.5 - 0.1 \ln 0.4) + 0.2 \times (-0.5 \ln 0.036 - 0.5 \ln(0.1/75)) \approx 1.57,$$

as it severely deviates from the true distribution. Since $1.57 \gg 1.09$, the NLL metric correctly and heavily penalizes the untruthful model $\hat{p}'$ that enables this failure of conditional calibration. This demonstrates that, unlike marginal metrics, the NLL inherently aligns the optimization objective with conditional calibration.

### I.2.3 THE NLL INCENTIVIZES TRUTHFULNESS EVEN UNDER INCOMPLETE INFORMATION (FROM A BAYESIAN POINT OF VIEW)

As a strictly proper scoring rule, the NLL is guaranteed to incentivize reporting the true distribution when the true distribution is known (Gneiting & Raftery, 2007; Buchweitz et al., 2025). However, Buchweitz et al. (2025) emphasize that even strictly proper scoring rules can asymmetrically penalize deviations from the truth when the true distribution is unknown, which might induces biases. When training data $\mathcal{D}_{\mathrm{train}}$ is finite and model parameters $\theta$ are unknown, one's belief over possible parameters can be expressed via a posterior $\mathbb{P}[\theta \mid \mathcal{D}_{\mathrm{train}}, \pi]$ in a Bayesian framework. The corresponding *posterior predictive distribution*

$$\mathbb{P}[Y_{\mathrm{new}} \mid X_{\mathrm{new}}, \mathcal{D}_{\mathrm{train}}, \pi] = \mathbb{E}\left[\mathbb{P}[Y_{\mathrm{new}} \mid X_{\mathrm{new}}, \theta] \mid \mathcal{D}_{\mathrm{train}}, \pi\right]$$

captures total predictive uncertainty, integrating both aleatoric uncertainty $\mathbb{P}[Y_{\mathrm{new}} \mid X_{\mathrm{new}}, \theta]$ (inherent noise) and epistemic uncertainty $\mathbb{P}[\theta \mid \mathcal{D}_{\mathrm{train}}, \pi]$ (parameter uncertainty).

From a Bayesian perspective, the posterior predictive distribution uniquely minimizes the expected NLL:

$$\mathbb{E}\left[\mathrm{NLL}\big(\{(X_{\mathrm{new}}, Y_{\mathrm{new}})\}, \hat{p}\big) \mid \mathcal{D}_{\mathrm{train}}, \pi\right].$$

Minimizing the NLL thus leads to a model that incorporates total predictive uncertainty. Averaging over the posterior increases predictive entropy relative to the expected entropy under the parameter posterior, i.e.,

$$H\left[\mathbb{E}\big[\mathbb{P}[Y_{\mathrm{new}} \mid X_{\mathrm{new}}, \theta] \mid \mathcal{D}_{\mathrm{train}}, \pi\big]\right] > \mathbb{E}\left[H\big[\mathbb{P}[Y_{\mathrm{new}} \mid X_{\mathrm{new}}, \theta]\big] \mid \mathcal{D}_{\mathrm{train}}, \pi\right].$$

This inequality expresses that the NLL-optimal predictor—the posterior predictive distribution—has higher entropy (more uncertainty) than the expected entropy. One might view this as a bias towards overestimating uncertainty, yet this "bias" precisely encodes epistemic uncertainty: when the true distribution is unknown, the predictive distribution must honestly represent uncertainty over parameters, resulting in a higher-entropy, more uncertain prediction. Thus, minimizing the NLL naturally yields a model that accounts for both aleatoric and epistemic uncertainty.

Therefore, the NLL serves as a principled scoring rule for evaluating models such as JUCAL, which explicitly aim to represent total predictive uncertainty and thereby achieve improved input-conditional calibration. This justifies its use both in the calibration step and as an evaluation metric on the unseen test dataset $\mathcal{D}_{\mathrm{test}}$.

### I.3 THEORETICAL JUSTIFICATION OF DEEP ENSEMBLE

Our method builds on the deep ensemble (DE) framework; hence, it draws on similar theoretical justifications. Empirically, DEs have been shown to reduce predictive variance while maintaining low bias, as demonstrated by (Lakshminarayanan et al., 2017). Even without sub-sampling or bootstrapping, this idea is similar to bagging, for which Bühlmann & Yu (2002) provided a theoretical justification.

Moreover, DEs can also be mathematically justified: since NLL is a strictly convex function, Jensen's inequality implies that the NLL of a DE is always as good or better than the average NLL of individual ensemble members, i.e.,

$$\text{NLL}(\bar{p}, y_i) = -\log\left(\frac{1}{M}\sum_{m=1}^{M} p(y_i \mid \boldsymbol{x}_i, \theta_m)\right) \leq \frac{1}{M}\sum_{m=1}^{M} \left[-\log p(y_i \mid \boldsymbol{x}_i, \theta_m)\right]$$

where $p_m = \text{Softmax}(f_m)$. Overall, there are many intuitive, theoretical, and empirical justifications for DEs.

### I.4 INDEPENDENCE OF JUCAL TO THE CHOICE OF RIGHT-INVERSE OF SOFTMAX

In this subsection, we rigorously demonstrate that the calibrated probabilities produced by JUCAL are invariant to the specific choice of a right-inverse $\text{Softmax}^{-1}$ of the Softmax function. This property is crucial when JUCAL is applied to models where only the predictive probabilities $p$ are accessible (e.g., tree-based models), requiring the reconstruction of logits.

**Non-uniqueness of Inverse Softmax.** Because Softmax is invariant to translation by a scalar vector, it is not injective and therefore does not possess a unique two-sided inverse. Instead, it admits a class of right-inverses. Specifically, for any logit vector $\mathbf{z} \in \mathbb{R}^K$ and scalar $k \in \mathbb{R}$, $\text{Softmax}(\mathbf{z} + k\mathbf{1}) = \text{Softmax}(\mathbf{z})$, where $\mathbf{1} \in \mathbb{R}^K$ denotes the vector of all ones. Consequently, the set of all valid logit vectors consistent with a probability vector $p$ is given by:

$$\mathcal{Z}(p) = \{\log(p) + C\mathbf{1} \mid C \in \mathbb{R}\}, \tag{18}$$

where $\log$ is applied element-wise. When recovering logits from probabilities, one must select a specific representative from $\mathcal{Z}(p)$ (i.e., choose a specific right-inverse), typically by imposing a constraint such as $\sum z_k = 0$ or by simply setting $C = 0$. For example, in our implementation, we use $C = 0$, i.e., we define $\text{Softmax}^{-1}(p) = \log(p)$. In the remainder of this subsection, we prove that any other choice of right-inverse would result in exactly the same predictive distributions when applying JUCAL.

**Proof of Invariance.** Let $f_m$ be any logits corresponding to probabilities $p_m$. Consider an arbitrary alternative choice of a right-inverse for Softmax where each member's logit vector is shifted by a scalar constant $k_m \in \mathbb{R}$. The shifted logits are $\widetilde{f_m} = f_m + k_m\mathbf{1}$.[17]

First, we consider the effect on the temperature-scaled logits. The shifted temperature-scaled logits $\widetilde{f_m^{\text{TS}(c_1)}}$ are:

$$\widetilde{f_m^{\text{TS}(c_1)}} = \frac{\widetilde{f_m}}{c_1} = \frac{f_m + k_m\mathbf{1}}{c_1} = f_m^{\text{TS}(c_1)} + \frac{k_m}{c_1}\mathbf{1}. \tag{19}$$

Next, we calculate the shifted ensemble mean of the temperature-scaled logits, $\widetilde{\bar{f}^{\text{TS}(c_1)}}$:

$$\widetilde{\bar{f}^{\text{TS}(c_1)}} = \frac{1}{M}\sum_{j=1}^{M} \widetilde{f^{\text{TS}(c_1)}}_j = \frac{1}{M}\sum_{j=1}^{M}\left(f_j^{\text{TS}(c_1)} + \frac{k_j}{c_1}\mathbf{1}\right) = \bar{f}^{\text{TS}(c_1)} + \frac{\bar{k}}{c_1}\mathbf{1}, \tag{20}$$

where $\bar{k} = \frac{1}{M}\sum_{j=1}^{M} k_j$ is the average shift.

---

[17]Note that this proof also works if $k_m$ depends on $x$ and even if one would use different right-inverses for different ensemble members.

Substituting these into the JUCAL transformation definition, we obtain the shifted calibrated logits $\widetilde{f_m^{\text{JUCAL}(c_1,c_2)}}$:

$$
\begin{aligned}
\widetilde{f_m^{\text{JUCAL}(c_1,c_2)}} &= (1-c_2)\widetilde{\bar{f}^{\text{TS}(c_1)}} + c_2\widetilde{f_m^{\text{TS}(c_1)}} \\
&= (1-c_2)\left(\bar{f}^{\text{TS}(c_1)} + \frac{\bar{k}}{c_1}\mathbf{1}\right) + c_2\left(f_m^{\text{TS}(c_1)} + \frac{k_m}{c_1}\mathbf{1}\right) \\
&= \left((1-c_2)\bar{f}^{\text{TS}(c_1)} + c_2 f_m^{\text{TS}(c_1)}\right) + \left((1-c_2)\frac{\bar{k}}{c_1} + c_2\frac{k_m}{c_1}\right)\mathbf{1} \\
&= f_m^{\text{JUCAL}(c_1,c_2)} + \gamma_m\mathbf{1},
\end{aligned}
$$

where $\gamma_m = \frac{1}{c_1}((1-c_2)\bar{k} + c_2 k_m)$ is a scalar quantity specific to member $m$.

Since the Softmax function is shift-invariant, $\text{Softmax}(\widetilde{f_m^{\text{JUCAL}(c_1,c_2)}}) = \text{Softmax}(f_m^{\text{JUCAL}(c_1,c_2)} + \gamma_m\mathbf{1}) = \text{Softmax}(f_m^{\text{JUCAL}(c_1,c_2)})$. Consequently, $\widetilde{p_m^{\text{JUCAL}(c_1,c_2)}} = p_m^{\text{JUCAL}(c_1,c_2)}$, and the final calibrated predictive distribution $\bar{p}^{\text{JUCAL}(c_1,c_2)}$ remains identical regardless of the arbitrary constants $k_m$ chosen during the inverse operation. This proves that JUCAL is well-defined for probability-only models, i.e., models (such as decision trees, random forests, or XGBoost) that directly output probabilities instead of logits.

