# OpenReview forum: "JUCAL: Jointly Calibrating Aleatoric and Epistemic Uncertainty in Classification Tasks"
_ICLR.cc/2026/Conference — Submitted to ICLR 2026_

### Official Review · Reviewer_iK9t · 2025-10-28

**Soundness:** 2
**Presentation:** 2
**Contribution:** 2
**Rating:** 4
**Confidence:** 4

**Summary:**

This paper presents a calibration technique *JUCAL* that considers both aleatoric uncertainty and epistemic uncertainty, i.e., the method jointly calibrates two constants to weight and scale epistemic and aleatoric uncertainties by optimizing the negative log-likelihood on the validation/calibration dataset. Experiments with fine-tuned large language models (LLMs) on six text classification tasks demonstrate the improvement of the proposed method.

**Strengths:**

1. The paper is relatively well-structured, and the motivation is clearly stated.

2. The initiation behind the method design is well described, together with detailed background knowledge and introductions in the appendix.

3. The proposed method shows improvements over the baselines.

4. The proposed method is a post-processing technique, can potentially be implemented on various ensemble models and classification tasks.

**Weaknesses:**

1. The proposed method lacks theoretical justification (limited theory).

2. JUCAL considers both aleatoric and epistemic components during calibration. But, it is not clear how it would actually influence the aleatoric, epistemic, and total (predictive) uncertainty in terms of the widely used entropy measures (i.e, the equations 5-7 in the Appendix). It would be better to include more theoretical analysis in this context, as well as the empirical evaluations. For example, how would the deep ensembles perform on out-of-distribution tasks, using the original measures vs. the calibrated ones?

3. Regarding the evaluation metrics, it would be beneficial to consider the expected calibration error (ECE) and also highlight the prediction performance, e.g, the accuracy.

4. Regarding the experiment designs, while the method itself is introduced for general classification models, its applicability on other classification tasks, like image classification e.g., [1] remains underexamined. Similarly, in the abstract, it claims that *any trained ensemble of classifiers (e.g., transformers, CNNs, or tree-based methods)*. It is not fully empirically evaluated. The baselines are limited.

[1] Roschewitz, Mélanie, et al. "Where are we with calibration under dataset shift in image classification?

**Questions:**

1. In lines 230-231, could the authors please indicate whether the inverse of the softmax is actually implemented or just using the logit?

2. I think in most cases, the softmax of the averaged logits is not equal to the averaged softmax of individual logits. How would this point affect your method design?

3. Minor: The readability of Figure 3 is a bit low; maybe consider increasing the font size, etc. It would be better to include the cost of the calibration in the main body. *is never significantly outperformed by any of the baselines on any evaluated metric.* (line 467) sounds like a really strong claim.

4. Would the authors discuss some limitations and future work in the main body?

---

> ### Author Response · Authors · 2025-11-21
>
> Thank you very much for the detailed feedback.
>
> **Regarding the raised weaknesses:**
>
> @1. What kind of theoretical result are you interested in? We can add some results for the marginal coverage guarantee for a conformal version of JUCAL. Or we could add a result on asymptotic consistency. Or a result on JUCAL being asymptotically always better or equal than no-calibration, pool-then-cal and cal-then-pool. Please let us know what is unclear to you from a theoretical point of view, so that we can add clarification.
>
> @2. Would you be interested in a proof that increasing c_1, ceteris paribus, increases the aleatoric uncertainty? Or in a proof that increasing c_2,  ceteris paribus, increases the epistemic uncertainty? We can also add some plot where we show epistemic and aleatoric uncertainty for different values of c_1 and c_2 to provide more intuition.
>
> @3. We can add ECE and accuracy. In this case, we will also add a paragraph in the appendix discussing the limitations of ECE and accuracy, which are not strictly proper scoring rules.
>
> @4. Yes, adding image datasets and CNNs to our evaluation would be interesting, while right now we mainly focus on LLMs. Our experimental design massively reduced the CO2 footprint of our paper, as we did not train any models (except the toy models for the plots). Instead, we used the already trained ensemble of LLMs fine-tuned for more than 3k GPU-hours by the authors of a SOTA paper. This allows us to fully isolate our post-processing step with a low carbon footprint. If you want, we can add some experiments for CNNs on MNIST.
> Do you know if [1] Roschewitz et al offers a meta dataset too? In other words, are the predictions (e.g., logits) of their already trained ensemble members and the true labels publicly available such that everyone can benchmark their post-processing methods on their already trained models without having to retrain all these models on their own? If that is the case, we would be more than happy to evaluate JUCAl on their metadataset.
>
> **Regarding the raised questions:**
>
> @1. For neural networks, JUCAL can directly operate on the logit outputs. For models that output normalized probabilities (e.g., certain tree-based methods), we apply the inverse softmax to recover logits before applying JUCAL. We will clarify this in the revised manuscript.
>
>
> @2. We fully agree. The two are generally not equal, and this non-equivalence is central to our approach: logit diversification would have no effect if they were identical. We will make this motivation more explicit in the revision.
>
> @3. We will improve the readability of Figure 3. We fully agree that our empirical results summarized in line 467 are extraordinarily strong, as you can check in Tables 2-9 in Appendix F. To be more precise, we can write that JUCAL Greedy-50 was never significantly outperformed in any metric by any considered non-JUCAL method, as you can check in Tables 2-9 in Appendix F, where we define "significantly outperformed” via a paired test with alpha=5%”. In other words, across 192 (8 tables, 4 competitors, 6 datasets) comparisons of JUCAL Greedy-50 with other methods, there was not a single comparison where JUCAL Greedy-50 was statistically significantly outperformed. This demonstrated the outstanding robustness and reliability of JUCAL. We will formulate this more precisely in the revision.
> Furthermore, JUCAL Greedy-5 is, to the best of our knowledge*, never statistically significantly outperformed by any other considered competitor of ensemble size 5 in any of the metrics.
>
> *For DBPedia JUCAL Greedy-5, in certain metrics, all 4 displayed digits after the decimal sometimes agree with other Greedy-5 competitors. This creates situations where it would mathematically be possible that JUCAL Greedy-5 could be statistically significantly outperformed by a tiny, practically irrelevant margin, which is not reflected in the tables.
>
>
> @4. We agree that including this discussion would strengthen the paper. We will add a concise section in the main text highlighting the papers’ limitations and future research: Our experiments are limited to text classification. Empirical evaluation on other data modalities, such as, for example, tabular data, image data, graph data, video data, other time series data, and multi-modal data, would be interesting and important future work. As future work on a methodological level, we want to extend JUCAL to LLM-based question answering by combining it with ideas from semantic entropy.

---

### Official Review · Reviewer_GYZz · 2025-10-31

**Soundness:** 3
**Presentation:** 2
**Contribution:** 2
**Rating:** 2
**Confidence:** 4

**Summary:**

The authors propose JUCAL, a post-hoc calibration method for ensembles of classifiers that aims to jointly calibrate aleatoric and epistemic uncertainty . The method introduces two calibration constants, `c1` and `c2`, which are optimized by minimizing the negative log-likelihood (NLL) on a validation set. `c1` acts as a temperature parameter on the logits of individual ensemble members to calibrate aleatoric uncertainty, while `c2` scales the diversity of the logits across the ensemble to calibrate epistemic uncertainty . The authors claim this simple approach leads to significant improvements in NLL, predictive set size, and other uncertainty-related metrics, even allowing smaller ensembles to outperform much larger ones .

**Strengths:**

*   The paper addresses the important and underexplored problem of balancing aleatoric and epistemic components during uncertainty calibration. Standard methods like temperature scaling treat predictive uncertainty as a single quantity, and the idea of disentangling these sources for more granular control is well-motivated.
*   The proposed method, JUCAL, is simple, intuitive, and computationally inexpensive.
*   The experimental results on several text classification tasks are promising.

**Weaknesses:**

My main concerns are regarding the strength of the paper's core claims, the limited experimental scope, and the lack of crucial baselines.

*   **Overstated Disentanglement Claims:** The central claim of "jointly calibrating aleatoric and epistemic uncertainty" seems overstated and is not supported by sufficient experimental evidence. The primary evidence for successful disentanglement is presented in Figure 5, which shows that the estimated epistemic uncertainty (EU) decreases with more training data while aleatoric uncertainty (AU) does not . While this is a necessary sanity check, it contradicts recent literature suggests this separation is non-trivial. A recent benchmark (Mucsányi et al., NeurIPS 2024) shows a high correlation between common AU and EU estimators, which questions the practical benefit of calibrating them with separate parameters. The authors' own discussion acknowledges critiques of the mutual information-based decomposition they use (see Appendix A.2.1), yet the main paper proceeds as if the decomposition is clean. A more thorough analysis is needed to substantiate the claim that the observed performance gains stem from successful joint calibration rather than simply being a more flexible (two-parameter) fitting of the final predictive distribution.
In particular, the authors should compare against alternative estimators for AU and EU derived from ensembles, such as those based on the Bregman decomposition or simple variance of the logits. Furthermore, comparing the AU/EU estimates from JUCAL to those from dedicated single-model methods like Evidential Deep Learning (EDL) or Deep Deterministic Uncertainty (DDU) would provide a much stronger test of whether JUCAL is truly capturing distinct uncertainty components.

*   **Missing Crucial Baselines:** The experimental evaluation is narrow, comparing only against "no calibration" and "pool-then-calibrate". The authors seem to have missed some crucial baselines that would be necessary to properly contextualize their contribution. For instance, in Section 4.1, they introduce "calibrate-then-pool" as a method that primarily adjusts aleatoric uncertainty. A comparison to this baseline is missing and would be essential for an ablation study to understand the individual contributions of `c1` and `c2`.

*   **Limited Experimental Scope:** The experiments are confined to text classification tasks using ensembles of fine-tuned LLMs . To support the claim that JUCAL is a "new go-to method for calibrating ensembles in classification" , the evaluation must be extended to other domains, especially computer vision, and other model families like CNNs and ViTs. The dynamics of aleatoric and epistemic uncertainty can vary significantly across different data modalities and model architectures, and demonstrating robustness in these diverse settings is critical.

*   **Presentation of Results:** The presentation could be improved. The bar charts in Figure 4 are cluttered and hard to read, making it difficult to assess the relative performance improvements, especially given the small error bars . More effective visualizations, such as plots of relative improvement or tables with statistical significance tests, would improve the clarity of the results.

**Questions:**

See above

---

> ### Author Response · Authors · 2025-11-21
> **answer part 1**
>
> We thank the reviewer for the detailed and constructive feedback.
>
> **Regarding the Disentanglement of uncertainties:**
>
> We think there was the following misunderstanding: The main goal of our paper is to improve predictive distributions. We use epistemic and aleatoric uncertainty as a tool to improve the predictive distributions with respect to the strictly proper scoring rule NLL. In Appendix I.2.2, we explain why the NLL measures input-conditional calibration. Additionally, we improve other metrics of the predictive distribution, such as set size, AUROC, and AURAC. This is our main contribution. This is an important contribution since many practitioners care about the performance of the predictive distribution. The NLL is an important and widely used metric.
> The main goal of our paper is **not** uncertainty disentanglement, but the improvement of the predictive distribution (using epistemic and aleatoric uncertainties as tools to do so). As a side result, we show in Figure 5 the different contributions of epistemic and aleatoric uncertainty.
>
> We agree that our results in Figure 5 are surprising from the perspective of (Mucsányi et al., NeurIPS 2024). However, surprising new discoveries are not a bad thing in science. We fully agree that this opens the door for much future work to be done to better understand how both these experiments (ours and the ones of Mucsányi et al.) can be true simultaneously. We fully agree that there are multiple possible explanations at present, and future research is needed to narrow them down. Do you suggest that we should mention the tension between our results and the ones of Mucsányi et al. in the main paper instead of only discussing it in the appendix? And maybe in the appendix, we should dive even deeper into this discussion?
> Note that in Line 456 we mention already “More research is needed”.
>
> **Regarding “The authors' own discussion acknowledges critiques of the mutual information-based decomposition they use (see Appendix A.2.1), yet the main paper proceeds as if the decomposition is clean”**
>
> Note that in Line 455 we mention already “We used Equations (6) and (7) from Appendix A.2.1 to compute the values presented in Figure 5, while there would be other alternatives too.” This sentence mentions explicitly that there are other alternatives and links explicitly to Appendix A.2.1.
> If you want, we can mention already in the main paper that the choice of alternatives is considered quite controversial in the literature.
>
> **Regarding “In particular, the authors should compare against alternative estimators for AU and EU derived from ensembles, such as those based on the Bregman decomposition or simple variance of the logits.”**
>
> Thanks for the suggestion! According to our intuition, variance(logits)/mean(logits-min(logits))² captures epistemic uncertainty, maybe kind of well. Is there any popular disentanglement which is defined like this, or should we define it like this ourselves? Maybe this would help to better explain the intuition behind JUCAL, and it would have an effect on Figure 5.
> We want to emphasize that the JUCAL algorithm is completely independent of how one defines epistemic and aleatoric uncertainty. So these different definitions of epistemic and aleatoric would not have any effect on the main results.
>
>
>
> **Regarding the crucial baselines:**
>
> The non-calibrated baseline is the SOTA Greedy ensemble algorithm introduced by Arango et al. (2024). While this method is not itself a calibration approach, it produces highly diverse ensembles and was shown to outperform all other ensemble construction strategies evaluated in that work. We apply SOTA pool-then-calibrate on top of this strong baseline.
> This motivated our decision to exclude calibrate-then-pool from the initial set of baselines. Nevertheless, we agree that including calibrate-then-pool would make the comparison more complete, and we will incorporate this baseline in the revised version.
>
> We think the complaint that we “only” experiment with LLMs and not with vision models might be slightly biased. There are many UQ papers that only experiment with vision models and not with LLMs. Do you have the same complaint for them as well? LLMs are actually quite an important topic (vision models obviously as well). Our experimental design massively reduced the CO2 footprint of our paper, as we did not train any models (except the toy models for the plots). Instead, we used the already trained ensemble of LLMs fine-tuned for more than 3k GPU-hours by the authors of a SOTA paper. This allows us to fully isolate our post-processing step with a low carbon footprint. If you want, we can add some experiments for CNNs on MNIST. Are you aware of any other benchmark where the predictions of an already trained ensemble are publicly available, such that postprocessing methods can be compared against each other? We would be very happy to evaluate our postprocessing method on more such benchmarks.

---

> ### Author Response · Authors · 2025-11-21
> **answer part 2**
>
> **Regarding the presentation of results**
>
> We chose to visualize the results primarily through charts to provide an intuitive and easily digestible comparison across methods, without requiring the reader to scan detailed numerical values. However, we agree that for readers interested in finer-grained comparisons, numerical tables offer additional clarity and precision. For this reason, the comprehensive tables are already in our Appendix F “Tables and Figures”. Please let us know if Tables 2-9 in Appendix F on pages 35-37 are exactly what you were looking for. We shade the results that are not significantly outperformed by the best-performing method. I.e., every method that is not shaded was statistically significantly outperformed by the best performing mode (paired test, α = 5%).

---

### Official Review · Reviewer_wMgH · 2025-10-31

**Soundness:** 3
**Presentation:** 2
**Contribution:** 2
**Rating:** 4
**Confidence:** 4

**Summary:**

The paper proposes post-hoc calibration method for deep ensembles that jointly calibrates aleatoric and epistemic uncertainty using two scaling parameters c1 and c2. The method aims to address the limitation of temperature scaling and conformal methods, which do not balance the epistemic and aleatoric uncertainty. Concretely, JUCAL (1) applies temperature scaling (via c1) to each ensemble member (calibrate-then-pool style) to affect aleatoric uncertainty, and (2) linearly rescales each temperature-scaled member toward/away from the ensemble mean using c2 to adjust ensemble diversity (epistemic uncertainty) without changing the ensemble mean. JUCAL is evaluated on text classification tasks using pre-trained LLM ensembles, demonstrating improvements in NLL and prediction set size.

**Strengths:**

- The paper is generally well written
- JUCAL is conceptually simple, easy to implement, and can be applied to off-the-shelf ensembles without model re-training.
- The experiments compare across ensemble sizes and demonstrate consistent improvements on NLL and predictive set size.

**Weaknesses:**

**Lack of ablation studies on core claims**

The paper's main idea is that c1 calibrates aleatoric uncertainty while c₂ calibrates epistemic uncertainty. However, no ablation studies are provided to validate this claimed separation. Without these ablations, the interpretation of c1 and c2 as targeting specific uncertainty types remains speculative.

**Lack of support for main claims**

The conclusion states: “our approach provides a principled and impactful advance in uncertainty calibration for deep learning.“ However, no calibration metrics are reported. The paper evaluates NLL, AORAC, AOROC, and prediction set size, but omits standard calibration metrics (e.g. ECE, MCE, Brier score, class-wise calibration analysis, reliability diagrams). The absence of these metrics is especially evident given that the paper explicitly positions itself as advancing "uncertainty calibration." Without these, it's unclear if JUCAL actually improves calibration.
Moreover, describing the approach as 'principled' suggests derivation from theoretical foundations or formal guarantees. However, the paper provides neither a rigorous connection to established calibration theory nor mathematical justification for why the specific parameterization in Equation 2 should successfully decompose epistemic and aleatoric uncertainty.


**Limited experimental depth**

Gaps include missing ablation studies (e.g., comparing JUCAL against c1-only optimization, which is equivalent to the calibrate-then-pool approach mentioned in Section 4.1), omission of standard calibration metrics (see points above), and absence of key baselines—notably, the paper claims improvements in prediction set size yet fails to compare with conformal prediction methods designed specifically for this task with formal coverage guarantees. The evaluation is limited to pre-trained transformer ensembles on NLP tasks.

**Limitations of the method are not discussed**

For example, optimizing two parameters (c1, c2) via grid search requires sufficient calibration data to avoid overfitting, yet the paper provides no analysis of minimum sample size requirements.

**Questions:**

How does JUCAL relate to proper scoring rules or calibration error metrics (ECE, MCE)?

---

> ### Author Response · Authors · 2025-11-21
>
> We thank the reviewer for the detailed and constructive feedback.
>
> **Regarding scoring rules:**
>
> We note that Negative Log-Likelihood (NLL) is a proper scoring rule [Gneiting 2007]. In Appendix I.2.2, we explain how the NLL measures input-conditional calibration. Nonetheless, we will additionally report Expected Calibration Error (ECE) and Maximum Calibration Error (MCE) to provide a more comprehensive assessment of calibration. In this case, we will also add a paragraph in the appendix discussing the limitations of ECE and MCE, which are not strictly proper scoring rules. Or which metric would you consider most important?
>
> **Regarding additional experiments:**
>
>  We will include further empirical analysis, specifically (i) ablation studies to isolate the contribution of each component, and (ii) conformal prediction experiments to complement our evaluation of uncertainty estimates.
> @(i): What type of ablation are you referring to? The goal of this experiment would be to see whether c_1 mainly adjusts the aleatoric uncertainty and c_2 mainly adjusts the epistemic uncertainty?
>
> **Regarding the comment on the theoretical connection to JUCAL’s parametrization:**
>
> You wrote: “However, the paper provides neither a rigorous connection to established calibration theory nor mathematical justification for why the specific parameterization in Equation 2 should successfully decompose epistemic and aleatoric uncertainty.”
>
> Our goal is not to rigorously decompose epistemic and aleatoric uncertainty. Instead, we propose a simple heuristic aimed at improving predictive uncertainty. Appendix I.2.2 explains why NLL measures input-conditional calibration, since NLL is a strictly proper scoring rule. Appendix A outlines several theoretical perspectives on the relationship, along with the important distinctions, between aleatoric and epistemic uncertainty. Moreover, Appendix D gives intuition on how JUCAL’s parameter choices help shape both aleatoric and epistemic uncertainty. We will revise the manuscript to make this intent explicit and to clarify how the scaling parameters influence the resulting uncertainty quantification.
> For clarity on the impact of the parameters c1 and c2, would a plot of how epistemic and aleatoric change for different values of c1 and c2 be helpful? If so, we could provide this type of plot. Is this type of plot the ablation studies that you mentioned in (i)?
>
> **Regarding “calibrate-then-pool”:**
>
> We thought that the literature hints already that pool-then-calibrate is superior, but we will also compare against calibrate-then-pool.
>
> **Regarding “compare with conformal prediction methods designed specifically for this task with formal coverage guarantees”:**
> For plain JUCAL, we don’t use any additional unseen calibration dataset. If we have access to an unseen calibration dataset, we can compare conformalized JUCAL G5 against conformalized G5, for example. We can add such experiments. As explained in Appendix I.1, conformalized JUCAL also has formal marginal coverage guarantees under the exchangeability assumption.
>
> **Regarding “limited to pre-trained transformer ensembles on NLP tasks”:**
> We can add MNIST too as another experiment, as our method is not limited to LLMs.
>
>
> **References.**
>
> [Gneiting 2007] Gneiting, T., & Raftery, A. E. (2007). Strictly proper scoring rules, prediction, and estimation. Journal of the American statistical Association, 102(477), 359-378.

---

### Official Review · Reviewer_J1qw · 2025-11-02

**Soundness:** 3
**Presentation:** 3
**Contribution:** 4
**Rating:** 8
**Confidence:** 3

**Summary:**

Most existing calibration methods treat all instances uniformly, without accounting for the underlying uncertainty associated with each prediction. However, instances can differ significantly, some may exhibit high aleatoric uncertainty, while others reflect high epistemic uncertainty . Applying the same calibration to all such cases can lead to overconfidence on out-of-distribution samples and underconfidence on in-distribution samples with high aleatoric noise.
The paper introduces Joint Uncertainty Calibration (JUCAL), a post-hoc method that uses uncertainty estimates from an ensemble model to calibrate the predicted class probabilities more effectively. Specifically, it scales the aleatoric and epistemic components of the ensemble’s predictive distribution, allowing the calibration to adapt to the nature of the uncertainty in each instance.
The method is evaluated on multiple text classification benchmarks, where it consistently outperforms some standard calibration techniques across four metrics.

**Strengths:**

The method applies temperature scaling in a new way, taking into account ensemble differences.
The method is intuitive, simple to understand and can be applied post-hoc to already trained ensembles. This makes the method easy to use.

**Weaknesses:**

The experiments could include other post-hoc calibration methods as a comparison. Also, calibrate-then-pool could be added to comparison, since it is the special case of JUCAL when $c_2 = 1$.

Section 5.1 briefly mentions that ensemble members are based on pretrained architectures like GPT-2, BERT, and T5, but it remains ambiguous whether the ensembles consist of a single architecture or multiple architectures. Since the nature of the ensemble affects both uncertainty decomposition and calibration behavior, this detail is important for reproducibility and for interpreting the results.

Paper's motivation is that traditional calibration methods tend to be overconfident on out-of-distribution samples and underconfident on in-distribution samples with high aleatoric uncertainty. This claim would be better supported by explicitly evaluating JUCAL under OOD settings for example, by testing on textual datasets from different distributions than the training data (e.g., evaluating on a News classification dataset when models were trained on DBpedia). Such experiments would strengthen the argument that JUCAL provides more robust calibration across varying uncertainty regimes.

**Questions:**

In Table 10, both DBpedia Full and SetFit Full are shown with 25 members. Could you clarify whether Greedy-50 in these settings still uses only 25 models?

If the ensemble consists of a mix of architectures (e.g., BERT, T5, GPT-2), could you elaborate on what types of models were selected by the greedy algorithm? It would be helpful to know whether the selection favored certain architectures or maintained diversity, as this might impact both calibration behavior and generalization.

If possible, I’d be interested in seeing how the test NLL varies with different values of $c_1$ and $c_2$. Does JUCAL’s performance degrade smoothly, or is it sensitive to the specific combination of these parameters? A sensitivity analysis or ablation could help clarify how stable the calibration effect is with respect to these parameters.

---

> ### Author Response · Authors · 2025-11-21
>
> We thank the reviewer for the detailed and constructive feedback!
>
> **Regarding additional baselines:**
>
> We appreciate the suggestion to include additional post-hoc calibration methods. Prior work has described pool-then-calibrate as generally superior to calibrate-then-pool, which led us to omit the latter from our initial set of baselines. Nonetheless, we agree that including calibrate-then-pool would provide a more complete comparison. We will add this baseline in the revised version.
>
> **Regarding the ensemble member architectures:**
>
> The reviewer is correct that Section 5.1 may leave ambiguity about whether ensembles mix architectures or use a single architecture per ensemble. We allow the ensemble to mix multiple different ensembles. This is in accordance with the PCS framework for veridical data science, which emphasizes the importance of considering the uncertainty arising from all judgment calls across the entire data science life cycle. Choosing a specific architecture is a subjective judgment call. If different architectures perform reasonably on the validation dataset but give vastly different predictions for a specific datapoint, then one should be uncertain about this datapoint instead of fully trusting one randomly selected architecture. Theoretically, it could happen that Greedy-M only selects models from one specific architecture if one architecture yields clearly superior performance on the validation dataset. For some large datasets, such as DBpedia-full and Set-Fit-full, only GPT2 was trained. Do you want us to clarify this more explicitly in the paper? While we highlight the dataset dimensions and provide detailed train/validation/test splits, we agree that additional clarity about the architectural composition of the ensembles could be valuable for reproducibility and interpretation. We will expand this section to explicitly describe this.
>
> **Regarding OOD evaluation:**
>
> We appreciate the reviewer pointing out the motivation related to overconfidence in OOD regimes. We do think that single models are usually overconfident OOD. First, we want to clarify that we don’t think that ensembles are always overconfident OOD. It can also be exactly the other way around, that they are unconfident OOD. JUCAL can correct this in both directions. Second, for OOD detection, we don't need to rescale the ratio of epistemic and aleatoric uncertainty, and we don't need absolute uncertainty; we can simply use the pure relative epistemic uncertainty (i.e., classify the datapoints with the largest epistemic uncertainty as OOD without relying on the absolute scale of the uncertainty). We don't think that JUCAL has a large effect on the relative epistemic uncertainties.
>
> **Regarding Greedy-50:**
>
> Greedy-50 follows the definition in Arango et al. (2024). The greedy algorithm performs m=50 selection steps with resampling, always choosing the member that minimizes ensemble NLL. The final ensemble contains exactly m entries, where beneficial members may appear multiple times. Therefore, in cases such as DBpedia Full or SetFit Full with 25 members, Greedy-50 operates over the 25 available models. Additional details can be found in the appendix, the accompanying codebase, and Arango et al. (2024).

---

### Author Response · Authors · 2025-12-03
**Updated Version**

We want to thank all reviewers for their valuable feedback.

We have updated the pdf to address the reviewers' concerns.

We've added calibrate-then-pool. See Figures 4 and 5 on page 10 and Figures 15-22 in Appendix F.2-F.3.

We've added 6 experiments for CNNs trained on image classification datasets. See Figure 5 on page 10 (and half of the figures in Appendix F.2-F.3), where JUCAl also outperforms all its competitors.

We've added multiple further metrics (Brier-Sore, ECE, Misclassification Rate=1-Accuracy). See Figures 4 and 5 on page 10 and Figures 15 and 16 in Appendix F.2. JUCAl shows as good or better overall performance than all considered competitors across all these new metrics suggested by the reviewers and all the metrics that were suggested by us originally.

We've added conformal experiments in Appendix F.3. Note that JUCAL does not need a conformal unseen calibration dataset, as JUCAL only reuses the already seen validation dataset. JUCAL outputs predictive distributions that can be conformalized in a separate step using an unseen calibration dataset. In Appendix F.3, we compare conformalized JUCAL against conformalized versions of its competitors, where we apply APS-conformalization on the same unseen calibration dataset for all competitors. JUCAl shows as good or better overall performance than all considered competitors across all considered conformal metrics (average set size and average log(set size)) (see Figures 17-22 in Appendix F.3).

(We've also added slightly more theoretical explanations (see Appendix I), while JUCAL is not a theory paper but a method paper.)

We've also added another paragraph and Figure 14 at the end of Appendix D.2 to show the influence of $c\_1$ and $c\_2$ as requested by two reviewers.

(Furthermore, we've also added 2 sentences on Limitations and Future work at the end of page 9.)

We think that our updated submission is now significantly stronger with the additional experiments based on valuable constructive feedback from the reviewers.

---

### Meta-Review · Area_Chair_DWAD · 2026-01-13

**Summary:**

Reviewers in general found the paper's idea to be simple, intuitive, and practically useful for post-hoc calibration of ensembles, with strong empirical gains in NLL and predictive set size. However, concerns were raised about (a) overstated claims regarding "joint" or "disentangled" calibration of aleatoric vs epistemic uncertainty without sufficient ablation or theoretical grounding, (b) initially missing or incomplete baselines (notably calibrate-then-pool and conformal variants), (c) limited evaluation scope (mostly NLP/LLM ensembles), and (d) lack of standard calibration metrics and sensitivity/ablation analyses in the original submission. While one reviewer was strongly positive, two reviewers viewed the contribution as incremental and the claims as too strong relative to the evidence.

**Reviewer Concerns:**

The rebuttal and revision do try to address some of the concerns, namely adding calibrate-then-pool, conformalized baselines, CNN/image experiments, additional calibration metrics (ECE, Brier, accuracy), parameter sensitivity plots, and improved discussion of ensemble composition and greedy selection. These strengthen the experimental parts and resolve most baseline/metric-related criticisms.

That being said, some reviewers remain unconvinced that the method truly "disentangles" aleatoric and epistemic uncertainty, viewing JUCAL instead as a flexible two-parameter calibration heuristic optimized for NLL. While the authors clarified this intent and softened claims, concerns/skepticism may persist about the strength of the conceptual contribution and the limited theoretical justification beyond proper scoring rules.

**Reviewer Scores:**

Reviewer J1qw (8) was unlikely to increase the already high score further (though the requested baselines, metrics, sensitivity analyses, and clarifications were added).

Reviewer wMgH (4) could possibly have increased score to a borderline accept, given the addition of calibration metrics, ablations, conformal experiments, and broader evaluation, though still viewing the contribution as incremental.

Reviewer GYZz (2) could possibly have raised the score by a but not to a sufficiently high score because doubts about disentanglement and conceptual novelty likely remain.

Reviewer iK9t (4) could possibly have increased score to weak accept, as most requested additions (ECE, accuracy, CNN experiments, limitations) were incorporated and clarified.

But despite this, the paper lacks an overall consensus for a acceptance.

---

### Decision · Program_Chairs · 2026-01-26

Reject